# The dynamics and geometry of choice in the premotor cortex

Mikhail Genkin[1], Krishna V. Shenoy[2,3], Chandramouli Chandrasekaran[4,5,6,7] & Tatiana A. Engel[1,8 ✉]

The brain represents sensory variables in the coordinated activity of neural populations, in which tuning curves of single neurons define the geometry of the population code[1,2]. Whether the same coding principle holds for dynamic cognitive variables remains unknown because internal cognitive processes unfold with a unique time course on single trials observed only in the irregular spiking of heterogeneous neural populations[3–8]. Here we show the existence of such a population code for the dynamics of choice formation in the primate premotor cortex. We developed an approach to simultaneously infer population dynamics and tuning functions of single neurons to the population state. Applied to spike data recorded during decision-making, our model revealed that populations of neurons encoded the same dynamic variable predicting choices, and heterogeneous firing rates resulted from the diverse tuning of single neurons to this decision variable. The inferred dynamics indicated an attractor mechanism for decision computation. Our results reveal a unifying geometric principle for neural encoding of sensory and dynamic cognitive variables.

Tuning curves of single neurons for sensory variables determine the geometry of the population code[1,2]. This coding principle was established by mapping out changes in trial-averaged firing rates of neurons in response to varying parameters of sensory stimuli[1,2,9]. For example, the orientation of a visual stimulus is a one-dimensional circular variable encoded in the primary visual cortex, where neural population responses organize on a ring mirroring the encoded variable[2,10] (Fig. 1a). The orientation tuning curves of single neurons jointly define the embedding shape of this ring in the population state space, that is, the geometry of the population code[1] (Fig. 1a, Extended Data Fig. 1a and Supplementary Note 1.1).

However, whether the same geometric coding principle holds for dynamic cognitive variables is unknown. Internal cognitive processes (for example, decision-making or attention) are not directly observable and unfold with a unique time course on single trials in sparse and irregular spiking of neural populations[3–8]. Thus, dynamic cognitive computations cannot be revealed by averaging neural responses over repeated trials. Moreover, individual neurons in association brain areas show diverse temporal response profiles during cognitive tasks[11–15], and the widespread assumption is that this heterogeneity reflects complex dynamics involved in cognition[16,17], implying that neural encoding of dynamic cognitive variables follows a fundamentally different principle than for sensory variables (Extended Data Fig. 1b and Supplementary Note 1.1).

Contrary to this view, we hypothesized that the complexity arises from the same coding principle as in sensory areas: the neural population dynamics encode simple cognitive variables, whereas individual neurons have diverse tuning to the cognitive variable, similar to neural tuning curves for sensory stimuli (Fig. 1b, Extended Data Fig. 1c

and Supplementary Note 1.1). To test our hypothesis, we developed a computational approach to simultaneously infer neural population dynamics on single trials and non-linear tuning functions of individual neurons to the unobserved population state. Two crucial technical advances within this approach make testing our hypothesis possible. First, we performed non-parametric inference over a continuous space of models to discover equations governing population dynamics directly from data[18,19], unlike previous methods that tested a small discrete set of models[4,20,21], without guarantees that any of these a priori chosen models faithfully reflect neural dynamics[22]. Second, the inference of non-linear tuning functions allows us to reconcile the diversity of single-neuron responses with the population-level encoding of a low-dimensional cognitive variable. By contrast, previous methods assume a rigid monotonic relationship between firing rates of all neurons and latent states and thus capture population dynamics with more latent dimensions, which may not directly correspond to the encoded cognitive variable[23–27].

We applied our approach to neural population activity recorded from the primate dorsal premotor cortex (PMd) during perceptual decision-making[11], a cognitive computation described by a decision variable reflecting the dynamics of choice formation on single trials[28,29]. The neural representation of the decision variable remains unknown as its unique trajectories on single trials are not observable[3,4], and decision-related responses of cortical neurons are complex and heterogeneous[8,11,20]. Our hypothesis states that neural population dynamics encode a one-dimensional decision variable, and heterogeneous neural responses arise from diverse tuning of single neurons to this decision variable (Fig. 1b). Using our computational approach, we provide three lines of evidence for our hypothesis: in dynamics of single

[1]Cold Spring Harbor Laboratory, Cold Spring Harbor, NY, USA. [2]Howard Hughes Medical Institute, Stanford University, Stanford, CA, USA. [3]Department of Electrical Engineering, Stanford University, Stanford, CA, USA. [4]Department of Anatomy & Neurobiology, Boston University, Boston, MA, USA. [5]Department of Psychological and Brain Sciences, Boston University, Boston, MA, USA. [6]Center for Systems Neuroscience, Boston University, Boston, MA, USA. [7]Department of Biomedical Engineering, Boston University, Boston, MA, USA. [8]Princeton Neuroscience Institute, Princeton University, Princeton, NJ, USA. ✉e-mail: tatiana.engel@princeton.edu

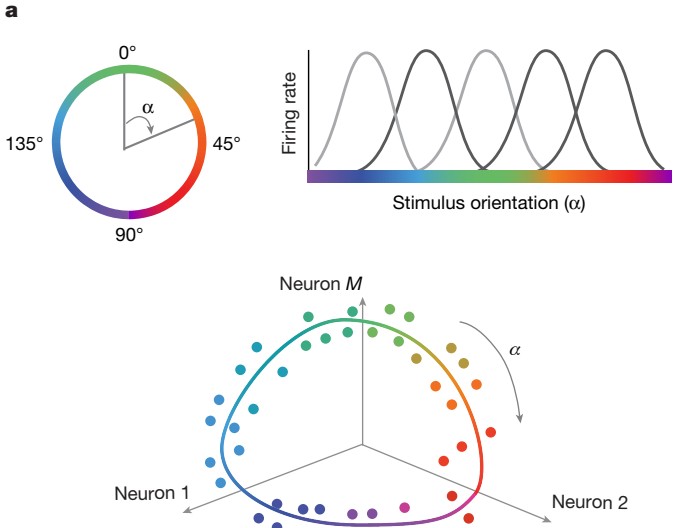

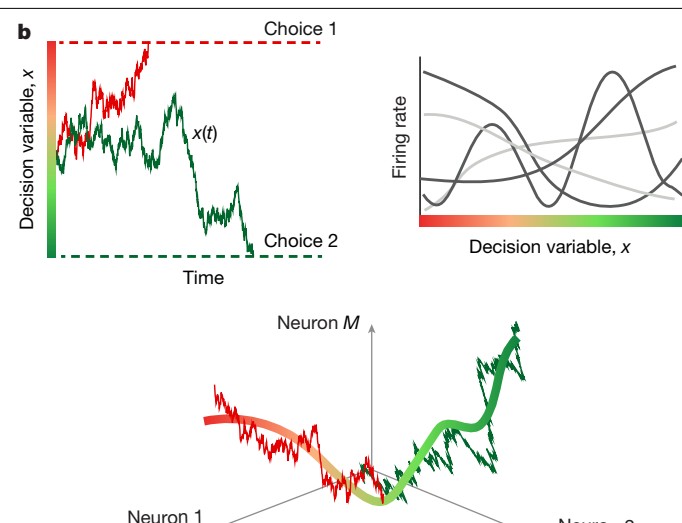

**Fig. 1 | Neural population codes for sensory and cognitive variables.**
**a**, Orientation of a visual stimulus is a one-dimensional circular variable $\alpha$ (top left). Single neurons in the primary visual cortex encode the orientation of a stimulus with bell-shaped tuning curves, which describe the trial-averaged firing rate of a neuron as a function of the stimulus orientation (top right). In the population state space, these neural responses form a ring mirroring the encoded variable (bottom; the dots denote trial-averaged population responses to different stimulus orientations indicated by colour, with the scatter illustrating estimation noise due to a finite number of trials; the line indicates the idealized noise-free ring manifold encoding the stimulus orientation). **b**, We hypothesize that the same geometric coding principle holds for dynamic cognitive variables. Specifically, a decision variable $x(t)$ is a one-dimensional variable representing the dynamics of choice formation on single trials (top left; trajectories coloured by the final choice). Single neurons may encode the decision variable with diverse tuning functions, which describe the instantaneous firing rate of a neuron on single trials as a function of the decision variable value (top right). During decision formation, neural population responses evolve along a one-dimensional manifold encoding the decision variable, which is embedded in a high-dimensional neural population state space (bottom; noisy lines illustrate stochastic trajectories of the decision variable on two example trials coloured by choice, and the solid line indicates the idealized noise-free decision manifold). The tuning curves of all neurons jointly define the embedding shape of the decision manifold in the population state space, that is, the geometry of the neural population code for choice.

neurons, neural population dynamics and their correspondence with animal choices.

## Neural recordings during decision-making

We analysed spiking activity recorded with linear multi-electrode arrays from the PMd of two monkeys performing a decision-making task[11] (Fig. 2a). The monkeys discriminated the dominant colour in a static checkerboard stimulus composed of red and green squares and reported their choice by touching the corresponding left or right target when ready (a reaction-time task). We varied the stimulus difficulty across trials by changing the proportion of the same-coloured squares in the checkerboard and grouped trials into four stimulus conditions according to the response side indicated by the stimulus (left versus right) and stimulus difficulty (easy versus hard; Fig. 2a; see Methods, 'Behaviour and electrophysiology').

Many single neurons in our recordings had decision-related responses with trial-averaged firing rates separating according to the chosen side (Fig. 2b). Although some neurons showed canonical firing rates ramping up or down with a slope dependent on the stimulus difficulty, most neurons exhibited heterogeneous temporal response profiles (Fig. 2b), seemingly incompatible with our hypothesis that all these neurons encode the same dynamic decision variable.

## Flexible inference framework

To test our hypothesis, we developed a flexible modelling framework that dissociates the dynamics and geometry of neural representations and enables estimating both simultaneously in data (Fig. 2c; see Methods, 'Flexible inference framework'). We modelled neural activity on single trials as arising from a dynamic latent variable $x(t)$. Each neuron $i$ has a unique tuning function $f_i(x)$ to this latent variable, analogous to tuning curves of single neurons to sensory stimuli (Fig. 1). The tuning functions of all neurons jointly define the geometry of trajectories traced by neural activity through the population state space on single trials (Fig. 1b, Extended Data Fig. 1c and Supplementary Note 1.1). The dynamics of the latent variable $x(t)$ along these trajectories are governed by a general non-linear dynamical system equation[18,19]:

$$\dot{x} = -D\frac{d\Phi(x)}{dx} + \sqrt{2D}\,\xi(t). \tag{1}$$

Here $\Phi(x)$ is a potential function that defines deterministic forces in the latent dynamical system, and $\xi(t)$ is a Gaussian white noise with magnitude $D$ that accounts for stochasticity of latent trajectories. Our modelling framework can generate data with identical dynamics but different geometry, or vice versa (Extended Data Fig. 2), dissociating the dynamics of the latent variable $x(t)$ from the geometry of its representation within the population state space. By contrast, trial-averaged firing rates conflate the dynamics and geometry of neural representations; hence, the geometry of the trial-averaged population activity does not uniquely define dynamics on single trials[4,30] (Extended Data Fig. 3).

To model decision-related activity, $x(t_0)$ was sampled from the distribution $p_0(x)$ of initial states at the beginning of each trial, and the trial terminated when $x(t)$ reached one of the decision boundaries in the latent space[19] (Fig. 2c). We modelled spikes of each neuron $i$ as an inhomogeneous Poisson process with the instantaneous firing rate $\lambda(t) = f_i(x(t))$ that depends on the current latent state $x(t)$ via the tuning function $f_i(x)$ (Fig. 2c). In our model, $\Phi(x)$, $p_0(x)$ and tuning functions $f_i(x)$ of all neurons are continuous functions that can take any non-linear shapes, enabling flexible discovery of both the low-dimensional latent dynamics and the non-linear geometry of single-trial trajectories in the population state space.

We simultaneously inferred the functions $\Phi(x)$, $p_0(x)$, $f_i(x)$ and the noise magnitude $D$ from spike data by maximizing the model likelihood

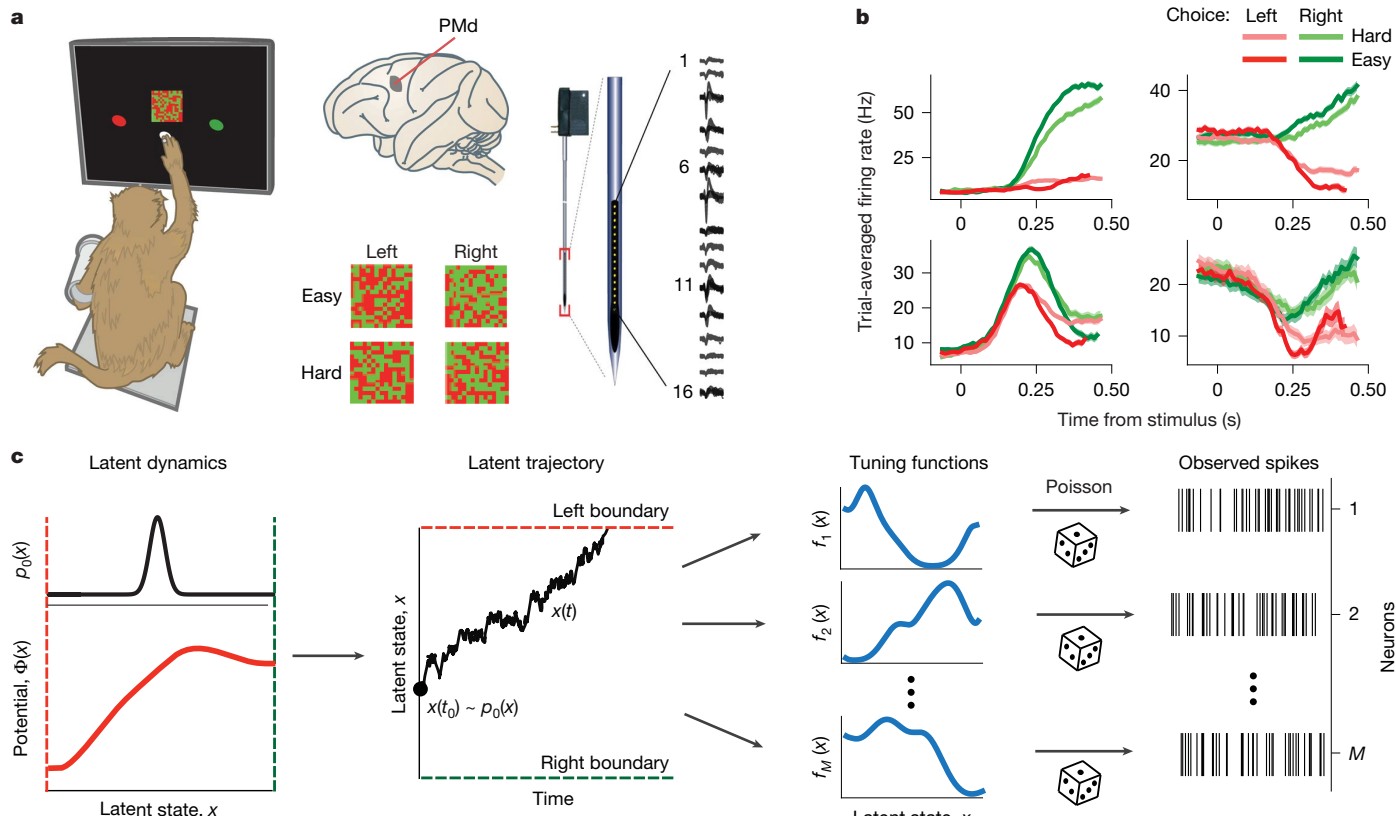

**Fig. 2 | Recording and modelling spiking activity during decision-making.** **a**, Monkeys discriminated the dominant colour in a static checkerboard stimulus composed of red and green squares and reported their choice by touching the corresponding target (left). While monkeys performed the task, we recorded spiking activity with 16 channel multi-electrode arrays from the PMd (right). Trial conditions varied by the response side indicated by the stimulus (left versus right) and stimulus difficulty (easy versus hard; bottom middle). The illustrations of the monkey, checkerboard, electrode array and brain were adapted from ref. 11, Springer Nature Limited. **b**, Trial-averaged firing rates of four example neurons sorted by the chosen side and stimulus difficulty. Although some neurons showed canonical ramping responses (top row), other neurons showed heterogeneous temporal response profiles (bottom row). The error bars are s.e.m. over trials. **c**, A framework for simultaneous inference of neural population dynamics and tuning functions of single neurons to the latent population state, which jointly define the non-linear geometry of neural representations in the population state space. We modelled neural population dynamics with the latent dynamical system in equation (1), in which the deterministic flow field arises from a potential $\Phi(x)$ (bottom left) and stochasticity is driven by a Gaussian white noise. On each trial, the latent trajectory $x(t)$ starts at the initial state $x(t_0)$ (middle left; black dot) sampled from the probability density $p_0(x)$ (top left). The trial ends when the trajectory reaches one of the decision boundaries corresponding to the left and right choice (middle left; red and green dashed lines). The observed spikes (right) of each neuron follow an inhomogeneous Poisson process with time-varying firing rate that depends on the latent variable $x(t)$ via neuron-specific tuning functions $f_i(x)$ (middle right).

(Supplementary Fig. 1 and Supplementary Methods 2; see Methods, 'Flexible inference framework'). Our modelling framework accurately identified ground-truth dynamics in synthetic data (Supplementary Figs. 1–5) and we further validated its accuracy in experimental data with the ground truth established through optogenetic perturbations[31] (Extended Data Fig. 4 and Supplementary Note 1.2).

## Decision dynamics in single neurons

First, we examined decision-related dynamics of single neurons by fitting a separate model to spikes of each neuron ($n = 128$ for monkey T and $n = 88$ for monkey O; see Methods, 'Selection of units for the analysis') in each stimulus condition. The inferred tuning functions $f_i(x)$, initial state distribution $p_0(x)$ and $D$ were similar across conditions and only the potential shapes were different (Extended Data Figs. 5 and 6). Stimulus independence of $p_0(x)$ is expected as stimulus information is not available before stimulus onset. The stability of tuning functions $f_i(x)$ indicates that stimulus affects only the dynamics of the decision variable but not the geometry of its representation in neural activity.

To test for the invariance of tuning functions, we performed shared optimization in which we fitted the model to all available trials and restricted $f_i(x)$, $p_0(x)$ and $D$ to be the same and only allowed the potential $\Phi(x)$ to differ across stimulus conditions. If tuning differed across conditions, the model with shared tuning functions would fit worse than the model with separate tuning functions in each condition. The likelihood was only slightly lower for the model with shared than separate tuning functions (Extended Data Fig. 7a; $\log(\mathcal{L}_{\text{shared}}/\mathcal{L}_{\text{separate}})$; median (Q1–Q3) −6.10 (−17.45 to 1.96), $n = 36$ for monkey T; −18.60 (−31.27 to −6.58), $n = 16$ for monkey O) and within the range obtained on synthetic data from the ground-truth model with shared tuning functions (Extended Data Fig. 7a; $\log(\mathcal{L}_{\text{shared}}/\mathcal{L}_{\text{separate}})$; median (Q1–Q3) 1.33 (−8.75 to 3.68), $n = 6$ for the shared ground-truth model). Furthermore, the inferred tuning functions were similar between shared and separate models, as indicated by a high value of their correlation coefficient (Extended Data Fig. 7b; median (Q1–Q3) 0.91 (0.81–0.93), $n = 36$ for monkey T; 0.94 (0.91–0.95), $n = 16$ for monkey O), confirming that tuning functions are consistent across stimulus conditions.

We therefore used shared optimization in further analyses, because it maximally leverages available data to produce more accurate inference (Supplementary Figs. 2–4). The model fit converged for most neurons (117 out of 128 neurons (91%) for monkey T and 67 out of 88

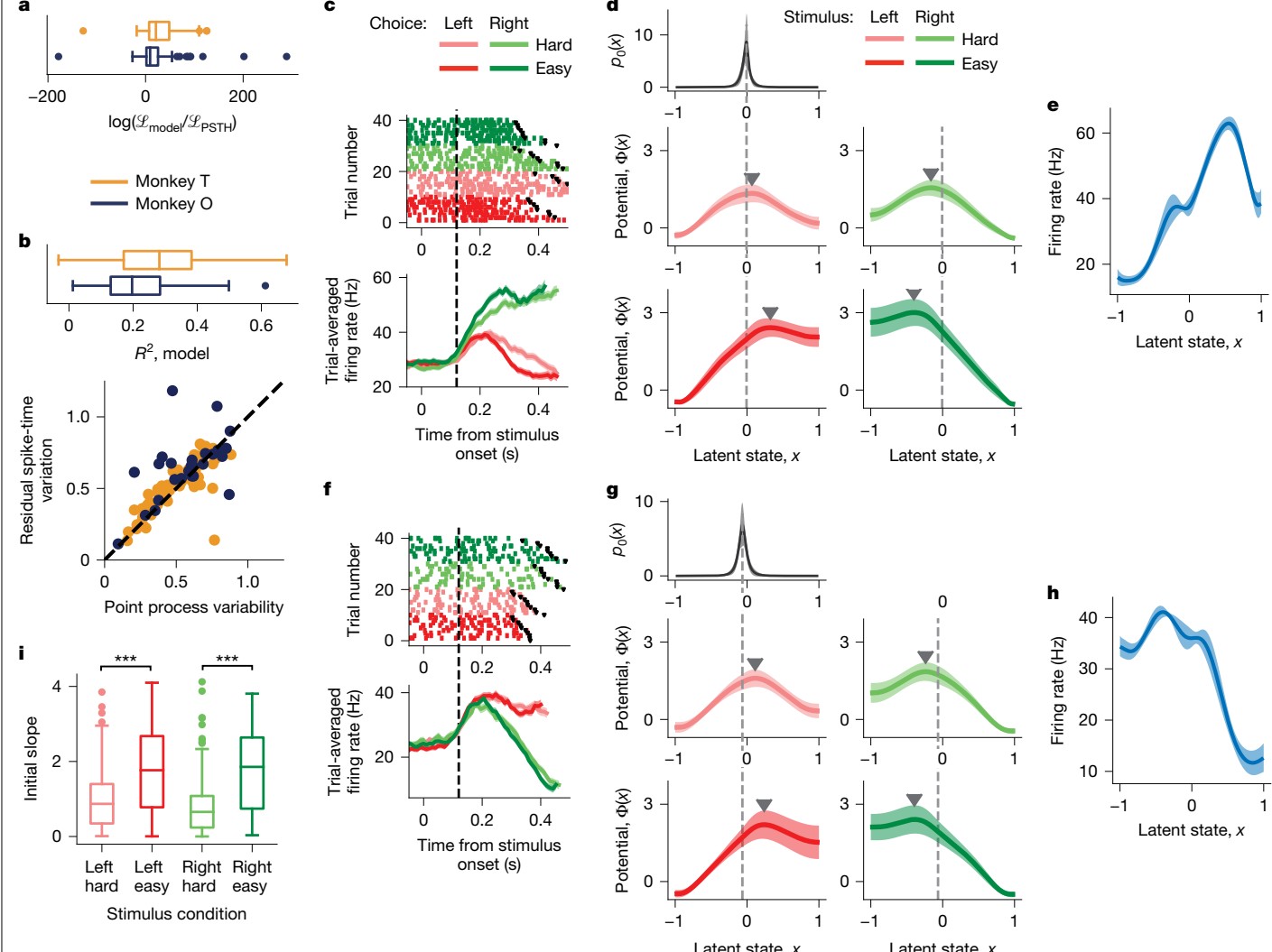

**Fig. 3 | Decision dynamics in single neurons. a**, The log-likelihood ratio of the single-neuron model relative to the peristimulus time histogram (PSTH) across neurons ($n = 117$ for monkey T and $n = 67$ for monkey O). **b**, Spike-time variation explained by the single-neuron model (top; $n = 111$ for monkey T and $n = 50$ for monkey O). The residual spike-time variation unexplained by the model ($y$ axis) correlates with the independently estimated point process variability ($x$ axis; bottom; $n = 64$ for monkey T and $n = 27$ for monkey O). **c**, Spiking activity (top; 40 example trials) and trial-averaged firing rates (bottom; PSTH) sorted by the chosen side and stimulus difficulty for an example neuron. The coloured dots mark spikes, and black dots indicate reaction time. The error bars are s.e.m. over trials. The time window used for model fitting starts at 120 ms after the stimulus onset (vertical dashed line) and extends until the reaction time on

each trial. **d**, Potentials discovered from spikes of the example neuron in **c** show a single barrier (marked by triangles) in all stimulus conditions (middle and bottom). The inferred initial state distribution $p_0(x)$ shared across conditions (top) peaks near the top of the linear slope of the potential (dashed vertical lines). **e**, The inferred tuning function shared across stimulus conditions for the neuron in **c**. The error bars in **d**,**e** are s.d. over 10 bootstrap samples. **f**–**h**, Same as **c**–**e** for another example neuron. **i**, The distribution across neurons of the potential slope at the maximum of $p_0(x)$ for four stimulus conditions (***$P < 10^{-10}$, $n = 184$, Wilcoxon signed-rank test). In the boxplots (**a**,**b**,**i**), the centre lines indicate medians, the boxes span the 25–75th percentiles, the whiskers extend to the nearest of 1.5 × the interquartile range or the most extreme data point, and outliers beyond the whiskers are shown as dots.

neurons (76%) for monkey O), which were used in further analyses (Methods, 'Outcomes of model fitting').

Our model produced significantly higher likelihood than the trial-averaged firing rate computed for each chosen side and stimulus difficulty (Fig. 3a; log($\mathscr{L}_{model}/\mathscr{L}_{PSTH}$), median (Q1–Q3) 21.2 (7.27–48.5), $P < 10^{-10}$, $n = 117$ for monkey T; 9.39 (1.73–25.6), $P = 1.3 \times 10^{-6}$, $n = 67$ for monkey O, Wilcoxon signed-rank test; see Methods, 'Evaluating model performance'). This result indicates that single-trial firing rates deviate considerably from their trial average, and our model successfully captured this variation. Furthermore, our model explained a substantial fraction of the total variation in spike times on single trials (Fig. 3b; coefficient of determination $R^2$; median (Q1–Q3) 0.28 (0.17–0.38) for monkey T; 0.20 (0.13–0.28) for monkey O; see Methods, 'Spike-time $R^2$'). Theoretically, the $R^2$ value cannot reach 1 because our model

predicts single-trial firing rates, leaving the point process variability unexplained. However, if the firing rate prediction is correct on each trial, we expect the point process variability to match the residual spike-time variation unexplained by the model. The point process variability estimated with an independent method[32] correlated tightly with the residual variation unexplained by our model (Fig. 3b; $r = 0.80$, $P < 10^{-10}$, $n = 64$ for monkey T; $r = 0.73$, $P = 1.4 \times 10^{-5}$, $n = 27$ for monkey O, Pearson correlation coefficient), indicating that our model accounted for nearly all explainable variance in firing rates on single trials.

Our model revealed that despite heterogeneous trial-averaged responses, single neurons showed remarkably consistent dynamics on single trials (Fig. 3c–h and Extended Data Fig. 8), which provides the first line of evidence for our hypothesis. In all stimulus conditions, the

inferred potentials displayed the same features: a nearly linear slope towards the decision boundary corresponding to the correct choice and a single potential barrier separating it from the boundary corresponding to the incorrect choice (Fig. 3d,g). The inferred distribution of initial states $p_0(x)$ was narrow and centred near the top of the linear slope, indicating that latent trajectories evolve smoothly towards the correct choice but have to overcome the potential barrier towards the incorrect choice (Fig. 3d,g). The potential shapes were highly consistent across neurons, as indicated by a high value of their correlation coefficient in each condition (0.86 ± 0.16 for monkey T, and 0.88 ± 0.14 for monkey O, mean ± s.d. across neurons and conditions; see Methods, 'Outcomes of model fitting'). For easy stimulus conditions, the potentials had a higher barrier and steeper slope than for hard conditions (Fig. 3d,g,i; easy versus hard; $P < 10^{-10}$, $n = 184$ for left stimulus; $P < 10^{-10}$, $n = 184$ for right stimulus, Wilcoxon signed-rank test), predicting more latent trajectories reaching the correct choice boundary and faster reaction times, consistent with the behaviour of animals. The heterogeneous trial-averaged responses (Fig. 3c,f) resulted from different shapes of the inferred tuning functions (Fig. 3e,h).

These results reject the idea that decision-related dynamics differ across neurons[20], which would correspond to diverse shapes of the potential $\Phi(x)$. Instead, we found that the overwhelming majority of single neurons follow the same dynamics described by a single-barrier potential and diverse tuning functions account for the heterogeneity of their trial-averaged firing rates.

## Decision dynamics in neural populations

Single neurons showed the same dynamics, and next we examined how these dynamics were organized in the population. One possibility is that all neurons follow the same trajectory $x(t)$ on each trial, indicating that the entire population encodes the same latent dynamical variable as we hypothesized (Fig. 1b). Alternatively, individual neurons may follow distinct trajectories on single trials even if their dynamics are described by the same potential, in which case different neurons can be at different latent states at the same time, for example, evolving towards opposite choice boundaries on the same trial[4]. To test these possibilities, we fitted our model to spikes of neural populations recorded simultaneously in the same session (15 populations for each monkey; see Methods, 'Selection of units for the analysis'). The population model assumes that all neurons share the same latent variable $x(t)$ (Fig. 2c) and hence has less freedom to explain neural responses than single-neuron models fitted to spikes of each neuron separately. If single-trial dynamics are not shared across all neurons, the population model would fit worse than single-neuron models.

The shared population model fit converged for most sessions (11 out of 15 sessions (73%) for monkey T and 13 out of 15 sessions (87%) for monkey O), which were used in further analyses (see Methods, 'Outcomes of model fitting'). The likelihood was not significantly different between population models fitted with shared and separate tuning functions (Extended Data Fig. 7c; log($\mathscr{L}_{shared}/\mathscr{L}_{separate}$); median (Q1–Q3) 13.5 (−35.6 to 72.7), $P = 0.58$, $n = 11$ for monkey T; −4.6 (−18.8 to 18.8), $P = 0.95$, $n = 14$ for monkey O, Wilcoxon signed-rank test), reinforcing the invariance of tuning functions. We therefore used the shared population model in further analyses.

We compared the performance of the population and single-neuron models in three ways. First, the population model had similar or higher likelihood than the single-neuron model (Fig. 4a; log($\mathscr{L}_{population}/\mathscr{L}_{singleneuron}$); median (Q1–Q3) 190.4 (91.8–290.3), $P = 0.001$, $n = 11$ for monkey T; 4.0 (−35.6 to 9.4), $P = 0.74$, $n = 13$ for monkey O, Wilcoxon signed-rank test; see Methods, 'Evaluating model performance'). Second, the population and single-neuron models explained a similar fraction of the total variation in spike times on single trials ($R^2$, median (Q1–Q3); population 0.25 (0.15–0.38), single neuron 0.27 (0.19–0.37), $P = 0.11$, $n = 80$ for monkey T; population 0.15

(0.09–0.30), single neuron 0.19 (0.09–0.26), $P = 0.015$, $n = 32$ for monkey O, Wilcoxon signed-rank test). Finally, we used the most stringent leave-one-neuron-out validation, in which we predicted activity of one neuron from the latent variable $x(t)$ inferred from spikes of all other neurons in the population (see Methods, 'Evaluating model performance'). The log-likelihood was higher for the population model in the leave-one-neuron-out validation than for the neuron's own trial-averaged firing rate (Fig. 4a; log($\mathscr{L}_{LONO}/\mathscr{L}_{PSTH}$); median (Q1–Q3) 48.9 (24.5–88.6), $P < 10^{-10}$, $n = 89$ for monkey T; 6.5 (−3.4 to 11.9), $P = 0.07$, $n = 59$ for monkey O, Wilcoxon signed-rank test). Together, these results show that the population model explained neural activity as well as or better than the single-neuron model, supporting our hypothesis that the entire population encodes the same latent dynamical variable on single trials.

As the population model predicted spikes as accurately as single-neuron models, it unsurprisingly revealed dynamics and geometry consistent with single-neuron results (Fig. 4b–d and Supplementary Fig. 6). The inferred tuning functions were similar between the population and single-neuron models, as indicated by a high value of their correlation coefficient (0.89 ± 0.12, $n = 85$ for monkey T; 0.89 ± 0.15, $n = 46$ for monkey O, mean ± s.d. across neurons). For all stimulus conditions, the potential had a single barrier (Fig. 4c), and heterogeneous single-neuron responses (Fig. 4b) were captured in non-linear tuning functions (Fig. 4d). The potential shapes were highly consistent across all populations, as indicated by a high value of their correlation coefficient in each condition (0.93 ± 0.09 for monkey T and 0.88 ± 0.14 for monkey O, mean ± s.d. across populations and conditions; see Methods, 'Outcomes of model fitting'). Moreover, the model likelihood significantly decreased when replacing the tuning functions, potentials or both with linear approximations (Extended Data Fig. 9), emphasizing that neural responses have non-linear dynamics and non-linear geometry. The consistency of potential shapes and the high fit quality for the population model provide the second line of evidence for our overall hypothesis.

## Predicting choice from latent dynamics

Finally, we tested how the dynamic variable encoded by PMd populations related to the decision-making behaviour. Our unsupervised models of neural dynamics are fitted without access to the choices of the animal. A correspondence between the inferred latent trajectories and those choices would indicate that the identified dynamics reflect single-trial decision formation.

We used our models to predict the choices of animals from neural activity. On each trial, we decoded the latent trajectory $x(t)$ from spikes and predicted the choice as the boundary to which this trajectory converged at the reaction time (see Methods, 'Predicting choice from neural activity'). Both single-neuron and population models predicted the choices of animals significantly above chance (Fig. 4e; balanced accuracy, median (Q1–Q3); single-neuron model 68.7% (61.0–79.8%), $P < 10^{-10}$, $n = 85$, population model 89.9% (83.4–96.4%), $P = 0.001$, $n = 11$ for monkey T; single-neuron model 59.2% (55.7–63.6%), $P < 10^{-10}$, $n = 46$, population model 73.2% (64.9–78.6%), $P = 2 \times 10^{-4}$, $n = 13$ for monkey O, Wilcoxon signed-rank test).

For a comparison, we trained a logistic regression decoder to predict the choices of animals from a vector of spike counts on single trials measured in overlapping 75-ms bins with a 10-ms step (see Methods, 'Predicting choice from neural activity'). Despite the decoder being directly supervised to predict the choice, our unsupervised models predicted choices with higher accuracy than the decoder (Fig. 4e; single-neuron decoder versus model $P = 1.2 \times 10^{-8}$, $n = 85$, population decoder versus model $P = 0.014$, $n = 11$ for monkey T; single-neuron decoder versus model $P = 0.019$, $n = 46$, population decoder versus model $P = 0.001$, $n = 13$ for monkey O, Wilcoxon signed-rank test), suggesting that the latent variable inferred by our models is the dynamic decision variable. Moreover, the population model predicted choices

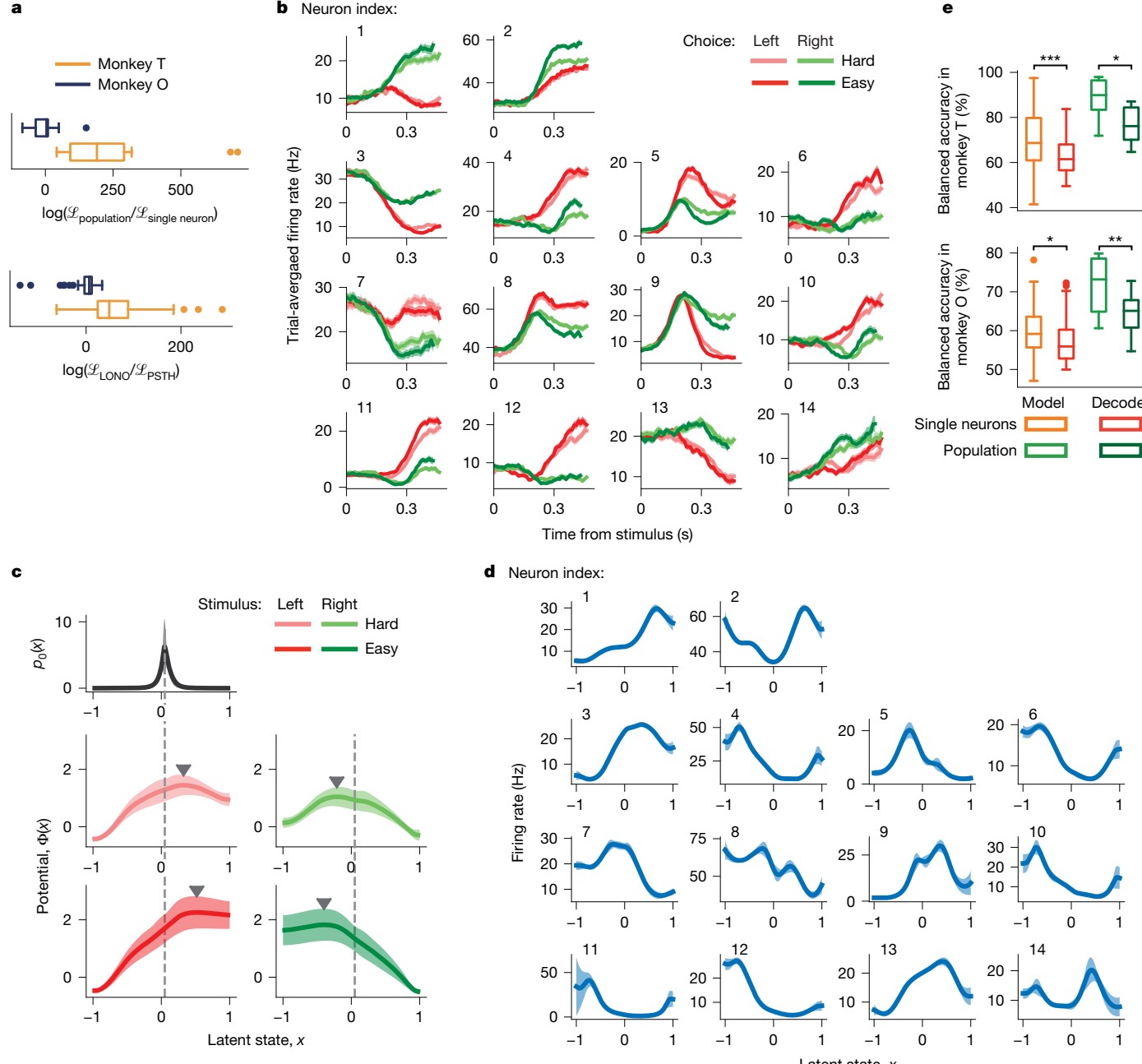

**Fig. 4 | Decision dynamics in neural populations. a**, The log-likelihood ratio of the population model relative to the single-neuron model for all populations (top; $P = 0.001$, $n = 11$ for monkey T; and $P = 0.74$, $n = 13$ for monkey O, Wilcoxon signed-rank test). The log-likelihood ratio of the population model in the leave-one-neuron-out (LONO) validation relative to PSTH across neurons (bottom; $P < 10^{-10}$, $n = 89$ for monkey T; and $P = 0.07$, $n = 59$ for monkey O, Wilcoxon signed-rank test). **b**, Trial-averaged firing rates (PSTHs) sorted by the chosen side and stimulus difficulty for 14 neurons recorded simultaneously on an example session. The error bars are s.e.m. over trials. **c**, Potentials governing the population dynamics discovered from spikes of the population in **b** show a single barrier (marked by triangles) in all conditions (middle and bottom). The inferred $p_0(x)$ shared across conditions (top) peaks near the top of the linear potential slope (vertical dashed lines). **d**, The inferred tuning functions shared

across conditions for the population in **b**. The error bars in **c**,**d** are s.d. over 10 bootstrap samples. **e**, The balanced accuracy of predicting monkey's choice using the single-neuron models (orange), population models (light green), and a logistic regression decoder trained on single-neuron (red) and population (dark green) activity across neurons (for single-neuron models versus decoder; $P = 1.2 \times 10^{-8}$, $n = 85$ for monkey T, and $P = 0.019$, $n = 46$ for monkey O) and across populations (for population models versus decoder; $P = 0.014$, $n = 11$ for monkey T, and $P = 0.001$, $n = 13$ for monkey O, Wilcoxon signed-rank test). *$P < 0.05$, **$P < 0.01$ and ***$P < 0.001$. In the boxplots (**a**,**e**), the centre lines indicate medians, the boxes span the 25–75th percentiles, the whiskers extend to the nearest of 1.5 × the interquartile range or the most extreme data point, and outliers beyond the whiskers are shown as dots.

with higher accuracy than the single-neuron model (Fig. 4e; $P = 1.1 \times 10^{-4}$ for monkey T and $P = 4 \times 10^{-5}$ for monkey O, Mann–Whitney $U$-test), reinforcing that the decision variable is encoded on the population level. These results provide the third and final line of evidence for our overall hypothesis.

In summary, we found that heterogeneous neural populations in the PMd encode the same dynamic decision variable with diverse tuning functions, which define the geometry of the population code for choice. This discovery reveals a unifying geometric principle for neural encoding of sensory and dynamic cognitive variables.

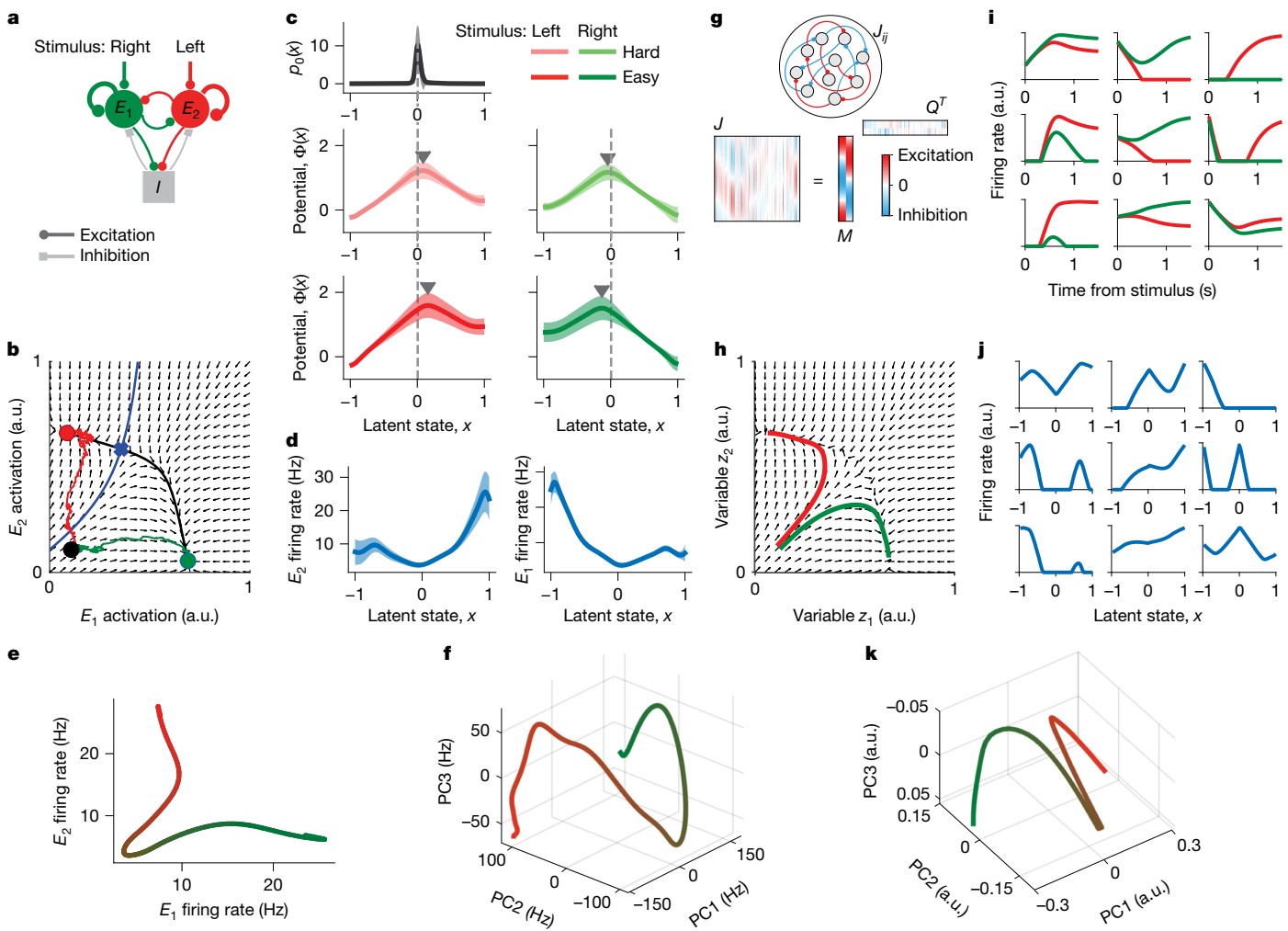

**Fig. 5 | Mechanism for decision computation. a**, The spiking network model of decision-making[33] consists of two excitatory neural pools ($E_1$ and $E_2$) and an inhibitory neural pool (*I*) that mediates winner-take-all dynamics. **b**, The mean-field network dynamics reduce to a two-dimensional flow field[34]. Two stable fixed-point attractors represent two choices (red and green circles), separated by a saddle point (blue cross). The separatrix (blue line) divides the basins of the choice attractors. At the trial start, the network is initialized in a symmetric low-activity state (black dot). Two example trajectories are shown for correct (green) and error (red) trials. a.u., arbitrary units. **c**, Potentials discovered by fitting spikes generated by the attractor network show a single barrier (marked by triangles) in all stimulus conditions (middle and bottom), similar to the PMd data. The inferred initial state distribution $p_0(x)$ shared across conditions (top) peaks near the top of the linear potential slope

(vertical dashed lines). **d**, The inferred tuning functions of two example neurons from each excitatory pool. The error bars in **c**,**d** are s.d. over 10 bootstrap samples. **e**, The decision manifold defined by the tuning functions in the spiking network model. **f**, The decision manifold defined by the tuning functions of all PMd neurons projected on the first three principal components (PCs). **g**, Recurrent network model with rank-two connectivity $J = M \times Q^T$ designed to replicate the classical attractor dynamics. **h**, The flow field governing the mean-field variables $z_1$ and $z_2$ in the rank-two network matches the classical mean-field attractor dynamics. **i**, Firing rates of example units in the rank-two network on the left (red) and right (green) choice trials. **j**, Tuning functions to the decision variable *x* of the example units in **i**. **k**, The decision manifold defined by the tuning functions of all units in the rank-two network projected on the first three PCs.

## Mechanism for decision computation

We discovered that both single-neuron and population activity in the PMd are described by the same dynamics — a potential with a single barrier — qualitatively distinct from the stepping and ramping hypotheses proposed previously[4,19–21] (Supplementary Fig. 5). This potential landscape indicates an attractor mechanism for decision-making, with the potential barrier separating choice-specific attractors[31,33–37].

We therefore used the classical spiking network model of decision-making[33] to establish a mechanistic interpretation of the one-dimensional decision variable and dynamics identified by our model in the PMd. In this network[33], two pools of excitatory neurons receive inputs supporting the left and right choices, and a pool of inhibitory neurons mediates winner-take-all dynamics, such that only one excitatory pool elevates the firing rate on each trial, signalling the

choice of the network (Fig. 5a). The mechanism of decision computation in this network can be understood using a mean-field approximation that reduces the network to a two-dimensional dynamical system in which the activity of two excitatory pools are the dynamic variables[34,38]. In this dynamical system, two stable attractors represent two choice alternatives separated by a saddle point (Fig. 5b). The stable manifold of the saddle is the separatrix that divides the attractor basins.

The attractor network predicted the same dynamics as uncovered by our model in the PMd. At the trial start, the initial network state falls within the basin of the attractor corresponding to the correct choice. On correct trials, the trajectory of the network follows the flow field to reach the correct-choice attractor. By contrast, on error trials, noise drives the trajectory across the separatrix into the incorrect-choice attractor, pushing against the flow field and thus overcoming a potential barrier. To verify this theory, we fitted our population model to

spiking activity generated by simulating the attractor network (see Methods, 'Spiking network model'). From the network spiking activity, the population model inferred potentials with a single barrier in each stimulus condition similar to the potentials discovered from the PMd data (Fig. 5c compared with Figs. 3d,g and 4c).

The attractor network also provided a mechanistic interpretation for the decision variable $x$ and tuning functions $f_i(x)$ in our model. On each trial, network trajectories start at a symmetric low-activity state and follow nearly one-dimensional stereotypic paths to reach either of two choice attractors (Fig. 5b). When varying stimulus difficulty, the shape of these one-dimensional trajectories remains nearly invariant, with only the speed and direction of dynamics along these paths changing, consistent with our findings in the PMd data. These one-dimensional trajectories traced by the network can be parametrized by a single latent variable, corresponding to the decision variable $x$ in our model (Supplementary Note 1.3). The firing rate of neuron $i$ at any point $x$ along the one-dimensional trajectory is given by the tuning function $f_i(x)$. Thus, the decision variable in our model parametrizes one-dimensional trajectories arising during decision-making and tuning functions capture the geometry of these trajectories in the population state space.

We compared the representational geometry of the decision variable between the two-pool attractor network and PMd data. In the two-pool attractor network, the tuning functions inferred by our model were identical for all neurons within each excitatory pool (Fig. 5d), as expected for homogeneous pools. As excitatory neurons have only two types of tuning functions, their responses naturally form a manifold that spans two linear dimensions (Supplementary Note 1.3). We can directly visualize this manifold by plotting the two types of tuning functions against each other (Fig. 5e). The manifold shape corresponds to the paths that network trajectories take from the initial state to the choice attractors (Fig. 5e comapred with Fig. 5b), reinforcing the link between tuning functions and geometry of trajectories arising during decision-making. As PMd neurons had heterogeneous tuning functions, the PMd manifold spanned many linear dimensions. To visualize the decision manifold in the PMd, we projected all tuning functions inferred in 24-well-fitted sessions onto the first three principal components, which explained 56.0%, 26.8% and 5.2% of the variance, respectively. The PMd manifold revealed two diverging branches encoding choice, with a higher-dimensional geometry than in the two-pool attractor network (Fig. 5f).

The complex geometry of the PMd decision manifold can arise from a distributed attractor mechanism, as intuitively explained in low-rank recurrent networks[39–43] (Extended Data Fig. 10). The rank-$k$ connectivity defines $k$ mean-field variables $z_i$ ($i = 1, \dots k$) that govern population dynamics, generating low-dimensional trajectories in the space of synaptic currents. The firing rate is a non-linear function of the synaptic current, resulting in high-dimensional geometry of trajectories in the firing-rate space. To illustrate this mechanism, we designed a rank-two recurrent network that replicates the classical attractor dynamics with distributed connectivity[39] (Fig. 5g; see Methods, 'Low-rank network model'). The flow field governing the mean-field variables $z_1$ and $z_2$ matches that in the classical mean-field attractor network, generating similar trajectories parametrized by a one-dimensional decision variable $x$ (Fig. 5h). In the firing rate space, single units have heterogeneous response profiles (Fig. 5i) and diverse non-linear tuning to the decision variable $x$ (Fig. 5j), with high-dimensional population geometry of the decision manifold (Fig. 5k), as in our PMd data. This example shows that recurrent networks can generate identical dynamics with distinct population geometry (Fig. 5e,k), mechanistically grounding our computational approach for discovering dynamics and geometry directly from data.

## Discussion

We identified the population code for choice in the primate premotor cortex, in which heterogeneous single-neuron activity arises from diverse tuning to a dynamic decision variable encoded in the evolving population state. Our work extends the framework of neural population geometry[1,2,10,44] to transient representations that unfold stochastically on single trials, revealing a unifying geometric principle for the encoding of sensory and dynamic cognitive variables.

This discovery was enabled by two technical advances within our computational approach. First, our modelling framework dissociates the dynamics and geometry of neural representations on single trials and enables identifying both simultaneously in data. Previous models did not allow for simultaneous inference of non-linear tuning functions and a non-linear latent dynamical system[4,20,21,23–27,45,46], requiring more latent dimensions and complex dynamics to capture non-linear population geometry (Supplementary Note 1.4). Second, our framework belongs to a new class of flexible and interpretable models[47–49], whereas most existing methods trade off flexibility for interpretability or vice versa. On the one hand, interpretable models often rely on rigid parametric assumptions that do not permit discovering dynamical laws beyond their a priori constraints[6,23,24,50,51] (Supplementary Note 1.4). On the other hand, flexible high-dimensional recurrent neural networks can approximate any dynamics, but do not yield interpretable low-dimensional flow fields[25–27]. Recurrent neural networks are typically interpreted by linearizing their dynamics around fixed points[12,36,52], providing merely local linear approximations of the dynamics. By contrast, our approach derives a low-dimensional non-linear dynamical system model of neural computations[16] directly from spike data, avoiding inductive biases of intermediate approximation schemes[53].

These technical innovations enabled us to test a hypothesis that heterogeneous responses of single neurons result from their diverse tuning to a decision variable encoded in single-trial population dynamics. Although decision-making has been long hypothesized to arise from attractor[33,34], drift diffusion[28] or stepping dynamics[3,4,20,21], no conclusive evidence has emerged thus far to arbitrate among these alternatives in data. For example, attempts to arbitrate between drift diffusion and stepping dynamics concluded that equal fractions of neurons show each type of dynamics[20], indicating that dynamics are not shared across the entire population. However, all these classic hypotheses postulate monotonically ramping trial-averaged firing rates and fail to account for heterogeneous responses of cortical neurons (Fig. 2b). By contrast, we showed that individual neurons follow the same dynamics on single trials shared across the population, suggesting that previous models may have reached the opposite conclusion due to their inflexible mapping of latent states to firing rates[4,20]. Moreover, our model reveals that behavioural errors arise from deviant dynamics on a manifold with stable geometry, despite distinct geometries of trial-averaged trajectories for correct and error trials (Supplementary Fig. 7 and Supplementary Note 1.5).

We found that PMd neurons have non-linear tuning to the decision variable, contrary to the common assumption that firing rates encode the decision variable linearly[8] or monotonically[4,20,21]. Linear tuning implies that trial-averaged firing rates ramp up or down monotonically, inconsistent with temporal response profiles of many neurons. In addition, linear models have often found that the axis encoding choice in the population state space rotates over time within a trial[50,54]. The rotating choice axes are parsimoniously explained as piecewise linear approximations of the non-linear geometry of the choice manifold that we discovered. Non-linear encoding of cognitive variables may be ubiquitous, as neurons in the hippocampal formation encode space and other abstract variables with non-monotonic tuning functions[55–57].

We found attractor dynamics in the PMd with high-dimensional geometry of choice representation. Low-rank recurrent networks can generate identical dynamics with distinct population geometry[43] (Fig. 5), thus dissociating population dynamics and geometry. Our modelling framework identifies dynamics and geometry of neural representations without imposing an inductive bias towards a specific circuit mechanism that generated them. Such statistical descriptions

reveal principles of neural coding and computation and allow for quantitative comparisons between distinct mechanistic models and experimental data[58].

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

# Methods

## Behaviour and electrophysiology

We analysed an experimental dataset previously described[11,59]. Two male monkeys (T and O, *Macaca mulatta*, 6 and 9 years of age) were used in the experiments. Experimental procedures were in accordance with the US National Institutes of Health (NIH) Guide for the Care and Use of Laboratory Animals, the Society for Neuroscience Guidelines and Policies, and Stanford University Animal Care and Use Committee (8856).

The monkeys were trained to discriminate the dominant colour in a static checkerboard stimulus composed of red and green squares and report their choice by touching the corresponding target. At the start of each trial, a monkey touched a central target and fixated on a cross above the central target. After a short holding period (300–485 ms), red and green targets appeared on the left and right sides of the screen. The colours of each side were randomized on each trial. After another short holding period (400–1,000 ms), the checkerboard stimulus appeared on the screen at the fixation cross and the monkey had to move its hand to the target matching the dominant colour in the checkerboard. Monkeys were free to respond when ready. Monkeys were rewarded for the correct choices and received longer inter-trial delays for the incorrect choices. Hand position was monitored by taping an infrared reflective bead to the index or middle fingers of each hand and used for measurement of speed and to estimate reaction time[60]. We used a real-time system combining xPC Target (v5.3) from MATLAB R2012b to control task timing and state transitions. This system communicated with a separate computer running Psychtoolbox (v3.0.9) on MATLAB R2012b for stimulus display.

The difficulty of the task was parameterized by an unsigned stimulus coherence expressed as the absolute difference between the number of red (R) and green (G) squares normalized by the total number of squares $|R - G|/(R + G)$. We used a 15 × 15 checkerboard, which led to a total of 225 squares. The task was performed with seven different unsigned coherence levels for monkey T and eight levels for monkey O. For each stimulus condition, our analysis required at least a small fraction of incorrect choices so that the neural activity fully explored the decision manifold. Therefore, we only analysed the four most difficult stimulus conditions for each monkey, which had sufficient number of error trials. To obtain sufficient data for the model fitting and validation, we merged these four stimulus conditions into two groups combining two easier conditions into one group and two harder conditions into another group. We refer to these two groups as easy and hard stimulus difficulties. As PMd neurons are selective for the chosen side but not for colour[11], we further divided the trials according to the side indicated by the stimulus (left or right) for each stimulus difficulty (easy or hard), resulting in four analysed conditions in total.

We used the cerebus system (Blackrock Microsystems) and the Central software (v6.0) to record neural activity in the PMd with a linear multi-contact electrode (U-probe) with 16 channels. After rigorous online sorting in Central, we used offline spike sorting through a combination of MATLAB (MatClust v1.7.0.0) and Plexon offline sorter (v3) to identify single neurons and multi-units. The average yield was approximately 16 and approximately 9 neurons per session for monkey T and monkey O, respectively, which primarily were well-isolated single units.

## Selection of units for the analyses

After spike sorting and quality control, we had 546 and 450 single neurons and multi-units recorded from monkeys T and O, respectively. From this dataset, we selected units for our analyses based on three criteria: (1) trial-averaged firing rate traces sorted by the chosen side (left versus right) reach 15 Hz for at least one reach direction at any time between stimulus onset and median reaction time; (2) the total number of trials across all conditions is at least 560; and (3) selectivity index for the chosen side is greater than 0.6 for monkey T and 0.55 for monkey O. The first two criteria ensure that a unit yields a sufficiently large number of spikes for model fitting[18], and the third criterion selects for units with decision-related activity.

For the first criterion, we used trial-averaged firing rate traces aligned to stimulus onset (PSTH) sorted by the chosen side, obtained by averaging over trials the spike counts measured in 75-ms bins sliding at 10-ms steps. For the third criterion, we measured the spike count of each neuron on each trial in a (0.2–0.35 s) window aligned to stimulus onset. Selectivity index was defined as the area under the receiver operating characteristic curve for discriminating left versus right chosen side based on the spike counts. Selectivity index ranges between 0.5 (no choice selectivity) and 1. For each monkey, we imposed a selectivity index threshold at the median across all neurons (0.6 for monkey T and 0.55 for monkey O), leading to selecting half of all neurons in each monkey. This criterion implies that analysed neurons had overall lower choice selectivity in monkey O than in monkey T, because choice selectivity was generally lower for neurons from monkey O in our dataset.

For monkey T and monkey O, 128 and 88 units, respectively, passed all three selection criteria and were used in single-neuron analyses. The majority were well-isolated single neurons (127 out of 128 units (99%) for monkey T, and 76 out of 88 units (86%) for monkey O), and the rest were multi-units. For population analyses, we included sessions that had at least 3 of the selected single units recorded simultaneously, yielding 15 populations for each monkey.

On each trial, we analysed PMd activity from 120 ms after stimulus onset (the appearance of a checkerboard stimulus on the screen) until the reaction time (the hand leaving the central target), which was estimated at the first time after checkerboard onset when speed of the hand was above 10% of the maximum speed for that trial. The delay of 120 ms was chosen to account for the lag in PMd response to the stimulus. We verified that the model fitting results were the same for a 80–120-ms range of delays.

## Flexible inference framework

We implemented our computational framework as a Python package NeuralFlow, which is publicly available online[61]. We modelled latent neural dynamics $x(t)$ as a stochastic non-linear dynamical system defined by the Langevin equation[19] (equation (1)) on the domain $x \in [-1; 1]$. In this equation, the potential $\Phi(x)$ gives rise to a deterministic force flow field $F(x) = -\mathrm{d}\Phi(x)/\mathrm{d}x$. As in one dimension, there are no rotational forces and any force results from the gradient of a potential, representing the flow field via potential does not restrict our model. In practice, we directly optimized the force $F(x)$ and then computed $\Phi(x)$ from the force for visualization. The term $\xi(t)$ is a white Gaussian noise $\langle \xi(t) \rangle = 0$, $\langle \xi(t)\xi(t') \rangle = \delta(t - t')$ with the magnitude $D$. In equation (1), we scaled the potential with $D$, which makes the equilibrium probability density of the Langevin dynamics invariant to the noise magnitude. This parametrization is equivalent to measuring the potential height in units of $D$ and does not restrict the space of dynamical systems spanned by our models. We scaled all potentials by $D$ after fitting for visualization and comparisons across conditions, neurons and populations. At the start of each trial, the initial latent state $x_0$ is sampled from a distribution with probability density $p_0(x)$.

We modelled spikes of each neuron as an inhomogeneous Poisson process with instantaneous firing rate $\lambda(t) = f(x(t))$ that depends on the current latent state via a neuron-specific tuning function $f(x)$. In population models, all neurons follow the same latent dynamics $x(t)$ and each neuron $i$ has a unique tuning function $f_i(x)$ ($i = 1...M$, where $M$ is the number of neurons in the population). In this case, the population dynamics $x(t)$ are shared by all neurons, and the tuning functions $f_i(x)$ jointly define the non-linear embedding of these dynamics into the neural population state space. Thus, the model was specified by a set of continuous functions: the potential $\Phi(x)$, the initial state distribution $p_0(x)$, a collection of tuning functions $\{f_i(x)\}$ and a scalar noise

magnitude $D$. We inferred all model components $\theta = \{\Phi(x), p_0(x), \{f_i(x)\}, D\}$ from spike data $\mathcal{Y}(t)$.

The spike data consisted of multiple trials $\mathcal{Y}(t) = \{Y_k(t)\}$ ($k = 1, 2 \ldots K$, where $K$ is the number of trials), and the time argument in $Y_k(t)$ indicates that the data are a sequence of event times on each trial. For independent trials, the total data likelihood is a product of likelihoods of individual trials

$$\log \mathcal{L}[\mathcal{Y}(t)|\theta] = \sum_{k=1}^{K} \log \mathcal{L}[Y_k(t)|\theta]. \tag{2}$$

Therefore, here we consider data for a single trial $Y_k(t)$ and omit the trial index to simplify notation. For each trial, $Y(t) = \{y_0, y_1, \ldots, y_N, y_E\}$ is a marked point process, that is, a sequence of discrete observation events. Each observation is a pair $y_j = (t_j, i_j)$, where $t_j$ is the time of event $j$ and $i_j$ is the type of this event. The first and last events mark the trial start time $t_0$ and trial end time $t_E$, and the $N$ remaining events ($j = 1, \ldots N$) are the spike observations where $t_j$ is the time of $j$th spike and $i_j$ is the index of the neuron that emitted this spike ($i_j \in \{1, \ldots M\}$, where $M$ is the number of simultaneously fitted neurons). The events are ordered according to their times.

We fitted the model by maximizing the data log-likelihood $\log \mathcal{L}[\mathcal{Y}(t)|\theta]$ over the space of continuous functions[18,19] (Supplementary Methods 2.1). The likelihood for each trial is a conditional probability density of observing the data $Y(t)$ given the model $\theta$ marginalized over all possible latent trajectories:

$$\mathcal{L}[Y(t)|\theta] = \int \mathcal{D}\mathcal{X}(t)\, P(\mathcal{X}(t), Y(t)|\theta). \tag{3}$$

Here $P(\mathcal{X}(t), Y(t)|\theta)$ is a joint probability density of observing the spike data $Y(t)$ and a continuous latent trajectory $\mathcal{X}(t)$ given the model $\theta$, and the path integral is performed over all possible latent trajectories. We omit the conditioning on $\theta$ in subsequent expressions for the probability densities of the data and latent states to simplify notation.

We derived the likelihood functional for non-stationary latent Langevin dynamics as previously described[19]. In brief, in a reaction-time decision-making task, a participant reports the choice as soon as the neural trajectory reaches a decision boundary for the first time. Thus, trials have variable durations defined by the neural dynamics itself, and the latent trajectory always terminates at one of the decision boundaries at the trial end. Accordingly, the likelihood calculation must integrate over all latent paths that terminate at one of the boundaries at the trial end time $t_E$ and do not reach a boundary at earlier times. Two components in our framework account for the statistics of latent trajectories in this case. First, the absorbing boundary conditions ensure that latent trajectories reaching a boundary before the trial end do not contribute to the likelihood. Second, the absorption operator $p(A|x_{t_E})$ enforces that the likelihood includes only trajectories terminating on the boundaries at the trial end time. Without the absorption operator, the likelihood includes all trajectories that terminate anywhere in the latent space and do not reach the domain boundaries before the trial end. Omitting either of these components results in incorrect inference, in which erroneous features arise in the dynamics due to non-stationary data distribution[19].

Here we provide a brief exposition of the likelihood calculation. The detailed mathematical derivation and extensive numerical testing of the non-parametric inference of non-stationary latent Langevin dynamics have been presented in our previous work[19]. In this study, we have introduced non-parametric inference of tuning functions $f_i(x)$ simultaneously with latent dynamics, as described below. To compute the path integral in equation (3), we consider a discretized latent trajectory $X(t) = \{x_{t_0}, x_{t_1}, \ldots, x_{t_N}, x_{t_E}\}$, which is a discrete set of points along a continuous path $\mathcal{X}(t)$ at each of the observation times $\{t_0, t_1, \ldots, t_N, t_E\}$. Once we calculate the joint probability density $P(X(t), Y(t))$ of the discretized

trajectory and data, we can obtain the data likelihood by marginalization over all discretized latent trajectories:

$$\mathcal{L} = \int_{x_{t_0}} \int_{x_{t_1}} \ldots \int_{x_{t_N}} \int_{x_{t_E}} dx_{t_0} \ldots dx_{t_E} P(X(t), Y(t)). \tag{4}$$

We do not bin spikes but process data spike by spike in continuous time. Accordingly, the calculation of the joint probability density $P(X(t), Y(t))$ must account for the spikes observed at times $Y(t) = \{t_j\}$ and also for the absence of spike observations during interspike intervals (ISIs; $t_{j-1}, t_j$). Using the Markov property of the latent Langevin dynamics in equation (1) and conditional independence of spike observations, the joint probability density $P(X(t), Y(t))$ can be factorized[19]:

$$P(X(t), Y(t)) = p(x_{t_0})\left(\prod_{j=1}^{N} p(y_j|x_{t_j})p(x_{t_j}|x_{t_{j-1}})\right)p(x_{t_E}|x_{t_N})p(A|x_{t_E}). \tag{5}$$

Here the terms $p(y_j|x_{t_j})$ represent the probability density of observed spikes, and the terms $p(x_{t_j}|x_{t_{j-1}})$ represent the transition probability density of latent states that accounts for the absence of spike observations during ISIs. Each term $p(y_j|x_{t_j})dt$ is the probability of observing a spike from neuron $i_j$ within infinitesimal $dt$ of time $t_j$ given the latent state $x_{t_j}$, hence $p(y_j|x_{t_j}) = f_{i_j}(x_{t_j})$ by the definition of the instantaneous Poisson firing rate. $p(x_{t_j}|x_{t_{j-1}})$ is the transition probability density from $x_{t_{j-1}}$ to $x_{t_j}$ during the time interval between the adjacent spike observations, which also accounts for the absence of spikes during each ISI in the data. This transition probability density decays with time at a rate given by the Poisson firing rate of all neurons, because it becomes less likely to observe no spikes for longer time intervals. $p(x_{t_0})$ is the probability density of the initial latent state. Finally, the term $p(A|x_{t_E})$ represents the absorption operator, which ensures that only trajectories terminating at one of the domain boundaries at time $t_E$ contribute to the likelihood[19].

The discretized latent trajectory $X(t) = \{x_{t_0}, x_{t_1}, \ldots, x_{t_N}, x_{t_E}\}$ is obtained by marginalizing the continuous trajectory $\mathcal{X}(t)$ over all latent paths connecting $x_{t_{j-1}}$ and $x_{t_j}$ during each ISI. These marginalizations are implicit in the transition probability densities $p(x_{t_j}|x_{t_{j-1}})$ in equation (5). The transition probability density $p(x_{t_j}|x_{t_{j-1}})$ accounts for the drift and diffusion in the latent space and also for the absence of spikes during each interval between adjacent spike observations. This probability density satisfies a modified Fokker–Planck equation, which we derived previously[19]:

$$\frac{\partial p(x, t)}{\partial t} = \left(-D\frac{\partial}{\partial x}F(x) + D\frac{\partial^2}{\partial x^2} - \sum_{i=1}^{M} f_i(x)\right)p(x, t) \equiv -\hat{\mathcal{H}}p(x, t). \tag{6}$$

Here $F(x) = -\Phi'(x)$ is the deterministic potential force, and the term $-\sum_{i=1}^{M} f_i(x)$ leads to the probability decay due to spikes emitted by any neurons in the population, such that $p(x_{t_j}|x_{t_{j-1}})$ includes only trajectories consistent with no spikes emitted between $t_{j-1}$ and $t_j$. The solution of this equation $p(x, t_j) = p(x, t_{j-1})\exp(-\hat{\mathcal{H}} \cdot (t_j - t_{j-1}))$ propagates the latent probability density forwards in time during each ISI. To model the reaction time task, we solved equation (6) with absorbing boundary conditions, which ensure that trajectories reaching a boundary before the trial end do not contribute to the likelihood[19]. In addition, the absorption operator $p(A|x_{t_E})$ in equation (5) enforces that the likelihood includes only trajectories terminating on the boundaries at the trial end time $t_E$[19] (Supplementary Methods 2.1). Together, these two conditions ensure that the likelihood includes only trajectories that reach one of the boundaries for the first time at the trial end time.

To fit the model to data, we derived analytical expressions for the gradients of the model likelihood with respect to each of the model components (Supplementary Methods 2.2). We computed functional derivatives of the likelihood with respect to latent dynamics as previously described[19]. In this study, we computed functional derivatives of the likelihood with respect to tuning functions (Supplementary

Methods 2.2). Instead of directly updating the functions $\Phi(x)$, $p_0(x)$ and $f_i(x)$ we, respectively, update the force $F(x) = -\Phi'(x)$ and auxiliary functions

$$F_0(x) \equiv p_0'(x)/p_0(x), \quad F_i(x) \equiv f_i'(x)/f_i(x). \tag{7}$$

The potential $\Phi(x)$ is obtained from $F(x)$ via

$$\Phi(x) = -\int_{-1}^{x} F(s)\mathrm{d}s + C, \tag{8}$$

where we fixed the integration constant $C$ to satisfy $\int_{-1}^{1} \exp[-\Phi(x)]\mathrm{d}x = 1$. As $C$ is an arbitrary integration constant, a specific choice of $C$ was not important as long as it was the same for all models. The initial state distribution $p_0(x)$ was obtained from the auxiliary function $F_0(x)$ in equation (7) via

$$p_0(x) = \frac{\exp\left(\int_{-1}^{x} F_0(s)\mathrm{d}s\right)}{\int_{-1}^{1} \exp\left(\int_{-1}^{s'} F_0(s)\mathrm{d}s\right)\mathrm{d}s'}. \tag{9}$$

The change of variable from $p_0(x)$ to $F_0(x)$ allowed us to perform an unconstrained optimization of $F_0(x)$, whereas equation (9) ensures that $p_0(x)$ satisfies the normalization condition for a probability density $\int_{-1}^{1} p_0(x)\mathrm{d}x = 1, p_0(x) \geqslant 0$. Finally, the tuning function $f_i(x)$ was obtained from the auxiliary function $F_i(x)$ in equation (7) via

$$f_i(x) = C_i \exp\left(\int_{-1}^{x} F_i(s)\mathrm{d}s\right), \tag{10}$$

where $C_i = f_i(-1)$ is the firing rate at the left domain boundary. This change of variable allowed us to perform an unconstrained optimization of $F_i(x)$, whereas equation (10) ensures the non-negativity of the firing rate $f_i(x) \geqslant 0$. We enforced the positiveness of the noise magnitude $D$ by rectifying its value after each update $D = \max(D, 0)$, and the same for each constant $C_i$.

We derived analytical expressions for the variational derivatives of the likelihood with respect to each continuous function defining the model $\delta\mathscr{L}/\delta F(x)$, $\delta\mathscr{L}/\delta F_0(x)$, $\delta\mathscr{L}/\delta F_i(x)$ and the derivatives of the likelihood with respect to scalar parameters $\partial\mathscr{L}/\partial D$ and $\partial\mathscr{L}/\partial C_i$ (Supplementary Methods 2.2). We evaluated these analytical expressions numerically for the iterative optimization. To compute the likelihood and its gradients numerically, we used a discrete basis in which all continuous functions, such as $F(x)$, are represented by vectors, and the transition, emission and absorption operators are represented by matrices[18,19] (Supplementary Methods 2.1). Thus, equation (5) was evaluated as a chain of matrix–vector multiplications.

## Optimization with ADAM algorithm

We fitted the model by minimizing the negative log-likelihood $-\log\mathscr{L}[\mathcal{Y}(t)|\theta]$ using ADAM algorithm[62] with custom modifications (Supplementary Methods 2.3). The standard ADAM update scales the gradient of each scalar parameter inversely with the running average of the squared magnitudes of its current and past gradients, computed separately for each parameter. As we optimized over continuous functions $F(x)$, $F_0(x)$ and $\{F_i(x)\}$, we scaled their gradients by the running average of the gradient's squared $L^2$-norm defined as $\|\boldsymbol{v}\|_2 = \sum_i v_i^2$. We used the following ADAM hyperparameters: $\alpha = 0.05$ for single neurons, $\alpha = 0.01$–$0.05$ for populations, $\beta_1 = 0.9, \beta_2 = 0.99, \epsilon = 10^{-8}$ for both single neurons and populations (the definitions of hyperparameters are in Supplementary Methods 2.3). We tuned these hyperparameters on synthetic data with known ground truth. For the scalar parameters $D$ and all $\{C_i\}$, we combined ADAM updates with line searches using the L-BFGS-B algorithm (L-BFGS-B method from the scipy.optimize.minimize toolbox). As a line search is computationally expensive, we performed only 30 line searches spaced logarithmically over the 5,000 epochs range, such that most line searches are concentrated at early epochs.

We combined ADAM with mini-batch descent randomly splitting the trials from each condition into 20 batches on each epoch. When we performed shared optimization, we fitted the model to all available trials restricting $F_0(x)$, $\{F_i(x)\}$ and $D$ to be the same and only allowing the potential force $F(x)$ to differ across stimulus conditions. In this case, we performed ADAM updates on all batches pooled across four stimulus conditions (80 batches total) in random order on each epoch. We updated the force $F^l(x)$ that defines the potential in condition $l$ only on batches from this condition, and we updated all shared components $F_0(x)$, $\{F_i(x)\}$, $D$, $\{C_i\}$ on every batch.

We accelerated the optimization algorithm on graphics processing units (GPUs) using cupy library[63]. GPU implementation provides a 5–10-fold acceleration over the CPU implementation with the exact factor depending on the spatial resolution of the discrete basis. The 5–10-fold GPU acceleration can only be provided by scientific-grade GPUs (for example, Tesla V100) that have sufficient number of double-precision streaming multiprocessors.

## Model selection

ADAM optimization produces a series of models across epochs, and we needed a model selection procedure for choosing the optimal model. On early epochs, the fitted models miss some true features of the dynamics due to underfitting, whereas on late epochs, the fitted models develop spurious features due to overfitting to noise in the data. The optimal model is discovered on some intermediate epochs. The standard approach for selecting the optimal model is based on optimizing the ability of the model to predict new data (that is, generalization performance), for example, using likelihood of held-out validation data as a model selection metric[64]. However, optimizing generalization performance cannot reliably identify true features and avoid spurious features when applied to flexible models[18,65], which generalize well despite overfitting[66]. We developed an alternative approach for model selection based on directly comparing features of the same complexity discovered from different data samples[18,19] (Supplementary Methods 2.4). As true features are the same, whereas noise is different across data samples, the consistency of features inferred from different data samples can separate the true features from noise, and model selection based on feature consistency can reliably identify the correct features[18,19].

To compare features discovered from different data samples, we need a metric for feature complexity $\mathcal{M}$. We defined feature complexity as the negative entropy of latent trajectories generated by the model[18,19] $\mathcal{M} = -S[\Phi(x), D, p_0(x); \Phi^R(x), D^R, p_0^R(x)]$. The trajectory entropy[67] is a functional defined as a negative Kullback–Leibler divergence between the distributions of trajectories in the model of interest $\{\Phi(x), D, p_0(x)\}$ and the distribution of trajectories in the reference model $\{\Phi^R(x), D^R, p_0^R(x)\}$. The reference model is a free diffusion in a constant potential ($\Phi^R(x) = \mathrm{const}$) with the same diffusion coefficient $D$ as in the model of interest. We derived the analytical expression for the trajectory entropy for non-stationary Langevin dynamics[19]:

$$\begin{aligned} S\Big[&\Phi(x), D, p_0(x); \Phi^R(x), D, p_0^R(x)\Big] \\ &= -\int \mathrm{d}x p_0(x)\ln\frac{p_0(x)}{p_0^R(x)} - \frac{D}{4}\int_0^\infty \mathrm{d}t \int \mathrm{d}x F^2(x)p(x,t). \end{aligned} \tag{11}$$

We chose the initial distribution $p_0^R(x)$ for the reference model to be uniform. We derived an expression for efficient numerical evaluation of equation (11) taking the integral over time analytically[19] (Supplementary Methods 2.4). Qualitatively, feature complexity reflects the structure of the potential $\Phi(x)$: potentials with more structure have higher feature complexity. The reference model with constant potential has zero feature complexity. During model fitting, the feature complexity consistently grows throughout the optimization epochs[19].

We compared models discovered from two non-intersecting halves of the data $\mathcal{D}_1$ and $\mathcal{D}_2$ to evaluate the consistency of their features

(Supplementary Fig. 1). We performed the ADAM optimization independently on each data split to obtain two series of models $\theta_n^1 = \{\Phi_n^1, D_n^1, p_{0,n}^1(x)\}$ and $\theta_n^2 = \{\Phi_n^2, D_n^2, p_{0,n}^2(x)\}$ (where $n = 1, 2...5,000$ is the epoch number) fitted on $\mathcal{D}_1$ and $\mathcal{D}_2$, respectively. We measured feature complexity of these models, $\mathcal{M}_n^1$ and $\mathcal{M}_n^2$, and quantified the consistency of features of the same complexity between models fitted on different data splits. We quantified the consistency of features between two models by evaluating Jensen–Shannon divergence ($D_{\mathrm{JS}}$) between their time-dependent probability densities over the latent space[19] (Supplementary Methods 2.4). At low and moderate feature complexity, the models contain true features of the dynamics in the data and their features agree between data splits reflected in low $D_{\mathrm{JS}}$ values. At high feature complexity, the models overfit to noise and contain spurious features that do not replicate between data splits, resulting in large $D_{\mathrm{JS}}$ values. To find the optimal feature complexity, we set the threshold $D_{\mathrm{JS,thres}} = 0.0015$ and selected $\mathcal{M}^*$ as the maximum feature complexity for which $D_{\mathrm{JS}} \leq D_{\mathrm{JS,thres}}$. This procedure returned two models of roughly the same feature complexity that represent the consistent features of dynamics across data splits. The threshold $D_{\mathrm{JS,thres}}$ sets the tolerance for mismatch between models and choosing higher $D_{\mathrm{JS,thres}}$ results in greater discrepancy between models obtained from two data splits. We set $D_{\mathrm{JS,thres}} = 0.0015$ based on fitting synthetic data with known ground truth; at this threshold value, the selected models reliably matched the ground-truth model.

We split all trials in halves by assigning even trails to $\mathcal{D}_1$ and odd trials to $\mathcal{D}_2$. In the experiment, stimulus conditions were sampled randomly on each trial. Therefore, the time difference between two adjacent trials of the same condition varies broadly, limiting possible temporal correlations between $\mathcal{D}_1$ and $\mathcal{D}_2$.

## Uncertainty quantification

We quantified the estimation uncertainty for fitted models using a bootstrap method[19]. To obtain confidence bounds for the inferred model, we generated ten bootstrap samples by sampling trials randomly with replacement from the set of all trials. To ensure that the two data samples $\mathcal{D}_1$ and $\mathcal{D}_2$ used for model selection do not overlap, we first randomly split all trials into two equal non-overlapping groups, and then sampled trials randomly with replacement from each group to generate $\mathcal{D}_1$ and $\mathcal{D}_2$. For shared optimization, we resampled the trials separately for each stimulus condition. For each bootstrap sample, we refitted the model and performed model selection using our feature consistency method. We then obtained the confidence bounds for the inferred potential, $p_0(x)$ distribution, and tuning functions by computing a pointwise standard deviation across 20 models produced by the model selection on two data splits from each of 10 bootstrap samples.

## Outcomes of model fitting

When fitting our model to spikes of single neurons and populations and performing model selection, we observed three possible outcomes: overfitting, underfitting and good fit.

In rare cases (0 out of 128 single neurons (0%) and 1 out of 15 populations (6.7%) for monkey T; and 1 out of 88 single neurons (1%) and 1 out of 15 populations (6.7%) for monkey O), the model selection produced a model that showed signs of overfitting (Supplementary Fig. 8). We detected overfitting as models with unrealistically high firing rates in the tuning function (hundreds of Hz), disproportionally high noise magnitude (in the range of $D \sim 3$–$5$, compared with $D \sim 0.2$–$0.6$ in regular fits) compensated by deep wells in the potential (overall depth of the potential of approximately 20, compared with approximately 2 in regular fits). These models produced severely underestimated reaction times (reaction time of approximately 10 ms in the model, compared with approximately 500 ms in the data) and did not predict choice of the monkey. This type of overfitting cannot be detected with standard validation approaches[18], for example, these models had similar likelihood on training and validation data.

Some selected models showed signs of underfitting in one of two types. In the first type (10 out of 128 single neurons (7.8%) and 2 out of 15 populations (13%) for monkey T; and 11 out of 88 single neurons (12.5%) and 1 out of 15 populations (6.7%) for monkey O), the potentials had the linear slope tilted towards the same boundary in all stimulus conditions, that is, the model had no decision signal (Supplementary Fig. 9a,b). In the second type (1 out of 128 single neurons (0.8%) and 1 out of 15 populations (6.7%) for monkey T; and 9 out of 88 single neurons (10%) and 0 out of 15 populations (0%) for monkey O), the potentials obtained from two data halves $\mathcal{D}_1$ and $\mathcal{D}_2$ had the linear slope tilted towards the opposite boundaries in at least one stimulus condition (Supplementary Fig. 9c–e). This disagreement about the correct choice side resulted in $D_{\mathrm{JS}}$ values rising high early in the optimization, leading to the selection of a model with low feature complexity before all consistent features had been discovered. These both types of underfitting probably arise when a model cannot detect a weak decision signal and mainly fits the condition-independent trend in neural activity.

All remaining models were considered a good fit and were used in further analyses. In these models, we quantified the potential shape by counting the number of barriers in the potential. A barrier is a potential maximum where the force, which is the negative derivative of the potential $F(x) = -\mathrm{d}\Phi(x)/\mathrm{d}x$, changes the sign from negative to positive. We also classified a potential minimum next to a boundary as a barrier, because the trajectory must get to the top of the potential to reach the boundary. At a potential minimum, the force changes the sign from positive to negative. We therefore counted the number of sign changes from negative to positive and vice versa in the force $F(x)$ in each stimulus condition. We used two force functions $F^1(x)$ and $F^2(x)$ produced by the model selection on two data splits $\mathcal{D}_1$ and $\mathcal{D}_2$ (bootstrap samples were not used in this analysis). We counted a sign change to occur within a local region if both $F^1(x)$ and $F^2(x)$ were negative for ten consecutive grid points to the left and positive for ten consecutive grid points to the right of that region, or vice versa. We only counted sign changes that were at least 30 grid points away from the domain boundaries.

The overwhelming majority of models had a single-barrier potential in all four stimulus conditions (102 out of 117 single neurons (87%) and 9 out of 11 populations (82%) for monkey T; and 66 out of 67 single neurons (98.5%) and 13 out of 13 populations (100%) for monkey O; Figs. 3e,h and 4d, Extended Data Fig. 8 and Supplementary Fig. 6). Some models had a monotonic potential (no barrier) in at least 1 stimulus condition and a single-barrier potential in the remaining conditions (9 out of 117 single neurons (8%) and 1 out of 11 populations (9%) for monkey T; and 0 out of 67 single neurons (0%) and 0 out of 13 populations (0%) for monkey O). The remaining models had a second small barrier in at least 1 stimulus condition and a single-barrier potential in the remaining conditions (6 out of 117 single neurons (5%) and 1 out of 11 populations (9%) for monkey T; and 1 out of 67 single neurons (1.5%) and 0 out of 13 populations (0%) for monkey O). The second barrier was typically shallow and located near the incorrect-choice boundary, where the estimation uncertainty is higher due to lower sampling probability of this region in the data.

We also analysed the potential shape in models that showed the first type of underfitting with no decision signal. These models had feature complexity similar to good fits, suggesting that the model selection identified similar features in the dynamics. The fit, however, captured only the condition-independent dynamics and missed the weak decision signal. These models can still inform us about the mechanism of decision-making. For example, in the two-pool attractor network model[33], inhibitory neurons do not have choice selectivity but they still reflect the attractor dynamics with a barrier separating correct and incorrect choices. Many of the models with no decision signal had a single-barrier potential (5 out of 10 single neurons (50%) and 0 out of 2 populations (0%) for monkey T; and 11 out of 11 single neurons (100%) and 1 out of 1 populations (100%) for monkey O), which further

supports our finding that the dynamics described by a single-barrier potential were prevalent in our PMd data.

When analysing spike-time variation explained by our models on single trials (Fig. 3b), for each neuron, we included only stimulus conditions that had at least 600 spikes across all trials. This restriction was necessary for an accurate estimation of the spike-time variation explained by the model, which was computed on raw spike times without binning or smoothing. For single-neuron models, this restriction produced 111 and 50 single neurons for monkey T and monkey O, respectively (Fig. 3b). For population models, this restriction produced 80 and 32 single neurons for monkey T and monkey O, respectively, which were part of the well-fitted populations. The comparison between the residual spike-time variation unexplained by single-neuron models and the point process variation estimated by the independent method was performed for 64 neurons from monkey T and 27 neurons from monkey O, which had sufficiently high firing rate for the independent method to produce a reliable estimate[32]. For behaviour prediction (Fig. 4e), we additionally only included conditions that had at least five incorrect choices in both training and validation datasets, which did not change the number of analysed populations. This condition was necessary for the baseline comparison, which required training a logistic regression decoder for choice prediction. In this analysis, we used all well-fitted population models and the single-neuron models for the exact same set of neurons that were part of the used populations.

## Spike-time $R^2$

To quantify how well our models fitted spiking activity on single trials, we designed a metric specifically for measuring the fraction of the total variation in spike times on single trials explained by a model. We used the standard coefficient of determination $R^2$ defined as the proportion of the total variation in the data that is predicted by a statistical model:

$$R^2 = 1 - \frac{CV^2_{residual}}{CV^2_{total}}. \tag{12}$$

Here $CV^2_{total}$ is the total variation in the data, and $CV^2_{residual}$ is the residual variation unexplained by the model. As we modelled single-trial dynamics, our metric quantified the variation in spike times on single trials. We quantified the total variation in the data $CV^2_{total}$ using the squared coefficient of variation of ISIs, which is the ratio of the ISI variance to the squared mean ISI[68]. Then, $CV^2_{residual}$ is the residual variation in ISIs unexplained by the model. As our model predicts the firing rate on single trials but not individual spikes, the residual variation is the variation in spike times after accounting for the firing rate variation predicted by the model.

To compute the residual variation in ISIs, we used the time rescaling theorem for doubly stochastic point processes[69]. For a doubly stochastic point process, the total variation in spike times arises from two sources: the variability of the instantaneous firing rate $\lambda(t)$ and the variability of the point process generating spikes from this firing rate. The time rescaling theorem states that we can eliminate the firing rate variation by mapping the spike times from the real time $t$ to the operational time $t'$ via squeezing or stretching the time locally in proportion to the cumulative firing rate: $t' = \Lambda(t) = \int_0^t \lambda(s) ds$. Accordingly, we used a model to predict the instantaneous firing rate $\lambda(t)$ of a neuron on each trial, map spikes to the operational time using this predicted firing rate $\lambda(t)$, and compute the residual ISI variation $CV^2_{residual}$ as the squared coefficient of variation of rescaled ISIs in the operational time. If the firing rate is predicted correctly on each trial, then the variation of rescaled ISIs in the operational time reflects only the point process variability. For example, rescaling spike times generated by an inhomogeneous Poisson process using the ground-truth firing rate yields a homogeneous Poisson process with the firing rate of 1 Hz. The total variation $CV^2_{total}$ was calculated using the raw ISIs in the real time.

To compute the residual spike-time variation $CV^2_{residual}$, we predicted the instantaneous firing rate $\lambda(t)$ with our model on each trial. First, we used the Viterbi algorithm to predict the most probable latent path $\hat{X}(t)$ given the observed data $Y(t)$. We generalized the max-sum Viterbi algorithm with backtracking[70] to our case of continuous-space continuous-time latent dynamical system (Supplementary Methods 2.5). Then, from the most probable latent path $\hat{X}(t)$, we computed the instantaneous firing rate for a neuron $i$ using its tuning function $\lambda(t) = f_i(\hat{X}(t))$. Finally, we rescaled spike times via $t_i' = \int_{t_0}^{t_i} \lambda(t) dt$ (where $t_0$ is the trial start time and $t_1, \cdots, t_N$ are the original spike times) using the trapezoidal rule to approximate the time integral and compute $CV^2_{residual}$ of rescaled ISIs.

The advantage of the $R^2$ metric is that its value is directly interpretable as the fraction of the total variation in the data explained by a model. Specifically, $R^2 = 0$ results from the baseline prediction of a firing rate that is constant within and across trials. Positive $R^2$ values indicate that a prediction is better than this baseline, and negative $R^2$ values indicate that a prediction is worse than the baseline. For a doubly stochastic point process, the value of our $R^2$ metric can never reach 1, because our models predict firing rates on single trials but not individual spikes, hence leaving the point process variability unexplained.

We used the unexplained residual variation in spike times to compare the performance of our model to the ceiling performance expected if the model was correct (Fig. 3b). If the firing rate prediction was correct on each trial, then the residual variation $CV^2_{residual}$ would match exactly the expected point process variation $CV^2_{pp}$ of rescaled ISIs in the operational time. For an inhomogeneous Poisson process, the expected $CV^2_{pp}$ of ISIs in the operational time equals one[69]. However, the spiking of neurons across many brain regions deviates significantly from the Poisson irregularity. In particular, most neurons in the parietal and premotor cortical areas spike more regularly than expected for an inhomogeneous Poisson process[71,72]. Therefore, we used an independent method based on doubly stochastic renewal point processes to estimate the point process variability $CV^2_{pp}$ for each neuron[32]. This method does not require knowledge of the firing rate dynamics on single trials and assumes only that the firing rate evolves smoothly in time. Thus, $CV^2_{pp}$ is the expected residual variation in rescaled ISIs if the firing rate prediction was correct on each trial. The tight correspondence between $CV^2_{residual}$ and $CV^2_{pp}$ indicates that the model accounts for nearly all explainable firing-rate variation in the data close to the ceiling performance.

The independent method for estimating the point process variability based on doubly stochastic renewal point processes has been derived and extensively tested in a separate publication[32]. In brief, for a doubly stochastic renewal point process, the variance $Var(N_T)$ of spike count $N_T$ measured in time bins of size $T$ can be partitioned into two components: the firing rate and the point process variation[32]:

$$Var(N_T) = Var(\lambda T) + \frac{1}{6}(1 - \phi^2) + \phi E[N_T] + \mathcal{O}(T^{-1}). \tag{13}$$

Here $\phi$ is a parameter that controls the point process variability defined as $CV^2$ of ISIs in the operational time, and $\lambda(t)$ is the instantaneous firing rate that is assumed to be approximately constant within a single bin. To estimate $\phi$ from data, we applied equation (13) to spike counts measured in two bin sizes $T$ and $2T$ to yield two equations, which can be solved to obtain a quadratic equation for $\phi$[32]:

$$\frac{1}{2}(\phi^2 - 1) - (4 E[N_T] - E[N_{2T}])\phi + 4 Var(N_T) - Var(N_{2T}) = 0. \tag{14}$$

Here the spike-count mean and variance for each bin size $E[N_T]$, $E[N_{2T}]$, $Var(N_T)$ and $Var(N_{2T})$ are measured directly from the spike data, and $\phi$ is the only unknown variable. Thus, we solved equation (14) to estimate $\phi = CV^2_{pp}$ from data and compared this $\phi$ to the residual spike-time variance unexplained by our model $CV^2_{residual}$ (Fig. 3b).

## Evaluating model performance

We evaluated model performance using multiple metrics: spike-time variance explained ($R^2$), model likelihood relative to several baselines, and behavioural choice prediction. We performed all these analyses within a cross-validation framework using the same two non-overlapping data splits $\mathcal{D}_1$ and $\mathcal{D}_2$ as used for the model selection. For example, we used the model fitted on the dataset $\mathcal{D}_1$ to predict the instantaneous firing rate and compute $R^2$ of each neuron on the dataset $\mathcal{D}_2$, and vice versa. We analysed each stimulus condition separately and averaged $R^2$ across conditions. We report $R^2$ averaged over the two data splits. The same cross-validation framework was used for all model likelihood comparisons and choice prediction analyses.

We compared the likelihoods of single-neuron and population models against each other and several baselines to determine which model components are essential for capturing spiking activity on single trials.

**Single-neuron model versus PSTH.** As a benchmark, we compared the performance of our single-neuron model against the baseline based on the trial-averaged firing rate (PSTH; Fig. 3a). The PSTH predicts the instantaneous firing rate $\lambda(t)$ on each trial using the trial-averaged firing rate traces sorted by the chosen side and stimulus difficulty. We computed the trial-averaged firing rates for the left and right choice trials in a 75-ms window sliding in 10-ms steps on the dataset $\mathcal{D}_1$, and used them as a prediction of the instantaneous firing rates for the left and right choice trials on the dataset $\mathcal{D}_2$, and vice versa. Thus, the baseline PSTH prediction (but not our model) uses the information about the choice of the animal on both the training and the validation trials.

We compared the likelihood of the single-neuron model with the PSTH likelihood (Fig. 3a). The likelihood was computed for the observed spike times $\{t_1, t_2, \ldots, t_N\}$ of a single neuron during the time interval $[t_0; t_E]$ on each trial. The PSTH likelihood for the observed spike sequence is given by the standard likelihood of an inhomogeneous Poisson process[73]:

$$\mathscr{L}_{\text{PSTH}} = \exp\left(-\int_0^{t_E} \lambda(t)\mathrm{d}t\right) \prod_{i=1}^N \lambda(t_i), \tag{15}$$

where we take the instantaneous firing rate $\lambda(t)$ to be the PSTH for each chosen side and stimulus difficulty. We computed the integral over time in equation (15) using the trapezoidal rule.

Unlike PSTH, the full likelihood of our model on each trial (equation (3)) is a joint probability density of the observed spikes $\{t_1, t_2, \ldots, t_N\}$ and a latent trajectory reaching a decision boundary for the first time at the trial end time $t_E$ (the reaction time), marginalized over all possible latent trajectories. Therefore, for comparison with PSTH, we needed to compute the model likelihood $\mathscr{L}_{\text{model}}$ of spike times only, excluding the probability of the reaction time $t_E$. To obtain this likelihood, we normalized the full model likelihood equation (3) by the likelihood $\mathscr{L}_{t_E}$ of the latent trajectory reaching a boundary for the first time at time $t_E$:

$$\mathscr{L}_{\text{model}} = \mathscr{L}[Y(t)|\theta]/\mathscr{L}_{t_E}. \tag{16}$$

This normalized likelihood $\mathscr{L}_{\text{model}}$ is the probability density of the observed spikes only, marginalized over all latent trajectories consistent with the reaction time $t_E$. In other words, $\mathscr{L}_{\text{model}}$ is a probability density that the observed spikes were generated from any latent trajectory that reaches a boundary for the first time at time $t_E$. Thus, $\mathscr{L}_{\text{model}}$ is the likelihood of the observed spikes $\{t_1, t_2, \ldots, t_N\}$ only and is directly comparable with the PSTH likelihood $\mathscr{L}_{\text{PSTH}}$. We computed the likelihood $\mathscr{L}_{t_E}$ analogously to the full model likelihood, but using only $\{t_0, t_E\}$ observations. Specifically, we evolved the latent probability density $p(x_{t_E}|x_{t_0})$ from the trial start $t_0$ to end time $t_E$ by solving the Fokker–Planck equation (equation (6)) with the drift and diffusion terms but without the term $-\sum_{i=1}^M f_i(x)$ for the probability decay due to spike emissions (that is, the standard Fokker–Planck equation).

**Single-neuron model versus shuffled control.** We performed a permutation test to estimate the baseline for the explained variation $R^2$ in spike times on single trials. For all single-neuron models that provided a good fit, we inferred the latent trajectories on validation trials using the Viterbi algorithm and predicted single-trial firing rates of each neuron from these latent trajectories using their tuning functions. We then computed the explained spike-time variation $R^2$ on shuffled trials: we used the firing rate predicted on one trial to compute $R^2$ for spikes on another randomly chosen trial. The explained spike-time variation was much lower for randomly shuffled trials than the original data (shuffled $R^2$, median (Q1–Q3); −0.10 (−0.24 to −0.04), $P < 10^{-10}$, $n = 101$ for monkey T; −0.02 (−0.15 to 0.01), $P < 10^{-10}$, $n = 35$ for monkey O, Wilcoxon signed-rank test). The negative $R^2$ values indicate that the shuffled prediction performed worse than predicting a constant firing rate, which corresponds to $\text{CV}_{\text{total}}^2 = \text{CV}_{\text{residual}}^2$ and $R^2 = 0$. The shuffled prediction was also significantly worse than the prediction based on PSTH. This result confirms that our models explain the spike times on single trials significantly better than the permutation baseline and PSTH, which shows again that our model successfully captured variable firing-rate dynamics on single trials.

**Single-neuron versus population model.** We compared the likelihoods of the single-neuron and population models (Fig. 4a). The single-neuron model predicts spikes of each neuron independently of other neurons in the population. Accordingly, the single-neuron model likelihood of spikes from all $M$ neurons in the population is a product of $M$ single-neuron model likelihoods for each neuron. However, a product of the full likelihood for $M$ single neurons accounts $M$ times for the probability that the latent trajectory reaches a boundary for the first time at time $t_E$, whereas the population-model likelihood accounts for this probability only once. Therefore, we used the normalized model likelihood $\mathscr{L}_{\text{model}}$ (equation (16)) when comparing single-neuron and population models. $\mathscr{L}_{\text{model}}$ is the likelihood of spike data only, marginalized over all latent trajectories consistent with the reaction time $t_E$ on each trial, and, therefore, directly comparable between single-neuron and population models.

**LONO validation versus PSTH.** We compared the likelihood of the population model in leave-one-neuron-out (LONO) validation with the PSTH likelihood based on the neuron's own trial-averaged firing rate (Fig. 4a). For each neuron $k$, the likelihood was computed for the observed spike times $Y_k(t) = \{(t_j, k)\}$ during the time interval $[t_0; t_E]$ on each trial, where index $j$ is restricted to spike times of neuron $k$. We computed the PSTH likelihood for each neuron using equation (15) with the instantaneous firing rate $\lambda(t)$ given by the neuron's own PSTH for each chosen side and stimulus difficulty. We computed the LONO likelihood $\mathscr{L}_{\text{LONO}}$ also using equation (15), but with the instantaneous firing rate $\lambda(t)$ of neuron $k$ predicted by the population model from spikes of all other neurons in the population on each validation trial. Specifically, we used spikes of $M - 1$ neurons in the population excluding spikes of neuron $k$: $Y_{M-1}(t) = \{(t_j, i_j)\}$, where $t_j$ is the time of $j$th spike and $i_j$ is the index of the neuron that emitted this spike ($i_j \neq k$). We applied the Viterbi algorithm to $Y_{M-1}(t)$ to predict the most probable latent trajectory $\hat{X}(Y_{M-1}(t))$ on each trial. We then compute the instantaneous firing rate for neuron $k$ using its tuning function $\hat{\lambda} = f_k(\hat{X}(Y_{M-1}(t)))$, which provides a prediction of the firing rate of neuron $k$ at times $t_j$ when other neurons in the population spiked. We then interpolated $\hat{\lambda}$ with cubic splines to obtain the firing rate prediction $\lambda(t)$ at times when neuron $k$ spiked and substituted it into equation (15) to compute the likelihood, approximating the integral with the trapezoidal rule.

## Predicting choice from neural activity

We used our models to predict the choice of an animal from neural activity. We performed a cross-validation procedure with the same two non-overlapping data splits $\mathcal{D}_1$ and $\mathcal{D}_2$ as used for the model selection. We used the models fitted on the dataset $\mathcal{D}_1$ to predict the choice of the animal on the dataset $\mathcal{D}_2$, and vice versa, and reported the average accuracy over the two data splits. We applied the Viterbi algorithm to neural activity on validation trials to predict the most probable latent path $\hat{X}(t)$. By the design of the Viterbi algorithm with absorbing boundary conditions, the trajectory must terminate at one of the domain boundaries, therefore we predicted choice as the value of $x(t_E)$ at the trial end.

As a baseline for the comparison with our models, we also predicted the choice of the animal with a logistic regression decoder using the same two data splits for the decoder training and validation. As an input to the decoder, we have provided a vector of spike counts measured in 75-ms bins sliding in 10-ms steps on single trials. We truncated each trial at 0.5 s after stimulus onset resulting in a 42-dimensional input vector for single neurons and $42 \times M$ dimensional input vector for populations, where $M$ is the number of neurons in the population. We normalized the inputs to have zero mean and unit variance across trials for each condition in each time bin.

Our data are imbalanced as monkeys make more correct choices than errors, especially on easy trials. We therefore report balanced accuracy for both our models and the linear decoder (Fig. 4e). The balanced accuracy is the average between true-positive and true-negative rates.

The accuracy of choice prediction did not change when replacing the tuning functions and potentials with linear approximations (Extended Data Fig. 9b), showing that the accuracy of choice prediction does not uniquely identify single-trial dynamics.

## Spiking network model

We simulated a spiking recurrent neural network model of decision-making with the same parameters as in ref. 33 using the Python package Brian 2 (ref. 74). We only changed the value of the N-methyl-D-aspartate (NMDA) conductance for inhibitory neurons from $g_{\mathrm{NMDA}} = 0.13$ nS to $g_{\mathrm{NMDA}} = 0.128$ nS to match the reaction times of the spiking network to the experimental data. We simulated four stimulus conditions based on the stimulus difficulty (easy versus hard) and side (left versus right) for comparison with our PMd data. We set the stimulus coherence parameter $c = 17.5\%$ for easy-stimulus and $c = 7.5\%$ for hard-stimulus conditions and generated approximately 3,200 trials of data per condition. The reaction time was defined on each trial as time when one of the population firing rates (smoothed with a moving average over a 200-ms time window) crosses the threshold of 30 Hz. We fitted our population model to the responses of two neurons from each of the two selective excitatory pools (that is, four simultaneous neural responses in total). We performed the same shared optimization across four conditions as for the PMd data using the same hyperparameters for optimization and model selection. To obtain the decision manifold for the network model (Fig. 5e), we plotted the inferred tuning functions of two neurons from excitatory pools against each other, because tuning functions of all neurons from the same pool are identical.

We used the mean-field approximation to reduce the dynamics of the network to a two-dimensional dynamical system model with the same parameters as for the spiking network[34]. The mean-field dynamics are described by two variables $s_i$ ($i = \{1, 2\}$) representing activations of NMDA conductance for two excitatory neural pools:

$$\dot{s}_i = -\frac{s_i}{\tau_s} + (1 - s_i)\gamma f(I_i), \tag{17}$$

where $\gamma = 0.641$ and $\tau_s = 100$ ms. The firing rate $r_i$ of neural pool $i$ is a function of the total synaptic current $I_i$[34]:

$$r = f(I) = \frac{aI - b}{1 - \exp[-d(aI - b)]}, \tag{18}$$

with $a = 270$ Hz nA$^{-1}$, $b = 108$ Hz, $d = 0.154$ s. The synaptic input to neural pool $i$ includes recurrent and external stimulus currents:

$$I_i = J_{11} s_i + J_{12} s_j + I_{\mathrm{stim},i} + I_{\mathrm{bg}}. \tag{19}$$

The effective connection weights are $J_{11} = 0.2609$ nA and $J_{12} = -0.0497$ nA. The background current is $I_{\mathrm{bg}} = 0.3260$ nA. The stimulus current is $I_{\mathrm{stim},\{1,2\}} = 0.0208 \cdot (1 \pm c)$ nA, with the stimulus coherence $c \in [-1, 1]$.

To find the stable and unstable fixed points on the phase plane, we numerically found zeros of the flow field $\vec{R}(s_1, s_2)$ of the two-variable mean-field model in equation (17). We numerically solved the equation $\vec{R}(s_1, s_2) = 0$ starting from a few different initial conditions using MATLAB fsolve function. To find the stable and unstable manifolds of the saddle, we followed the path along $-\vec{R}(s_1, s_2)$ and $\vec{R}(s_1, s_2)$, respectively, starting the trajectory near the saddle point.

## Low-rank network model

To illustrate how diverse tuning to the decision variable arises from the interplay between recurrent connectivity and firing-rate non-linearity, we designed a rank-two recurrent neural network (RNN) that replicates the classical attractor dynamics with distributed connectivity. The network dynamics are governed by the equations:

$$\dot{y} = -y + [Jy + b]_+. \tag{20}$$

Here vector $y \in \mathbb{R}^N$ represents synaptic activation variables $y_i$ of RNN units ($i = 1, \ldots N$), and vector $b \in \mathbb{R}^N$ represents the constant input $b_i$ to each unit. The recurrent connectivity matrix $J \in \mathbb{R}^{N \times N}$ has rank two, such that $J = M \cdot Q^T$ is an outer product of matrices $M \in \mathbb{R}^{N \times 2}$ and $Q^T \in \mathbb{R}^{2 \times N}$. The rectified linear activation function $[\cdot]_+$ models the firing-rate non-linearity of single neurons. The firing rate of neuron $i$ is $r_i = [h_i + b_i]_+$, where $h_i = \sum_{j=1}^N J_{ij} y_j$ is the recurrent input to neuron $i$, and $(h_i + b_i)$ is the total synaptic input current to neuron $i$.

The dynamics of this rank-two RNN are governed by two-dimensional mean-field variables $z = Q^T y$, which follow the equations:

$$\dot{z} = -z + Q^T[Mz + b]_+. \tag{21}$$

We designed the connectivity matrices[39] $M$ and $Q^T$ so that the two-dimensional flow field in equation (21) replicates the flow field of the classical mean-field attractor network in equation (17) for zero stimulus coherence $c = 0$. We chose the elements of $M$ to be $m_{i,1} = \cos(2\pi i/N)$ and $m_{i,2} = \sin(2\pi i/N)$, and sampled the elements of the input vector $b$ randomly from a uniform distribution on $[-0.06$ to $0.06]$. Next, by equating the RNN flow field in equation (21) to the target flow field $\vec{R}(z)$ in equation (17) we obtained

$$Q^T[Mz + b]_+ = \vec{R}(z) + z. \tag{22}$$

This relationship defines a linear regression problem for determining the elements of the matrix $Q^T$. Accordingly, we uniformly sampled $K$ points $z^k$ ($k = 1, \ldots K$) from the state space $[0, 1] \times [0, 1]$ and evaluated the terms in equation (22) at these points to obtain matrix $A \in \mathbb{R}^{N \times K}$ with columns $a_k = [Mz^k + b]_+$ and matrix $B \in \mathbb{R}^{2 \times K}$ with columns $b_k = \vec{R}(z^k) + z^k$. We then found $Q^T$ by solving $Q^T A = B$ using ridge regression with a regularization parameter $\lambda = 0.01$.

We present results for an RNN with $N = 500$ units (Fig. 5 and Extended Data Fig. 10). We simulated the RNN trajectories for the left and right choices by initializing the RNN near the low-activity baseline state with a slight bias towards the corresponding attractor. We chose the initial conditions $z(0) = (0.12, 0.106)$ for the left choice and $z(0) = (0.106, 0.12)$

for the right choice, and computed the corresponding initial conditions for $y$ as $y(0) = (Q^T)^+ z(0)$, where $(Q^T)^+$ is a pseudoinverse of $Q^T$. We parametrized the resulting trajectories with a decision variable $x \in [-1, 1]$, where $-1$ and $1$ correspond to the left and right choice attractors, respectively, and $0$ corresponds to the symmetric initial state. We defined $x$ to increase linearly with the cumulative arc length along the trajectories, such that $x$ grows at a uniform rate along the trajectory length. The tuning curve of each neuron is defined by its firing rate along these trajectories as a function of the decision variable $x$.

## Statistics and reproducibility

The sample sizes used in this study are consistent with standard practices in systems neuroscience involving non-human primates[6,8,12,29]. Data were collected from two macaque monkeys (75 sessions for monkey T and 66 sessions for monkey O), providing a robust number of sessions per animal. No statistical methods were used to predetermine sample size. All key findings were replicated in both monkeys. No randomization or blinding was performed because there was only one experimental group. Investigators were not blinded to the identity of the animal during data collection. Trial types were assigned randomly by the task control software. All statistical tests were two-sided unless otherwise noted.

## Reporting summary

Further information on research design is available in the Nature Portfolio Reporting Summary linked to this article.

## Data availability

The neural recording data are available on Figshare (https://doi.org/10.6084/m9.figshare.29052116.v1 (ref. 59)). The synthetic data used in this study can be reproduced using the source code.

## Code availability

The source code to reproduce the results of this study is available as the NeuralFlow Python package on GitHub (https://github.com/engellab/neuralflow (ref. 75)) and archived on Zenodo[61] (https://doi.org/10.5281/zenodo.15426288).

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

**Acknowledgements** This work was supported by the Swartz Foundation (to M.G.); NIH grant R01 EB026949 (to T.A.E.), RF1 DA055666 (to T.A.E.), S10OD028632-01 (to M.G. and T.A.E.), K99/R00 NS092972 (to C.C.), NS121409 (to C.C.), NS122969 (to C.C.) and R21NS135361 (to C.C.); the Alfred P. Sloan Foundation Research Fellowship (to T.A.E.); the Brain and Behavior Research Foundation (to C.C.); Moorman Simon Interdisciplinary Career Fellowshop (to C.C.); and the Whitehall Foundation (to C.C.). We thank C. Aghamohammadi for sharing the method for estimating point process variability; J. Roach for sharing the code for the analysis of the mean-field network model; and J. Pillow, M. Churchland, Y. Lu, H. Inagaki and L. Duncker for helpful discussions.

**Author contributions** M.G. and T.A.E. designed the research and developed the computational analysis framework. C.C. and K.V.S. designed the experiments. M.G. developed the code, performed the computer simulations and analysed the neural recording data. T.A.E. developed the code and performed the simulations of the low-rank recurrent network. C.C. performed the experiments, spike sorting and data curation. M.G., T.A.E. and C.C. wrote the paper.

**Competing interests** The authors declare no competing interests.

**Additional information**
**Correspondence and requests for materials** should be addressed to Tatiana A. Engel.

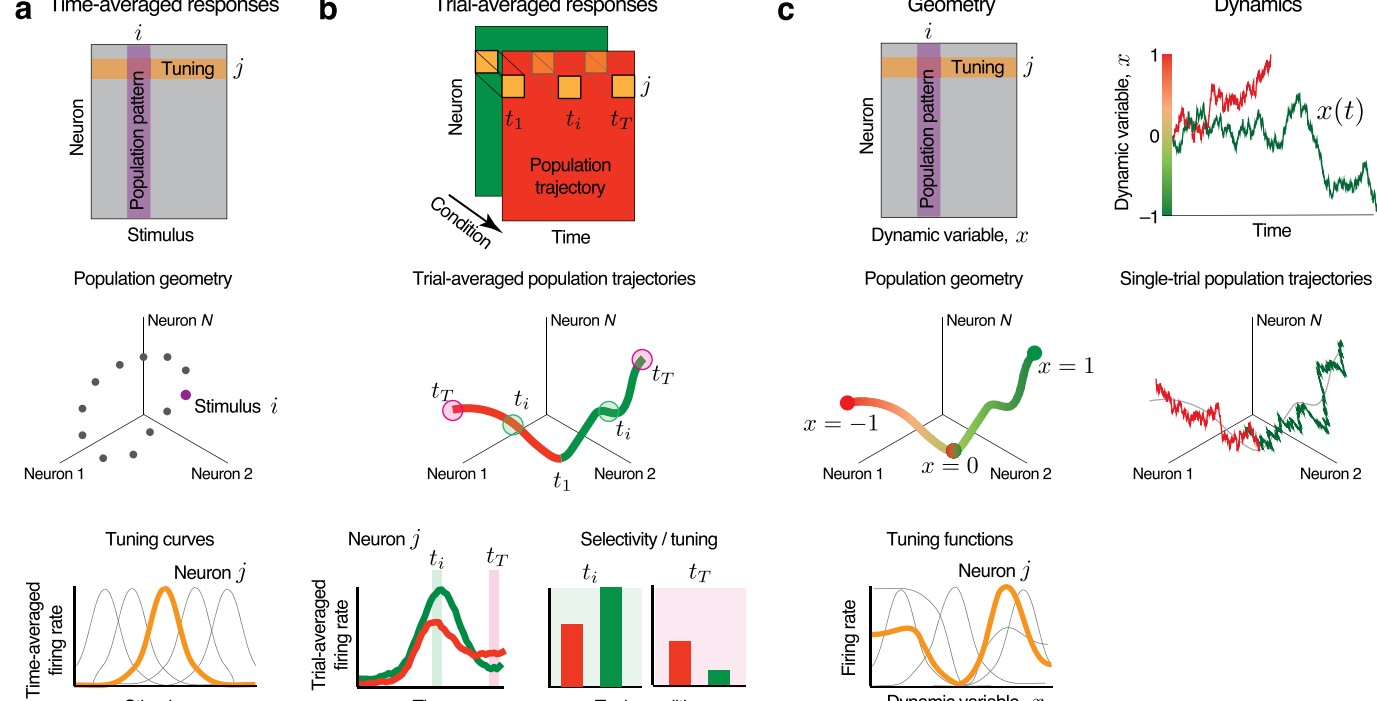

**Extended Data Fig. 1 | The duality between neural tuning and population geometry. a**, Time-averaged responses of $N$ neurons to $K$ stimuli form an $N \times K$ neural response matrix (*top*). The columns of this matrix are points in the neural population state space, where each axis corresponds to the firing rate of one neuron and each point represents a stimulus (*middle*, purple point indicates the $i$th column, which is the population response pattern for stimulus $i$). Population geometry refers to the arrangement of these points in the population state space. The rows of the same neural response matrix define the tuning curves of individual neurons to the stimuli (*lower panel*, orange curve indicates the $j$th row, which is the tuning curve of neuron $j$). **b**, Time-varying neural activity during cognitive tasks is often described by an $N \times T \times K$ tensor containing trial-averaged responses of $N$ neurons at $T$ time points during the trial across $K$ task conditions (*top*). Each $N \times T$ sub-matrix of this tensor is a trial-averaged population trajectory in the corresponding task condition (*middle*). Each row of this sub-matrix is the trial-averaged firing rate of one neuron over time (*lower left*). Traditionally, selectivity (tuning) of a neuron has been characterized as the dependence of its trial-averaged firing rate on the external task variable (task condition) at a specific time in the trial (orange squares in the top panel), as illustrated for neuron $j$ at two different times (*lower right*, $t_i$ mint shading, $t_T$ pink shading). Since single neurons have complex temporal response profiles, their selectivity for external variables often changes over time. For example, a

neuron may show higher firing rate in one condition early in the trial but shift its preference to the other condition toward the trial end (*lower right*). Thus, tuning/selectivity for external task variables lacks temporal consistency and has no dual relationship with neural population trajectories. **c**, Rather than selectivity for external task variables, we consider neural tuning to internally generated dynamic variables that implement cognitive computations. The dynamic variables evolve with distinct time courses on single trials (*top right*). Tuning functions describe how firing rate of a neuron depends on the value of the dynamic variable, independent of time in the trial or task condition (*top left*). These tuning functions uniquely specify the representational geometry of the dynamic variable in the population state space, precisely as they do for sensory stimuli (cf. panel a). For example, the value of the decision variable at any given time is encoded by the position of population activity along a one-dimensional manifold in the population state space (*middle left*). The population trajectories traverse this manifold as they evolve toward one or another choice on each trial. The shape of these population trajectories is uniquely specified by tuning functions of single neurons to the decision variable (*lower panel*), with the exact same dual relationship between neural tuning and population geometry as for sensory stimuli. The dynamics of the decision variable describe how population activity progresses along this manifold on single trials (*middle right*).

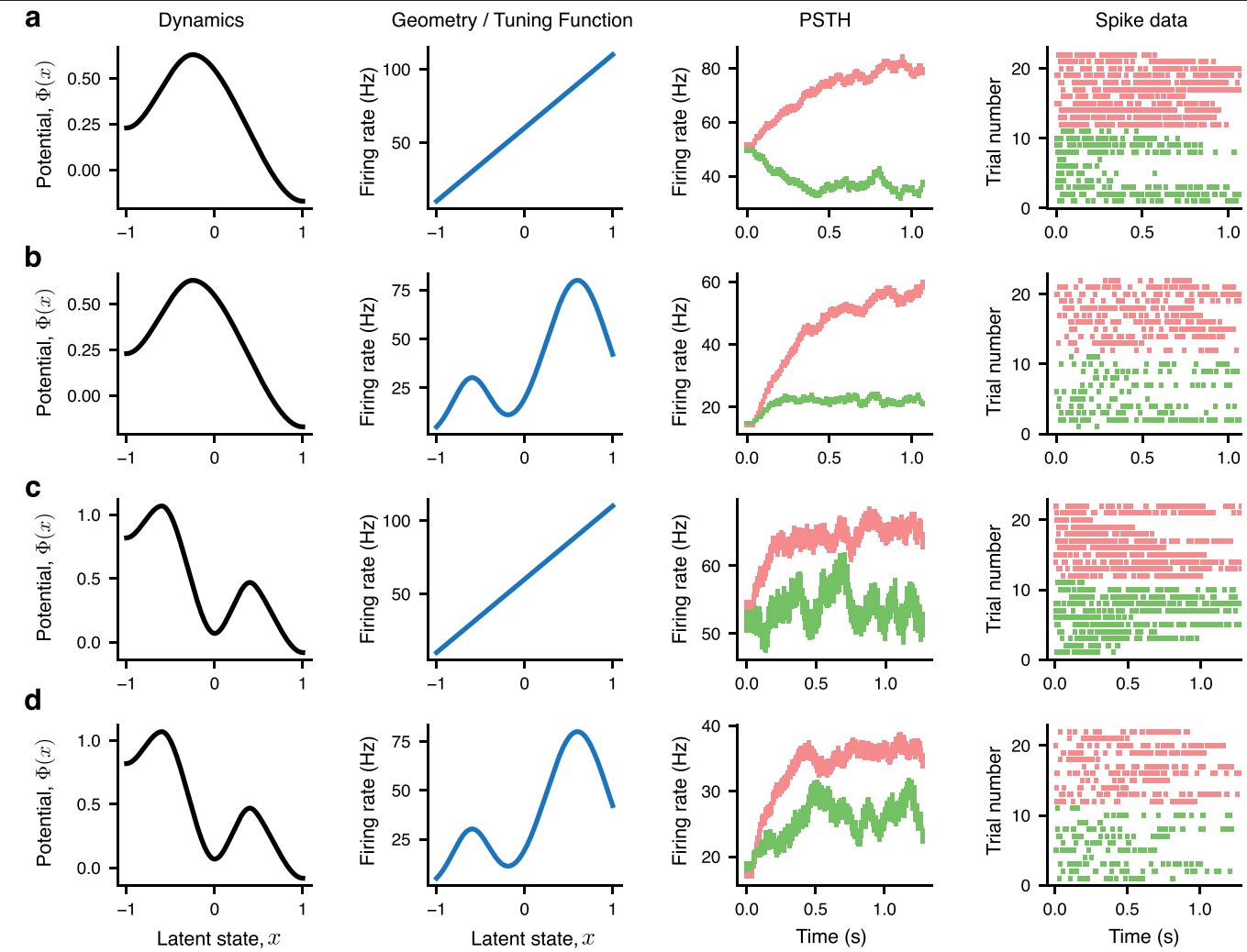

**Extended Data Fig. 2 | Our modeling framework dissociates dynamics and geometry of neural representations on single trials.** Our modeling framework can generate data with identical dynamics but different geometry, or vice versa, thus dissociating the dynamics of the latent variable $x(t)$ from the geometry of its representation in the neural population state space. Synthetic data generated by our model with the latent dynamics defined by: **a,b**, a single-barrier potential; **c,d**, a two-barrier potential (first column). For each latent dynamics, we generate spike data for one example neuron using either linear (**a,c**) or nonlinear non-monotonic (**b,d**) tuning functions (second column), which define the encoding geometry of the latent variable in the firing rate space. These qualitative differences in the dynamics and geometry of neural responses are conflated in the trial-averaged firing-rate traces (third column, PSTH). PSTH is computed over 1,000 trials sorted by choice, which is defined as the boundary to which the latent trajectory converged on each trial. PSTH is shown until the mean reaction time, error bars are s.e.m over trials. The different dynamics and geometry are also difficult to discern in stochastic spike trains (fourth column, spikes are shown for 20 example trials colored by choice). In all simulations, the initial state distribution $p_0(x)$ is a narrow Gaussian centered at $x = 0$. The noise magnitude is $D = 0.2$ (**a,b**) and $D = 0.5$ (**c,d**).

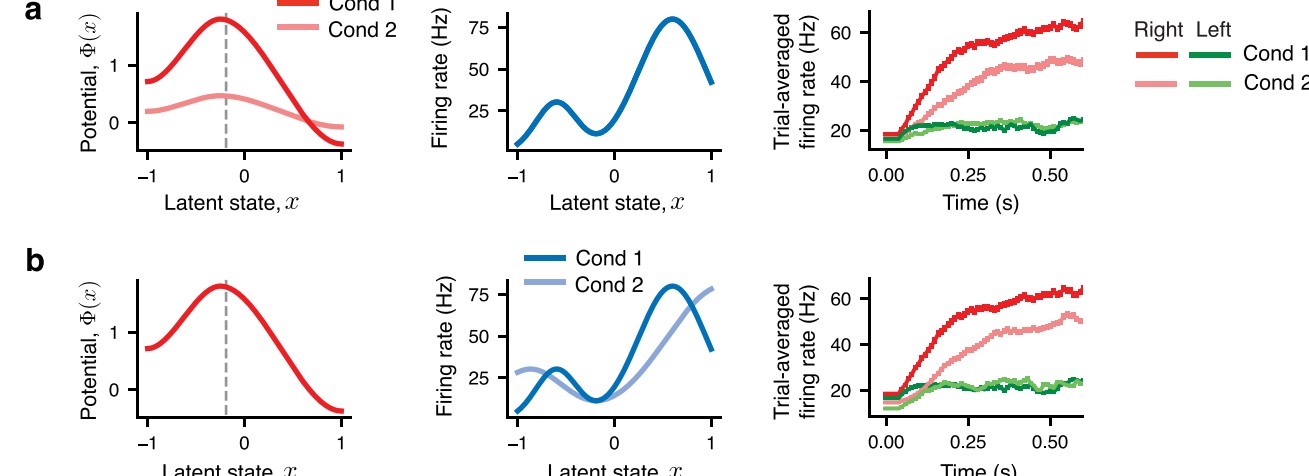

**Extended Data Fig. 3 | Trial-averaged responses conflate dynamics and geometry of neural representations. a**, We generated spikes from our model for one example neuron in two conditions, which had different dynamics (*left*; red – condition 1, steep potential; pink – condition 2, shallow potential) but the same tuning function in both conditions (*center*). For these synthetic data, we computed trial-averaged firing rate (PSTH, *right*) in each condition sorted by choice (defined as the boundary to which the latent trajectory converged on each trial; red – right choice, green – left choice). On the right-choice trials, the trial-averaged firing rates are distinct in two conditions (c.f. red and pink PSTH traces), despite single-trial firing rates evolve along the same path defined by the condition-invariant tuning function. These differences in the PSTH result from different speed of neural dynamics in two conditions. In condition 2, the trajectories $x(t)$ evolve slower towards the right boundary than in condition 1 (due to shallower potential), hence single-trial firing rates increase slower. Accordingly, at a fixed time after the trial start, the firing rate reaches a lower value on average in condition 2 than in condition 1. Thus, differences in dynamics between conditions appear as differences in the trial-averaged firing rate traces. **b**, Very similar trial-averaged responses (*right*) can also arise from the model, in which the dynamics are the same in two conditions (*left*) but the tuning function is different between conditions (*center*; blue – condition 1; light blue – condition 2). In this case, PSTH differences between conditions on the right-choice trials result from different single-trial firing rates defined by the different tuning function in each condition, while the dynamics are condition-invariant. Thus, trial-averaged responses conflate the dynamics and geometry of neural representations, as similar trial-averaged responses can arise from differences in the dynamics (panel a) or tuning functions (panel b) between conditions. The same picture applies to all neurons in the population, which entails that the geometry of the trial-averaged population activity does not uniquely define a model of single-trial neural population dynamics. In all simulations, the initial state distribution $p_0(x)$ is a narrow Gaussian with the center indicated by a dashed grey line. The noise magnitude is $D = 0.27$.

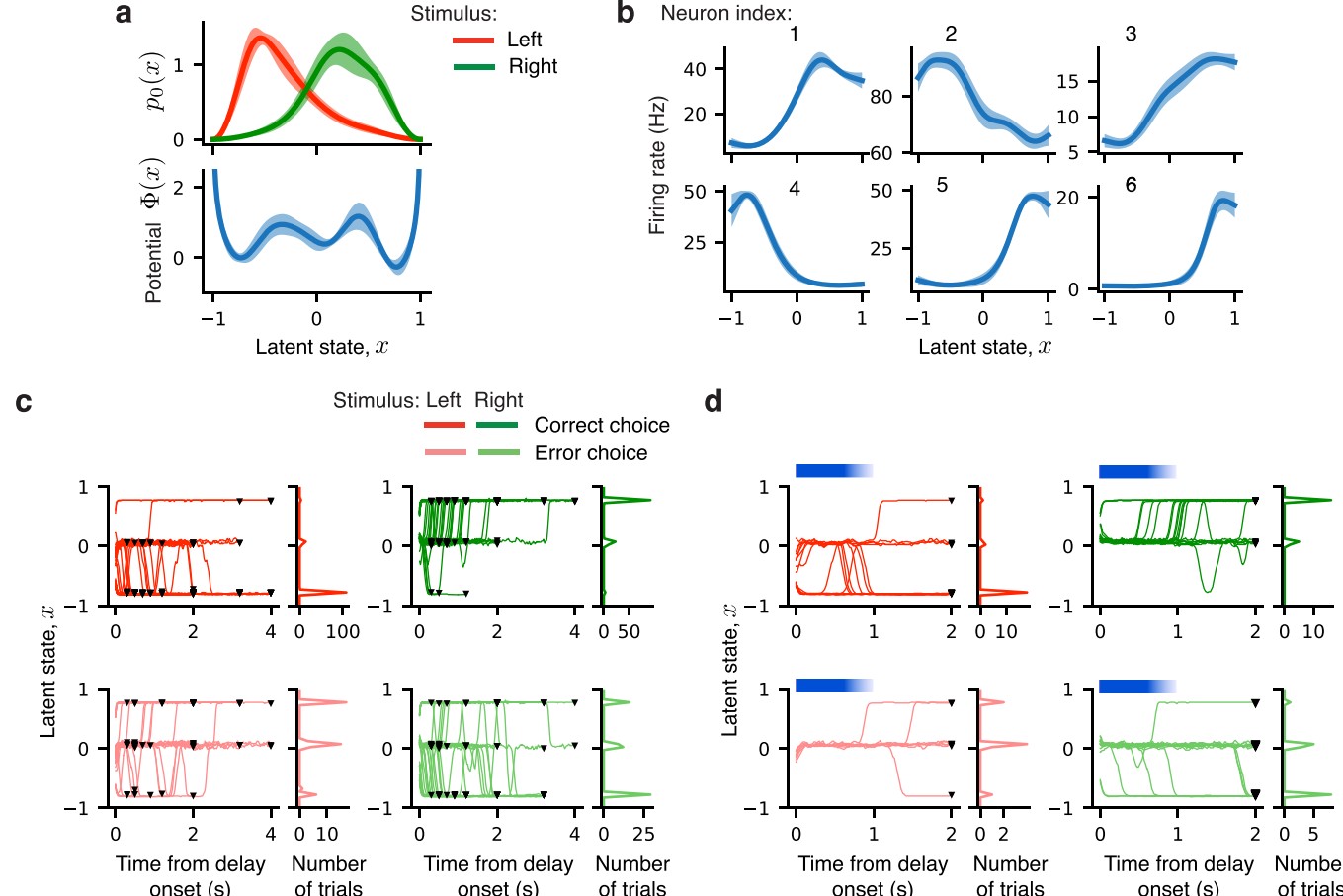

**Extended Data Fig. 4 | Validation of the inference framework in experimental data. a**, Potential governing single-trial population dynamics in the anterior lateral motor cortex (ALM) of mice during the delay period of a delayed-response auditory discrimination task[31] (*lower panel*). The potential, shared between left and right stimulus conditions, was discovered from spikes of 6 simultaneously recorded ALM neurons. The potential shape reveals three discrete attractor wells separated by barriers. The inferred initial state distribution $p_0(x)$ peaks near the left attractor on the left stimulus trials (red) and near the right attractor on the right stimulus trials (green), while also carrying substantial weight near the middle attractor (*upper panel*). **b**, The inferred tuning functions shared across stimulus conditions for 6 neurons from the population in panel **a**.

Error bars in panels **a**,**b** are s.t.d. over 10 bootstrap samples. **c**, Latent trajectories $x(t)$ decoded with the fitted model from the population spiking activity on each correct (upper row) and error trial (lower row) for left (left column) and right stimulus (right column). Black triangles mark the latent state at the end of the delay period on each trial. Histograms to the right of each panel show the distribution of latent states at the end of the delay period across trials. **d**, Same as panel **c** for latent trajectories $x(t)$ decoded from the population spiking activity on photoinhibition trials using the model fitted to unperturbed trials. Bilateral photoinhibition started at the delay period onset and was deployed for 600 ms, followed by a 400 ms ramping down of inhibition (blue strip). See Supplementary Note 1.2 for a detailed description of the analysis.

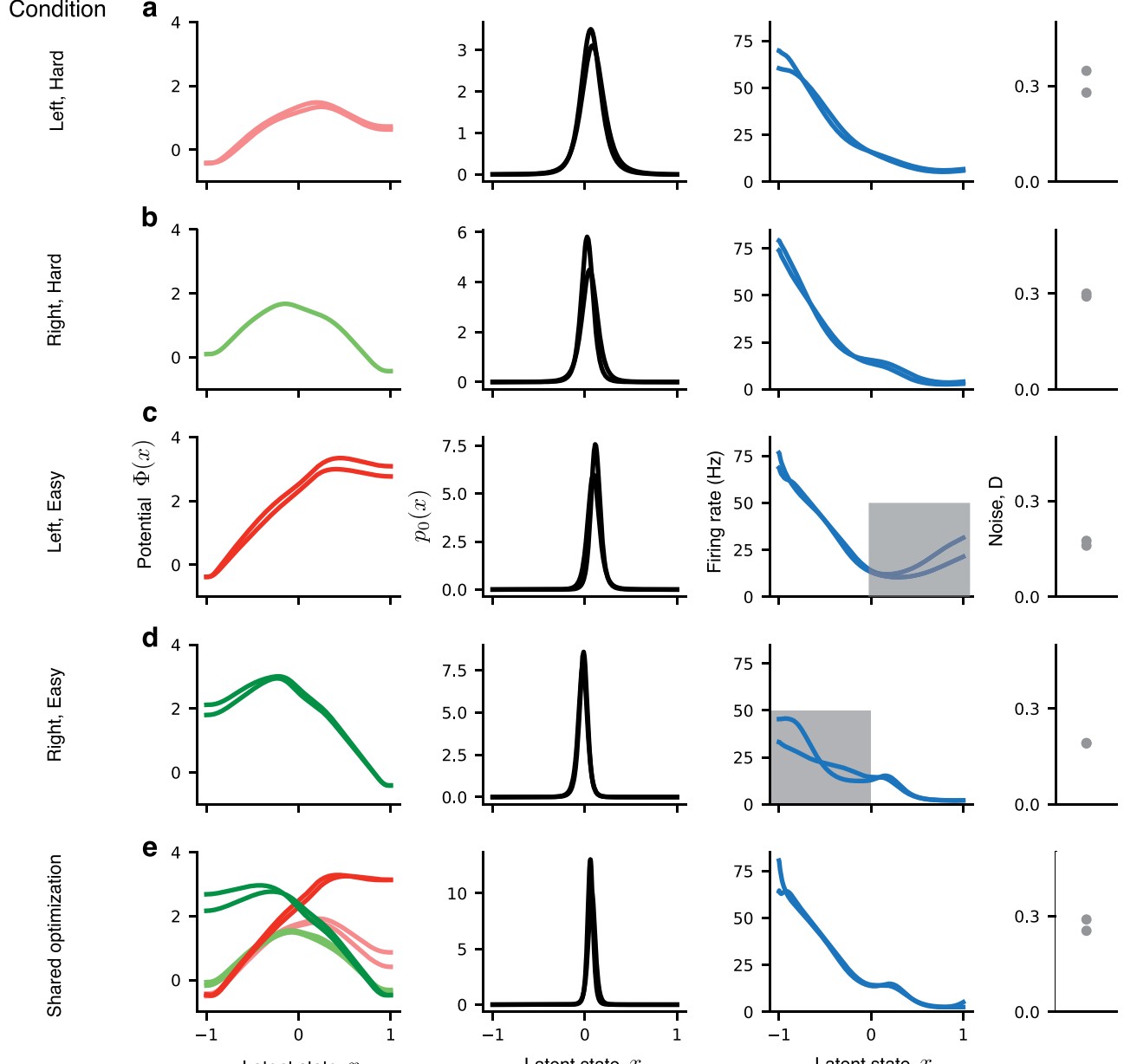

**Extended Data Fig. 5 | Inferred tuning functions were similar across stimulus conditions for an example PMd neuron. a**, Inferred potentials (left), $p_0(x)$ distributions (middle left), tuning functions (middle right) and noise magnitudes (right) obtained on two data halves $\mathcal{D}_1$ and $\mathcal{D}_2$ for the model fitted separately to the left-hard condition trials. The optimal fit was selected using our model selection method based on feature consistency. **b**, Same as panel **a** for the right-hard condition. The slope of the inferred potential points towards the opposite boundary than in the left-hard condition, while the inferred tuning function is largely the same. **c**, Same as panel **a** for the left-easy condition. The inferred tuning function is the same as in the hard conditions in the left part of the domain, where the dynamics evolve towards the correct choice, but the tuning function is inferred less accurately in the right side of the domain (grey highlight) due to very small number of rightward choices

(error trials) in the left-easy stimulus condition. **d**, Same as panel **c** for the right-easy condition. The tuning function is inferred less accurately in the left side of the domain (grey highlight) due to very small number of left choices (error trials) in the right-easy condition. This effect is also observed on synthetic data from the ground-truth model that has the same tuning function in all conditions (Extended Data Fig. 6). **e**, Shared optimization across all four stimulus conditions, in which tuning functions, $p_0(x)$, and noise magnitude are restricted to be the same and only the potential $\Phi(x)$ can vary across stimulus conditions. The shared optimization enables more accurate inference of tuning functions, because it learns a single tuning function across four conditions so that the number of leftward and rightward choices are approximately balanced in the data, and the dynamics equally explore both sides of the decision manifold.

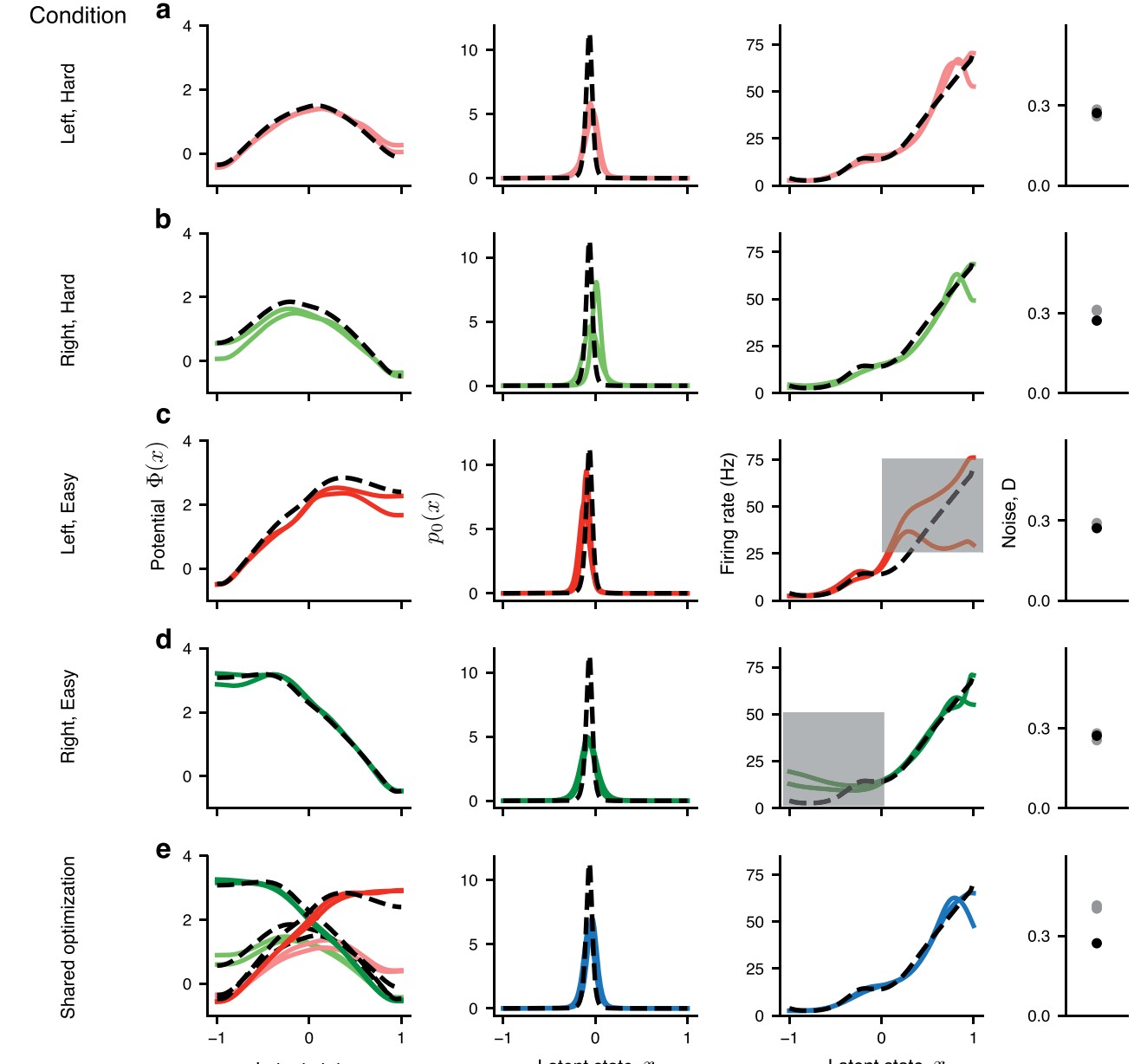

**Extended Data Fig. 6 | Shared optimization enables more accurate inference in synthetic data.** We generated synthetic data from the ground-truth model (shown with black dashed lines in all subplots) in which tuning functions, $p_0(x)$ and $D$ were the same in all conditions. We matched the ground-truth model to the fitted shared model of one experimental neuron and used the same number of trials as in the experimental data. **a**, Inferred potentials (left), $p_0(x)$ distributions (middle left), tuning functions (middle right) and noise magnitudes (right) obtained from the synthetic data on two data halves $\mathcal{D}_1$ and $\mathcal{D}_2$ by fitting the model separately to the left-hard condition trials. **b**, Same as panel **a** for the right-hard condition. The slope of the inferred potential points towards the opposite boundary than in the left-hard condition, while the inferred tuning function is largely the same. **c**, Same as panel **a** for the left-easy condition. The inferred tuning function is the same as in the hard conditions in

the left part of the domain, where the dynamics evolve towards the correct choice, but the tuning function is inferred less accurately in the right side of the domain (grey highlight) due to very small number of rightward choices (error trials) in the left-easy stimulus condition. The same effect is observed in the experimental data (c.f. Extended Data Fig. 5). **d**, Same as panel **c** for the right-easy condition. **e**, Shared optimization across all four stimulus conditions, in which tuning functions, $p_0(x)$, and noise magnitude are restricted to be the same and only the potential $\Phi(x)$ can vary across stimulus conditions. The shared optimization enables more accurate inference of tuning functions, because it learns a single tuning function across four conditions so that the number of leftward and rightward choices are approximately balanced in the data, and the dynamics equally explore both sides of the decision manifold.

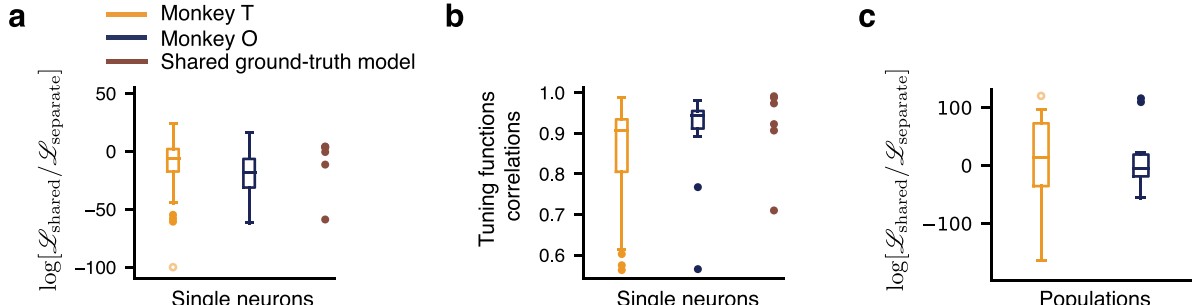

**Extended Data Fig. 7 | Tuning functions were largely consistent across stimulus conditions in PMd data. a**, The log-likelihood ratio of the single-neuron model fitted with shared tuning functions, initial state distribution $p_0(x)$, and noise magnitude $D$ across stimulus conditions relative to the model fitted with separate tuning functions, $p_0(x)$, and $D$ in each condition. This analysis was performed for a subset of single neurons (monkey T: 36 neurons from 3 sessions, monkey O: 16 neurons from 3 sessions). We expect the separate model to have the same or higher likelihood compared to the shared model, because the shared model is a special case of the separate model. The likelihood was only slightly lower for the shared than separate model ($\log(\mathscr{L}_{shared}/\mathscr{L}_{separate})$, median [Q1, Q3]; monkey T: −6.10 [−17.45, 1.96], $n = 36$; monkey O: −18.60 [−31.27, −6.58], $n = 16$), which suggests that the tuning functions were similar across conditions. As a reference, we generated synthetic spike data for 6 single neurons from the ground-truth model in which the tuning functions, $p_0(x)$ and $D$ were the same in all conditions, and then fitted these data with the shared and separate models. We matched the ground-truth model of each synthetic neuron to the fitted shared model of one experimental neuron used in this analysis (4 neurons from monkey T, 2 neurons from monkey O). We also matched the number of fitted trials to the experimental data for each synthetic neuron. The range of the log-likelihood ratio for the shared ground-truth model ($\log(\mathscr{L}_{shared}/\mathscr{L}_{separate})$, median [Q1, Q3]; 1.33 [−8.75, 3.68], $n = 6$) was similar to that obtained from the experimental data, supporting the conclusion that tuning functions were similar across conditions. **b**, To quantify the similarity

of tuning functions inferred by the separate single-neuron model in each condition, we computed their average Pearson correlation coefficient with the tuning function inferred by the shared model for each neuron. High values of the correlation coefficient indicate that the inferred tuning functions were largely consistent across stimulus conditions (median [Q1, Q3]; monkey T: 0.91 [0.81, 0.93], $n = 36$; monkey O: 0.94 [0.91, 0.95], $n = 16$). Comparable values of the correlation coefficient were obtained for the shared ground-truth model (median [Q1, Q3]; 0.95 [0.91, 0.98], $n = 6$). The distribution of the correlation coefficient is shown for the same set of experimental and synthetic neurons as in panel **a. c**, The log-likelihood ratio of the population model fitted with shared tuning functions, $p_0(x)$, and $D$ relative to the population model fitted with separate tuning functions, $p_0(x)$, and $D$ in each condition, for all successfully converged shared population fits (monkey T: 11 populations; monkey O: 14 populations). One outlier with a large positive value 1344.4 is clamped at the value 120 (open orange circle) for better visibility. The log-likelihood ratio was not significantly different between the shared and separate population models ($\log(\mathscr{L}_{shared}/\mathscr{L}_{separate})$, median [Q1, Q3]; monkey T: 13.5 [−35.6, 72.7], $p = 0.58$, $n = 11$; monkey O: −4.6 [−18.8, 18.8], $p = 0.95$, $n = 14$, Wilcoxon signed-rank test), providing strong support for the invariance of tuning functions across conditions. In all box plots, center lines indicate medians; boxes span the 25th to 75th percentiles; whiskers extend to the nearest of 1.5 × the interquartile range or the most extreme data point; outliers beyond the whiskers are shown as dots.

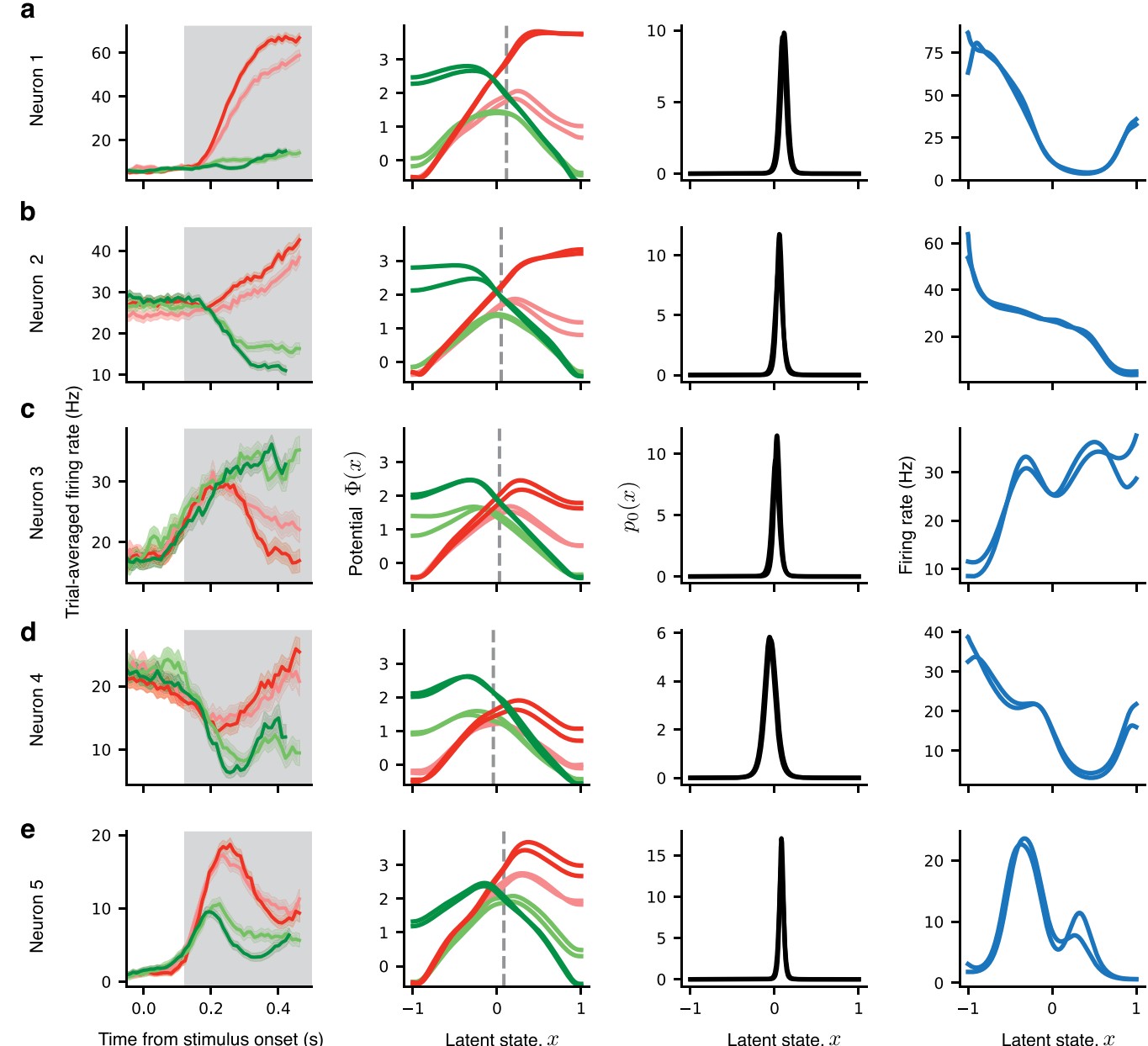

**Extended Data Fig. 8 | The inferred models with a single-barrier potential for additional example single neurons in PMd. a**, Trial-averaged firing rates sorted by the chosen side and stimulus difficulty (left, error bars are s.e.m. over trials), the inferred potentials for four stimulus conditions (middle left), $p_0(x)$ distribution shared across conditions (middle right), and tuning functions shared across conditions (right) inferred on two data halves $\mathcal{D}_1$ and $\mathcal{D}_2$ for a single PMd neuron. Time window used for model fitting starts at 120 ms after the stimulus onset (gray shading) and extends until the reaction time on each trial. **b-e**, Same as panel **a** for four other example neurons. Despite heterogeneous profiles of the trial-averaged firing rates, all models show the same dynamics described by a single-barrier potential and narrow zero-centered $p_0(x)$ distribution. The response heterogeneity results from diverse tuning functions to the latent variable $x$.

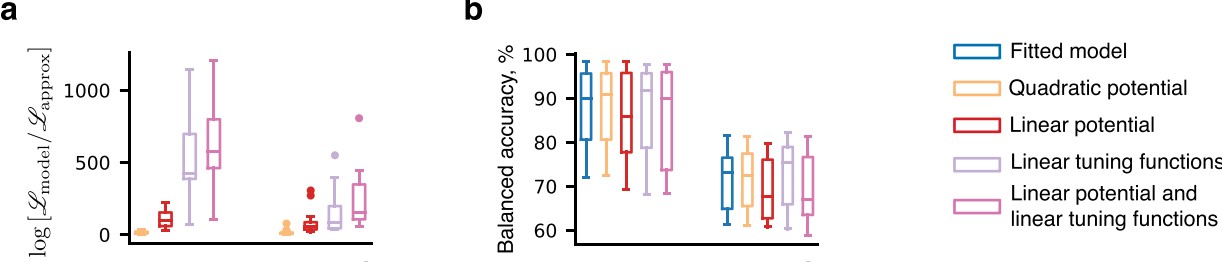

**Extended Data Fig. 9 | PMd responses have nonlinear dynamics and nonlinear geometry. a**, The log-likelihood ratio of the fitted population model relative to the population models in which the potential, tuning functions or both were approximated by a best-fitting analytical function: quadratic approximation of the potential (orange), linear approximation of the potential (red), linear approximation of all tuning functions (light purple), and linear approximation of both the potential and all tuning functions (purple). The log-likelihood ratio is positive for all sessions, indicating that replacing the potential and/or tuning functions with their polynomial approximation significantly reduces model likelihood ($\log(\mathscr{L}_{model}/\mathscr{L}_{approx})$, median [Q1, Q3]; monkey T: quadratic potential 11.7 [7.9, 21.4], $p = 0.002$, linear potential 96.2 [59.2, 153.8], $p = 0.001$, linear tuning functions 426.7 [386.2, 697.6], $p = 0.001$, linear potential and linear tuning functions 573.0 [460.9, 799.2], $p = 0.001$, $n = 11$; monkey O: quadratic potential 9.7 [8.0, 15.0], $p = 2 \cdot 10^{-4}$, linear potential 58.8 [36.1, 86.5], $p = 2 \cdot 10^{-4}$, linear tuning functions 85.3 [44.1, 197.9], $p = 2 \cdot 10^{-4}$, linear potential and linear tuning functions 154.5 [106.3, 348.1], $p = 2 \cdot 10^{-4}$, $n = 13$, Wilcoxon signed-rank test). Approximating the potential with a quadratic function produced only slightly lower likelihood than for the original fitted model, consistent with the observation that the discovered potential shapes were approximately parabolic. **b**, The distribution of balanced accuracy for predicting the monkey's choice using the fitted population model (blue, median [Q1, Q3]; monkey T: 89.9 [80.6, 95.7]; monkey O: 73.2 [64.9, 76.5]), and

population models in which the potential, tuning functions or both were approximated by a best-fitting polynomial: quadratic approximation of the potential (orange, monkey T: 90.9 [80.6, 95.7]; monkey O: 72.5 [65.5, 77.4]), linear approximation of the potential (red, monkey T: 85.9 [77.7, 95.8]; monkey O: 67.7 [62.7, 76.1]), linear approximation of all tuning functions (light purple, monkey T: 91.7 [78.8, 95.7]; monkey O: 75.3 [65.9, 78.9]), and linear approximation of both the potential and all tuning functions (purple, monkey T: 90.0 [73.7, 96.0]; monkey O: 67.0 [63.5, 76.7]). The distributions do not differ significantly between the fitted model and its approximations, except for slightly reduced accuracy in the linear potential and tuning model for monkey T, and in the linear potential model for monkey O (monkey T: quadratic potential $p = 0.09$, linear potential $p = 0.26$, linear tuning functions $p = 0.17$, linear potential and linear tuning functions $p = 0.047$, $n = 11$; monkey O: quadratic potential $p = 0.34$, linear potential $p = 0.03$, linear tuning functions $p = 0.15$, linear potential and linear tuning functions $p = 0.24$, $n = 13$, Wilcoxon signed-rank test). Thus, many qualitatively distinct models of single-trial neural dynamics and geometry can predict the binary choice nearly equally well, showing that the accuracy of choice prediction alone is not sufficient to determine what single-trial dynamics are consistent with neural responses in PMd. In all box plots, center lines indicate medians; boxes span the 25th to 75th percentiles; whiskers extend to the nearest of 1.5 × the interquartile range or the most extreme data point; outliers beyond the whiskers are shown as dots.

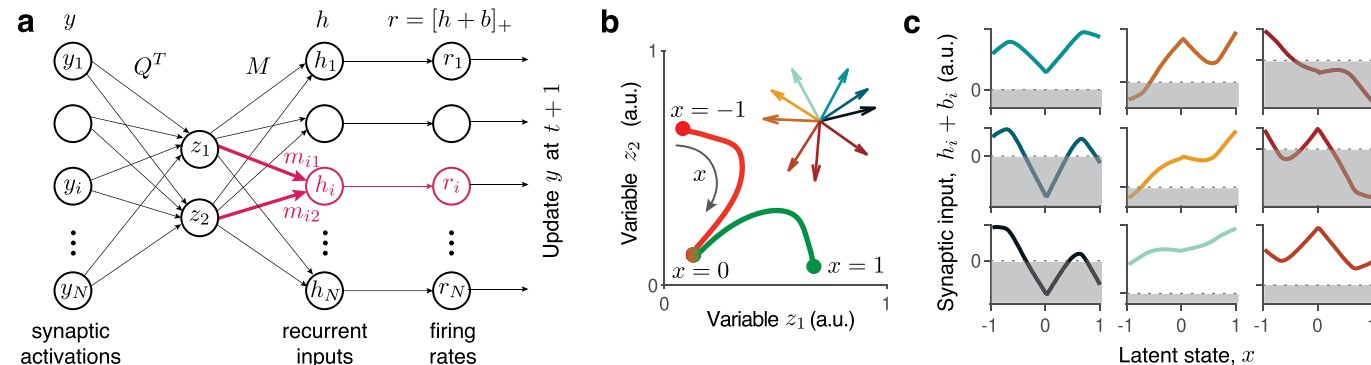

**Extended Data Fig. 10 | Mechanism underlying diverse tuning to the decision variable in rank-two recurrent network. a**, The rank-two recurrent connectivity matrix $J = M \cdot Q^T$ is an outer product of matrices $M \in \mathbb{R}^{N \times 2}$ and $Q^T \in \mathbb{R}^{2 \times N}$, where $N$ is the number of units in the network. On each time step, synaptic activation variables $y$ are updated by filtering through the recurrent connectivity $J$ (equation (20) in Methods). The action of $J$ can be decomposed in two steps: first, $Q^T$ projects $y$ onto two-dimensional mean-field variables $z$, then $M$ embeds $z$ into $N$-dimensional space of recurrent synaptic inputs $h = Jy$ to all units. Accordingly, the dynamics within the space of synaptic input currents $h$ are confined to a two-dimensional linear subspace spanned by the columns of $M$. The dynamics within this subspace are governed by the variables $z$, which follow equation (21) in Methods. The firing rate $r_i$ of each unit $i$ is a rectified linear (ReLU) function of the total synaptic input $h_i + b_i$ this unit receives: $r_i = [h_i + b_i]_+$. The firing-rates $r$ drive the update of the synaptic activation variables $y$ on the next time step. The firing-rate nonlinearity bends the network's trajectories into additional dimensions, leading to higher dimensionality and distinct geometry of trajectories in the firing rate space $r$ compared to the synaptic input space $h$. The synaptic input $h_i$ to unit $i$ is a one-dimensional projection of the two-dimensional variables $z$, defined by the corresponding elements $m_{i1}$ and $m_{i2}$ of the matrix $M$ (red arrows). Thus, firing rate of each unit arises as a one-dimensional projection of the variables $z$, passed through ReLU nonlinearity. **b**, The dynamics in $z$ space replicate the classical mean-field attractor network[34], with trajectories converging to the left (red) or right (green) choice attractors (same network as in Fig. 5g–k in the main text). The one-dimensional decision variable $x \in [-1, 1]$ parametrizes these trajectories, such that $x = -1$ and $x = 1$ correspond to the left and right choice attractors, respectively, and $x = 0$ corresponds to the symmetric initial state at the trial start. Recurrent input $h_i$ to each unit $i$ is a one-dimensional projection of these trajectories onto the direction $\vec{e_i} = (m_{i1}, m_{i2})$. Arrows show the projection vectors $\vec{e_i}$ for the example units in Fig. 5i in the main text. **c**, Firing rate $r_i$ is a threshold-linear function of the total synaptic input current $h_i + b_i$, which is shown for the example units in panel **b** with the corresponding color. The firing-rate nonlinearity rectifies negative inputs (gray shading) to zero. The diversity of projection vectors $\vec{e_i}$, combined with the firing-rate nonlinearity, generates heterogeneous nonlinear tuning functions across units (the corresponding tuning functions are shown in Fig. 5j in the main text).

# Reporting Summary

## Statistics

For all statistical analyses, confirm that the following items are present in the figure legend, table legend, main text, or Methods section.

| n/a | Confirmed | |
|-----|-----------|---|
| ☐ | ☒ | The exact sample size (*n*) for each experimental group/condition, given as a discrete number and unit of measurement |
| ☐ | ☒ | A statement on whether measurements were taken from distinct samples or whether the same sample was measured repeatedly |
| ☐ | ☒ | The statistical test(s) used AND whether they are one- or two-sided<br>*Only common tests should be described solely by name; describe more complex techniques in the Methods section.* |
| ☐ | ☒ | A description of all covariates tested |
| ☒ | ☐ | A description of any assumptions or corrections, such as tests of normality and adjustment for multiple comparisons |
| ☐ | ☒ | A full description of the statistical parameters including central tendency (e.g. means) or other basic estimates (e.g. regression coefficient) AND variation (e.g. standard deviation) or associated estimates of uncertainty (e.g. confidence intervals) |
| ☐ | ☒ | For null hypothesis testing, the test statistic (e.g. *F*, *t*, *r*) with confidence intervals, effect sizes, degrees of freedom and *P* value noted<br>*Give P values as exact values whenever suitable.* |
| ☒ | ☐ | For Bayesian analysis, information on the choice of priors and Markov chain Monte Carlo settings |
| ☒ | ☐ | For hierarchical and complex designs, identification of the appropriate level for tests and full reporting of outcomes |
| ☐ | ☒ | Estimates of effect sizes (e.g. Cohen's *d*, Pearson's *r*), indicating how they were calculated |

*Our web collection on statistics for biologists contains articles on many of the points above.*

## Software and code

Policy information about availability of computer code

| Data collection | Mathworks xPC Target 5.3<br>MATLAB R2012b<br>Psychtoolbox Version 3.0.9<br>Central version 6.0<br>Matclust version 1.7.0.0 (2015)<br>Plexon Offline Sorter v3 |
|---|---|
| Data analysis | Data analysis was performed using custom Python 3 code. The source code to reproduce the analyses is available as the NeuralFlow Python package on GitHub (https://github.com/engellab/neuralflow) and archived on Zenodo (https://doi.org/10.5281/zenodo.15426288). |

For manuscripts utilizing custom algorithms or software that are central to the research but not yet described in published literature, software must be made available to editors and reviewers. We strongly encourage code deposition in a community repository (e.g. GitHub). See the Nature Portfolio guidelines for submitting code & software for further information.

# Data

Policy information about availability of data

All manuscripts must include a data availability statement. This statement should provide the following information, where applicable:
- Accession codes, unique identifiers, or web links for publicly available datasets
- A description of any restrictions on data availability
- For clinical datasets or third party data, please ensure that the statement adheres to our policy

Neural recording data are available on Figshare at https://doi.org/10.6084/m9.figshare.29052116.v1. The synthetic data used in this study can be reproduced using the source code.

# Research involving human participants, their data, or biological material

Policy information about studies with human participants or human data. See also policy information about sex, gender (identity/presentation), and sexual orientation and race, ethnicity and racism.

| | |
|---|---|
| Reporting on sex and gender | NA |
| Reporting on race, ethnicity, or other socially relevant groupings | NA |
| Population characteristics | NA |
| Recruitment | NA |
| Ethics oversight | NA |

Note that full information on the approval of the study protocol must also be provided in the manuscript.

# Field-specific reporting

Please select the one below that is the best fit for your research. If you are not sure, read the appropriate sections before making your selection.

☒ Life sciences        ☐ Behavioural & social sciences        ☐ Ecological, evolutionary & environmental sciences

For a reference copy of the document with all sections, see nature.com/documents/nr-reporting-summary-flat.pdf

# Life sciences study design

All studies must disclose on these points even when the disclosure is negative.

| | |
|---|---|
| Sample size | The sample sizes used in this study are consistent with standard practices in systems neuroscience involving non-human primates, e.g., Mante et al. Nature 503, 78–84 (2013); Engel et al. Science 354,1140–1144 (2016); Peixoto et al. Nature 80, 791–21 (2021); Steinemann et al. eLife 12, RP90859 (2024). Data were collected from two macaque monkeys (monkey T: 75 sessions; monkey O: 66 sessions), providing a robust number of sessions per animal. No statistical methods were used to predetermine sample size. |
| Data exclusions | No data were excluded in this study. |
| Replication | All key findings were replicated in both monkeys. |
| Randomization | Participants were not allocated into groups. Subject randomization is not relevant to this study. Both animals performed the same task under the same conditions, over many sessions and trials. All trials were randomized as were placements of electrodes on a day to day basis. |
| Blinding | Blinding is not relevant to this study as the macaques were not placed into different groups, they performed the same task under the same conditions from session to session. |

# Reporting for specific materials, systems and methods

We require information from authors about some types of materials, experimental systems and methods used in many studies. Here, indicate whether each material, system or method listed is relevant to your study. If you are not sure if a list item applies to your research, read the appropriate section before selecting a response.

## Materials & experimental systems

| n/a | Involved in the study |
|-----|----------------------|
| ☒ | ☐ Antibodies |
| ☒ | ☐ Eukaryotic cell lines |
| ☒ | ☐ Palaeontology and archaeology |
| ☐ | ☒ Animals and other organisms |
| ☒ | ☐ Clinical data |
| ☒ | ☐ Dual use research of concern |
| ☒ | ☐ Plants |

## Methods

| n/a | Involved in the study |
|-----|----------------------|
| ☒ | ☐ ChIP-seq |
| ☒ | ☐ Flow cytometry |
| ☒ | ☐ MRI-based neuroimaging |

## Animals and other research organisms

Policy information about studies involving animals; ARRIVE guidelines recommended for reporting animal research, and Sex and Gender in Research

| | |
|---|---|
| Laboratory animals | 2 male macaque mulatta. T – 7 years old, O – 11 years old. |
| Wild animals | The study did not involve wild animals. |
| Reporting on sex | Both macaques were males. Female macaque monkeys are difficult to obtain because of their use for breeding. The small number of animals used in primate electrophysiology studies precludes any statements about sex differences. |
| Field-collected samples | The study did not involve samples collected from the field. |
| Ethics oversight | Stanford University Institutional Animal Care and Use Committee approved the study protocol (8856). |

Note that full information on the approval of the study protocol must also be provided in the manuscript.

## Plants

| | |
|---|---|
| Seed stocks | NA |
| Novel plant genotypes | NA |
| Authentication | NA |

