## [Peer Review File · Nature]

The dynamics and geometry of choice in premotor cortex

Corresponding Author: Professor Tatiana Engel

Version 0:

Reviewer comments:

Referee #2

(Remarks to the Author)

Genkin et al. propose a framework in which neural population dynamics are determined by latent attractor dynamics, with the neural firing rates being determined by tuning curves that are functions of the dynamics. They apply their model to premotor cortical data during decision making in monkeys to show that their results return attractor dynamics consistent with previous theories of decision making in evidence accumulation.

The paper is clearly written for the most part and the analysis is quite nice. I definitely think it would be of high interest to systems neuroscientists interested in population coding. However, as you will see below, my main concern regards the novelty of the results which makes me hesitant to fully recommend, despite my appreciation of the solid modelling framework and strong ties to classic theories of decision making.

Major concerns.

1. While the mathematical modelling is very cool, I am left with the uneasy feeling that I am not sure what I have learned from this analysis that is new. Many classic papers have already studied evidence accumulation in premotor cortex, including the classic computational papers the authors cited for Fig. 5 and classic drift diffusion models. More recently, Mante et al. already showed attractor dynamics governing decision making in a more complex task. This approach is clearly distinct, but can we learn anything new with this model? I feel that there is a potential for disassociating dynamics and geometry through it (hinted at in the title) but this is never quite shown. Is there a way to demonstrate new dynamics that you wouldn't have been able to guess from the manifold geometry itself, in the way that Mante did? Maybe separately fitting the error trials could show that these are due to some changes in the dynamics that previously we would have assumed was due to noise, without access to a model that explicitly fits latent dynamics?

2. The authors should help to solidify the mechanistic interpretation of the framework. The idea of tuning curves in the sensory system is a phenomenological model based on the idea that sensory neurons are extracting latent features (eg orientation of a bar) from sensory input (pixels). But in the context of cognition, it seems unclear to me why there would be a nonlinear mapping between the evidence accumulation variable, which is already encoded by PMd neurons, and the firing rates of individual neurons. Going back to Mante et al., there the authors fit an RNN to find the attractor dynamics which has clear mechanistic interpretation (if lacking on some biological realism). Can the authors offer (perhaps in the discussion) a way to interpret their model mechanistically and the relationship to other population dynamics models that have latent task representations at their core?

3. I'm surprised that there isn't more discussion about other work in latent neural dynamics, including work from Maneesh Sahani, Srdjan Ostojic, Scott Linderman, John Cunningham ... and surely others, this is just an example. The authors should include more discussion about the relationship between their model and its assumptions to other formulations of latent dynamics which have become increasingly popular in recent years. As a start, how about discussing the relationship between the proposed model and latent linear dynamical systems? How important are the differences between this model and LDS? (nonlinear dynamics, nonlinear emission, Poisson noise from what I see)

4. On this note, I was underwhelmed by the comparison in Fig. 4e to a logistic decoder, as the decoder had completely different assumptions from the authors' model (e.g., being smoothed, fed in trial averaged activity etc). It would be more illuminating to strip down their model step by step to see what assumptions are actually important to predict performance. For example, replacing one or both of $d\Phi/dx$ and f by linear terms would help to determine what is actually important for

decision dynamics (it looks like the potential wells can be well captured by quadratics in which case $d\Phi/dx$ would be linear). This is a specific example but it's the only place I saw an actual benchmarking plot to another method (other than PSTH which is too basic).

5. In some places the statistical analysis could be improved or clarified.

- i. For example, permutation test to get an idea of what baseline is for the variance explained or for accuracy plots. When splitting the data into D1 and D2 for cross validation – how are these defined? Depending on the choice you might have temporal correlations between your datasets.
- ii. L153-159 I feel uncomfortable interpreting this analysis since it's only based on a single animal and the exclusion criterion for Monkey O is unclear. Was 88 neurons really not enough for a leave-one-out analysis?
- iii. A lot of the results regarding the potential well shapes are qualitative and should have more quantitative descriptions of how this was assessed
- iv. Overall I

6. The authors' mathematical framework is extensive and has been developed over multiple papers over the past few years. It is difficult to assess how much of that framework is new for this paper, or whether they simply applied an already-existing framework to a new dataset. Some clarification here would help, both in the main text, but also in the supplementary methods where they should point out which details (if any) have been developed especially for this study.

Minor comments

- Typo on line 57, should be x_0 or $x(0)$
- The authors should mention somewhere that equation 1 only describes fixed point attractor dynamics (no limit cycles)
- The authors should mention somewhere that Φ is condition dependent
- In equation 1, why is the noise amplitude also scaling the deterministic term? This means the gradient down the well is stronger when there is higher-variance noise. Could this lead to any issues comparing Φ across plots, if this scaling is not incorporated?
- L337 onwards, I personally find the time indexing odd as e.g. $Y(t)$ does not depend on t . But I leave it for the authors to decide how to clarify this.
- Equation 4, some more explanation of A would be helpful even at a high level
- How is the number of eigenvectors chosen? I didn't see this explained anywhere. Is it the same as the discretization N ? If not, how was this number chosen?
- The authors could further discuss Finkelstein et al., 2019, Nature Neuro, which is related thematically although a different region

Referee #3

(Remarks to the Author)

I have a love-hate relationship with this paper. I love the result, but I hate the fact that it was so hard for me to extract it. I partially blame the field, for encouraging words like "representation" and "geometry", both of which should be banned from all papers. OK, geometry can be used, but only if it's preceded by "Euclidean" or "non-Euclidean", but not when referring to neural activity. And representation never.

So let me start with what the authors did, which was pretty simple: they mapped sensory input to neural activity. The sensory input was a checkerboard stimulus composed of red and green squares; these were summarized by their difficulty, so there were effectively 4 stimuli: easy left, easy right, hard left and hard right. (Easy and hard refer to the difficulty of telling whether there were more green than red squares; left and right refer to the button the monkeys pressed.) The neural activity was in primate dorsal premotor cortex (PMd). So basically they built a map: $r_i(t) = g_i(s, t)$ where $r_i(t)$ is the firing rate of neuron i , s is stimulus, t is time, and $g_i(s, t)$ is some function.

That sounds a bit mundane, but what was interesting was the function, $g_i(s, t)$. It consisted of two parts: one was a "decision" variable, x , which evolved according to a stochastic differential equation that depended on the stimulus; the other was the spike generation process, where neuron i emitted spikes according to a Poisson process with rate $f_i(x)$, and did not depend (directly) on the stimulus. So the model looked like

$$\begin{aligned} dx/dt &= h(s, x) \\ r_i(t) &= f_i(x) \end{aligned}$$

where x is a 1-D variable (and $h(s, x)$ contains white noise, so x obeys a stochastic differential equation, but that's a detail).

And it gets better. First, the monkey made a decision when x reached a boundary (left when $x=+1$; right when $x=-1$). So x really can be interpreted as a decision variable. Second, and, I think, even more important, the "tuning curves", $f_i(x)$, were independent of the stimulus.

So this is very cool, and somewhat unexpected. We're used to low-dimensional trajectories. But here the activity follows the same one-dimensional trajectory on every trial. (OK, not exactly the same; there's noise, because x obeys a stochastic differential equation. But we never expected neurons to be noise-free.) The only thing that varies from trial to trial is the speed and direction along the 1-D trajectory.

Why do we care? I think it's because it has implications for what's going on in PMd. Let's write a very general equation for the membrane potential on neuron i in PMd,

$$dV_i/dt = f(\text{recurrent spikes}, \text{external spikes}).$$

Here "recurrent spikes" refers to spikes from within PMd and "external spikes" refers to spikes from other areas. There are two main ways this equation could be consistent with their results:

1. The external spikes are more or less constant in time, and they push the activity along a 1-D manifold.
2. The external spikes themselves evolve along a 1-D manifold (that corresponds to x), and the activity in PMd is driven to a fixed point that depends on the external spikes.

(1) seems a lot more likely to me, but (2) is also possible, and it could even be a combination. (And maybe something I haven't seen thought of?) But it seems to make strong, experimentally testable predictions about what PMd is doing, at least in this task.

So that's what I loved about the paper. Now what I didn't like.

First, I wasn't entirely convinced of their results. Meaning I wasn't entirely convinced of the above model ($dx/dt=h(s, x)$; $r_i(t)=f_i(x)$), for a number of reasons.

1. Supposedly Fig. 3a shows that their model explained the variance better than the PSTH. But that figure didn't make a lot of sense to me. The variance explained, which came (I believe) from Eq. 10, depends on the CV of the interspike interval distribution, which measures how much the true firing rate changes over time. Which has nothing to do with model fit. Or did I misinterpret?

In any case, the authors should do something simpler, and more standard. For instance, they could compare the log likelihood of the model versus a firing rate computed from the PSTH, or from a GLM.

That said, I personally don't care that much if their model fits better than the PSTH or a GLM; that doesn't seem to be the point. What I really want to know is whether it's a reasonable fit. For that maybe they could just report log likelihood of their model relative to the expected log likelihood if the model were correct? Or drop this altogether? The next couple points are, I think, more important.

2. Lines 75-6: "We noticed that the inferred tuning functions, initial state distribution $p_0(x)$ and noise magnitude D were similar across four stimulus conditions and only the potential shapes were different (Supplementary Fig. 2)."

"Similar" is a squishy word; the authors need to quantify how similar. Especially since I looked at Supplementary Fig. 2, and there are clear differences (especially where things are undersampled). I can suggest a simple test: run the model with all the tuning functions the same and run them with different tuning curves for each stimulus, and compare the log likelihoods under the two conditions. If they're not too far off, declare victory.

3. Two sets of simulations were run. In the first set, each neuron could have a different trajectory; that is, for neuron i there was a trajectory $x_i(t)$. In the second set (starting on line 122), all the neurons had a single trajectory (there was just $x(t)$). What seems most important for their model is to show that when using only one $x(t)$, the fits were still good. But again, they used the variance explained using Eq. 10, which I didn't understand at all. (And again, possibly because I misinterpreted it.) But here, why not do something simple: fit the model assuming individual trajectories $x_i(t)$, then fit it using a single trajectory $x(t)$, and compare the log likelihoods; in both case with the tuning curves, $f_i(x)$, independent of the stimulus.

I personally really hope that when they use a single latent trajectory and tuning curves that don't depend on stimulus, they do a good job predicting spike trains. Note that it doesn't have to be perfect, just good. But it's a strong claim, and it needs strong evidence.

The second reason I didn't like this paper is that it took me forever to extract the above picture. (Which hopefully is correct!) I think this was because the authors never spelled out what they were doing in a way I could understand. Consider lines 10-12,

"We hypothesized that neural encoding of dynamic cognitive variables follows the same principle as sensory variables: the encoded variable determines the topology of neural representation, and heterogeneous tuning curves of single neurons define the representation geometry."

I simply could not parse this (see first sentence of my review). I now know what it means, but only because I read the paper. About five times. So I had to absorb every fact in case it was important, and piece together a story on my own. If I hadn't been a reviewer, I would never have done this. Given that Nature papers are supposed to be for broad readership, it seems that one should avoid writing for a narrow audience.

Why not just say what was found, in plain language, which was: neurons are tuned to a single latent variable that evolves

under simple and intuitive dynamics? And point out why this is an important, and unexpected, finding.

And now I have lots of small, sometimes technical comments. As an author, I find that replies to reviewers are often longer than the paper, so feel free not to reply directly to these. Maybe reply only to the ones where you thought my suggestion was really bad?

1. Please put the figures in main text where they belong. I almost never review papers with the figures at the end; this one slipped by.
2. When you say "see Methods", please say where. It's a bit hard to have to look at all of Methods to find something.
3. In Figs. 3d and g, Φ is almost identical. Is this typical? If so, you should comment on that, as it's pretty amazing. If not, it would be good to show example neurons where the Φ 's are obviously different.
4. I 201-4, "The dynamics we found in PMd are qualitatively distinct from the stepping and ramping models proposed previously [11,21,24,25 and, instead, are consistent with the attractor mechanism hypothesized in neural circuit models [30-35]."

First, I don't understand this. Under your model, firing rates of neurons change continuously in time, so it rules out the stepping model. But why does it rule out the ramping model? Is it because the firing rates aren't monotonic in time? You should expand on this. (Although I'm not sure why it's included; it seems beside the point. Maybe it's just to irritate Mike Shadlen and Jonathan Pillow? ;))

Second, it's not obvious why your model is an attractor network. Given that you have absorbing boundaries, it would seem hard to even tell if it's an attractor network.

5. Including the spiking network model doesn't seem to bolster your case, since for that model the latent variable is 2-D. And it doesn't really add anything, in my opinion. If I were you I would take it out. But that's totally optional.

6. I 230: what does representation geometry mean here?

7. I 254-6: "our results suggest that the dynamics of internal cognitive computations may differ across brain areas and thus cannot be described by the same global variable inferred from behavior."

I can't parse this sentence. I think the problem is the word "same". Same as what?

9. Under the change of variable $t = s/D$, Eq. 1 becomes

$$dx/ds = -d\Phi/dx + \sqrt{2} \xi(s)$$

So D has no effect on the dynamics, other than to scale time and Φ . It doesn't matter for anything you did. But I'm curious why you put a D in front of $d\Phi/dx$. It seems non-standard.

10. paragraph starting on I 337: it's a little confusing to drop the subscript k : the first and second $Y(t)$ look the same but mean different things. Maybe call the first one \mathcal{Y} ?

And this is even more confusing in the next paragraph, which contains $Y(t)$. Which $Y(t)$ is it? Presumably a single trial. If so, is that the single trial likelihood?

11. I 352 and Eq. 3 (and other places): should it be $P(X(t), Y(t)|\theta)$ rather than $P(X(t), Y(t))$?

12. Bins with no spikes seem to be missing from Eq. 4. I was expecting factors of $(1 - f_{i_j})(x_{t_j} dt)$ whenever neuron i_j didn't fire. Or am I misinterpreting things?

13. Eq. 5: I'm assuming M is the number of neurons. If you didn't say this somewhere, you should.

14. In Eq. 5, the probability decays. You say on line 371 that that's "due to spikes emitted by any neuron in the population". But why should spike cause decay? I thought the only decay was due to absorption on the boundary.

15. Speaking of absorption, when x hits a boundary, is the trial over? Or does the trial keep going, with firing rates $f_{j(+1)}$? That seems important for inference, because if the trial ends when x hits a boundary, then on each trial you have to choose a path that ends when that trial is over. Techniques exist to do that, but (after glancing at SI), I don't think you do it. Please clarify.

16. What does $\hat{H}(t_j - t_{j-1})$ mean?

17. I 383: why that particular choice for C ? And is the integral from -1 to 1?

18. It would be very helpful if all important equations were displayed, not inline. In particular, (but not limited to!) the two

expressions on line 382, which should definitely be displayed. Then you could refer back to them when you write down Eqs. 7 and 8 (which took me a long time to figure out). Most people (meaning me) do not remember definitions.

Version 1:

Reviewer comments:

Referee #2

(Remarks to the Author)

I understand better the novelty of the manuscript, but I still have the same general hesitations. I hate to sound critical because I did like the paper and the modelling framework. I strongly agree that it is important to infer latent neural dynamics from data. It convincingly show that neural population activity and single neuron activity encodes some kind of decision variable that is consistent with attractor dynamics. What it's missing are (in order of decreasing importance): 1) causal perturbation tests to establish the attractor dynamics are correct, 2) a mechanistic interpretation of the model, and 3) a real link between dynamics and geometry as suggested in the title. I would appreciate if the authors would better demonstrate or defend these points.

1) I don't agree that the "results provide a direct demonstration of the attractor hypothesis in experimental data" without a perturbation test. The model's ability to infer other kinds of dynamics is validated Supp Figure 4 is in synthetic data, but inferring dynamics from real data is a different story, and notoriously difficult to do without testing other dynamical regimes (e.g., O'Shea et al show that the wrong dynamics are inferred with LDS if perturbation data is not included).

By the way, I believe the authors partially misunderstood my point about Mante's work. I know that they didn't fit RNNs to neural data, but they fitted ("trained" if you prefer) RNNs to perform the task, then compared it to data. What I meant here is that I felt that I learned a new dynamical mechanism for contextual decision making for that task. Of course it is important to validate classic attractor hypothesis from data, but even there, without a perturbation test I don't think the result is as definitive as the authors suggest.

This is my strongest criticism so if the authors have any perturbation data to include I would certainly reconsider. The next two criticisms are more related to interpretation, writing, and presentation of their model.

2) The authors did understand my second point about Mante that "RNNs are a type of neural circuit model". I don't agree that their model is more mechanistic than an RNN, mainly because I do not believe that tuning curves in a latent variable constitute a mechanistic model. This is a recurring theme in the paper that I frankly find difficult to accept. A spiking neuron is simply not an inhomogenous Poisson process from some population wide latent variable. Even if that is a useful framework. This is why I find the mechanistic interpretation limiting.

3) I do not agree with the authors that the geometry of neural tuning curves generates neural geometry. I know and love the Kriegeskorte review. But what I mean is that, causally, mechanistically, the tuning curves are not really generating neural geometry. It's the neural circuitry that generates the geometry of neural trajectories, and those generate the tuning curves. I can accept that allowing for heterogeneous tuning curves in the model are important for fitting neural dynamics but I really know how to interpret this without any mechanistic or even normative reason for why there should be such heterogeneous tuning curves. Finally, since the geometry is only generated through these tuning curves (as opposed to being shaped by high-dimensional neural dynamics) I don't see the link between geometry and dynamics that is suggested in the title. At the least I think the authors should have a more accurate title.

Referee #3

(Remarks to the Author)

Besides the fact that the authors continue to use geometry and didn't put the figure in the main text, this is a beautiful paper. I have only a few comments, almost all of them about places where I got confused. I'll leave it to the authors whether they want to make any changes. If they don't, that's fine; as far as I'm concerned, the paper could be published in its present form.

1. l 64-5: "Our modeling framework can generate data with identical dynamics but different geometry".

OK, I can't resist. Here "geometry" just means "tuning curves", right? Which I got only from looking at SI. This sentence -- and, indeed the rest of the paragraph -- would be so much easier to make sense of if you replaced "geometry" with "tuning curves". As mentioned at the start of my review, you're free to ignore this, but it's an important paragraph and it would be helpful if it were crystal clear. And I promise to never mention geometry again. ;)

2. I know the title on line 87 is "Decision dynamics in single neurons", but that didn't stop me from getting confused; for some reason I thought you used the same $\Phi(x)$ for all neurons within a population. Rereading the paper I now see that that's not what you did, but it took me a while to figure out. I think there's an easy fix: in this section, instead of writing $\Phi(x)$, write $\Phi_i(x)$. If you did that everywhere Φ shows up, and also added it to the section starting on line 87, things would be a lot more clear.

3. l 217-9: "Since the population model predicted spike times just as well as single-neuron models, we expect that it recovered the same dynamics as identified in single neurons. Indeed, the dynamics discovered by the population model

were consistent with single-neuron results (Fig. 4b-d)."

I was expecting a comparison between single neuron and population fits; either tuning curves or inferred trajectories, $x(t)$. But I didn't get that. I think a direct comparison would be helpful.

4. Fig. 4e. An insanely minor point: if you could move panel c to where panel e is, and put panel e at the end, that would help people like me who don't pay much attention (when you said Fig. 4b-d, I spent a while looking at panel e, which kind of confused me). I know, not really enough room, but you might be able to shrink panel a.

Version 2:

Reviewer comments:

Referee #2

(Remarks to the Author)

Thanks to the authors for the new analyses. The optogenetic data from Inagaki et al is a nice validation of their method, and the low rank RNN adds a lot of value to the manuscript by providing mechanistic insight regarding heterogeneous tuning. Overall the authors have made some efforts to address the mechanistic and robustness issues I raised previously. I believe these efforts improve the manuscript and I think the method is interesting and the analyses are sound.

In contrast, I am still skeptical as to the novelty of the main pitch of the paper. On one hand, the attractor dynamics that they have found do not seem to constitute a new dynamical mechanism for decision making. The authors have responded that the real finding of their paper is "that complex responses of single neurons in association brain areas arise from the same coding principle as in sensory areas: the neural population dynamics encode simple cognitive variables, while individual neurons have diverse tuning to the cognitive variable, similar to neural tuning curves for sensory stimuli". While I understood that this was the basis for their model, I was surprised to see that this is considered the novelty of their paper. I find the comparison to sensory systems rather weak, since cognitive tasks require computation, not simple encoding of external stimulus statistics. It's not clear to me why, from any normative perspective, neurons should be tuned to a latent cognitive variable that is already performing the task. In the sensory system, the tuning curves are a part of the computation, for example by decomposing high-dimensional visual inputs into independent features. Here, the tuning is post computation. I don't doubt that this framework can be useful as a phenomenological model that can infer latent dynamics - the authors have shown convincingly that this is the case. However I do not find this result in and of itself to be convincing as a computational mechanism, or even particularly exciting. Their concept of geometry is very particular to their model assumptions (see below) and not general nor mechanistic.

Specific comments responding to the authors' rebuttal:

"the reviewer's concern does not apply to our work, as we fit neural activity on single trials, unlike trial-averaged activity used by O'Shea et al"

I agree that fitting single trial activity is an improvement to fitting trial-averaged activity. However, whether it is sufficient to infer dynamics will depend on the noise statistics of the data and the robustness of the method. It is a strong statement to say that the concern doesn't apply at all. The authors should demonstrate the robustness of their inferred dynamics in some way, possibly by using a different method.

"We are unsure how this confusion arose. As far as we know, we never stated that our model is "more mechanistic than an RNN" nor that tuning curves "constitute a mechanistic model". "

This was a typing error on my part. I meant "as mechanistic as an RNN", not "more mechanistic". This is in response to the authors' statement from the previous rebuttal: "our framework offers mechanistic insights similar to methods based on RNN fitting". I hesitate to call the insights found in this paper "mechanistic", and the only part of the model that could be interpreted to me as such would be the heterogeneity of tuning curves, which is one of the core hypotheses. I don't find this to be mechanistic since it is not clear where the tuning comes from (as my point above).

"Population geometry does not generate tuning curves, nor do tuning curves generate population geometry."

I acknowledge the authors' point about a duality between tuning curves and trial-averaged population geometry as per Kriegeskorte & Wei. However, this duality is lost when taking time and trial-by-trial variability into account. Neural circuits generate single-trial population trajectories, which over many trials forms some geometric structure, and these trajectories can be reduced to tuning curves by trial/time averaging and projecting onto single-neuron axes. This is the basis for my claim that the geometry of neural *trajectories* generate tuning curves.

The authors claim several times that "tuning curves of single neurons define the geometry of the population code". I still think this is a very particular assumption that needs to be better addressed in this paper. I think this is really important because there is no standard definition for what population geometry means, and as a field it is important to be precise about these abstract concepts to avoid confusion. One simple work around would be for the authors to simply define early on what they mean by "population geometry", and to acknowledge the other forms of geometry that are not accounted for in their model

assumptions as caveats. In the ideal world they would give an example of where their model fails to recover the correct dynamics and geometry because their assumptions are not met.

“To the best of our understanding, our title does not suggest a link between dynamics and geometry.”

To me, “A and B in premotor cortex” as a title suggests a link between A and B. Otherwise, why would they be in the same paper?

However this point about the title is really a detail. The point I was trying to make was about the specific type of geometry that is addressed in the paper (as in my previous point above). The paper is much stronger in capturing dynamics than capturing geometry. And I personally think a simple title like “The dynamics of choice in premotor cortex” sounds much better.

Referee #3

(Remarks to the Author)

I was happy with the paper last time; I'm no less happy this time.

Point-by-point responses.

Referee #2:

Genkin et al. propose a framework in which neural population dynamics are determined by latent attractor dynamics, with the neural firing rates being determined by tuning curves that are functions of the dynamics. They apply their model to premotor cortical data during decision making in monkeys to show that their results return attractor dynamics consistent with previous theories of decision making in evidence accumulation.

The paper is clearly written for the most part and the analysis is quite nice. I definitely think it would be of high interest to systems neuroscientists interested in population coding. However, as you will see below, my main concern regards the novelty of the results which makes me hesitant to fully recommend, despite my appreciation of the solid modelling framework and strong ties to classic theories of decision making.

Reply:

We thank the reviewer for the positive evaluation of our approach and recognition of the broad significance of our results. As detailed in our responses below, the novelty of our findings is twofold.

First, we reconcile the heterogeneous responses of cortical neurons with the encoding of a single decision variable at the population level. The status quo in the field maintains that single neurons have heterogeneous dynamics during decision-making, reflecting complex computations involved in cognition. Our results show that heterogeneity arises from a fundamentally different coding principle: single neurons encode the same dynamic decision variable with diverse tuning functions.

Second, we reveal the attractor mechanism for decision-making in single-trial experimental data, invalidating influential *ad hoc* hypotheses such as ramping and stepping dynamics. While the attractor mechanism was hypothesized in theoretical models, our work provides a direct demonstration of this mechanism in experimental data.

Our point-by-point responses detail how we clarified the novelty of our findings in the revised manuscript.

Major concerns.

1) While the mathematical modelling is very cool, I am left with the uneasy feeling that I am not sure what I have learned from this analysis that is new. Many classic papers have already studied evidence accumulation in premotor cortex, including the classic computational papers the authors cited for Fig. 5 and classic drift diffusion models.

Reply:

These classic papers proposed several alternative hypotheses about decision-making dynamics in cortical circuits, the most prominent conjectures were attractor (Wang, *Neuron* 2002), drift-diffusion/ramping (Gold & Shadlen, *Ann Rev Neurosci* 2007), and stepping dynamics (Latimer et al., *Science* 2015). However, no conclusive evidence has emerged thus far to arbitrate among these alternative hypotheses in experimental data. Moreover, all these classic hypotheses failed to account for complex heterogeneous responses of cortical neurons during decision-making (Fig. 2b). Classical attractor network models consist of homogeneous pools of neurons, which have identical temporal response profiles and choice selectivity within each pool. Whether the heterogeneous responses of cortical neurons are compatible with the attractor mechanism remained unclear, and attractor models were not tested directly by fitting neural response data on single trials. Similarly, attempts to arbitrate between drift-diffusion and stepping dynamics triggered an extensive debate in the literature with the bewildering conclusion that an equal fraction of neurons was better described by each model class, but the exact outcome was sensitive to nuisance parameters in each model's implementation (Zoltowski et al., *Neuron* 2019). These studies have concluded that different neurons have different dynamics, suggesting that dynamics are not shared across the entire population. A caveat is, however, that all compared stepping and drift-diffusion models produce monotonically ramping trial-averaged firing rates, inconsistent with the response profiles of many experimental neurons (Fig. 2b). No resolution to this conundrum has been suggested thus far.

We propose and test a novel hypothesis that accounts for the heterogeneity of single-neuron responses: neural population dynamics encode a decision variable, while individual neurons have diverse tuning to this decision variable.

Two crucial technical advances within our approach enable us to test this hypothesis in data. First, the inference of nonlinear tuning functions allows us to reconcile the diversity of single-neuron responses with the population-level encoding of a low-dimensional decision variable. By allowing for heterogeneous tuning functions, we reach a conclusion opposite to Zoltowski et al.: our results unambiguously show that single neurons follow the same dynamics on single trials, shared across the population. Second, we perform nonparametric inference over a continuous space of models, whereas previous work compared a small discrete set of models without guarantees that any of these *a priori* chosen models faithfully reflect dynamics in the brain (Latimer et al., *Science* 2015; Zoltowski et al., *Neuron* 2019). Indeed, attractor, drift-diffusion, and stepping dynamics are just special cases within our flexible modeling framework (Supplementary Fig. 4) capable of identifying many other qualitatively distinct dynamics when supported by data (Genkin & Engel, *Nat Mach Intell* 2020; Genkin et al., *Nat Commun* 2021). Thus, our discovery of attractor dynamics is a direct demonstration of the attractor hypothesis in experimental data, but it goes beyond previous models by reconciling the attractor mechanism with heterogeneous single-neuron responses. **We clarify this point in new Supplementary Fig. 4 and on lines 384–400 in the revised Discussion.**

More recently, Mante et al. already showed attractor dynamics governing decision making in a more complex task. This approach is clearly distinct, but can we learn anything new with this model? I feel that there is a potential for disassociating dynamics and geometry through it (hinted at in the title) but this is never quite shown. Is there a way to demonstrate new dynamics that you wouldn't have been able to guess from the manifold geometry itself, in the way that Mante did?

Reply:

Mante et al. did not show attractor dynamics in experimental data. They found an approximate line attractor in a recurrent neural network (RNN) model trained to perform a context-dependent decision-making task. The only correspondence between this RNN and neural data is a qualitative resemblance of their trial-averaged population trajectories projected on a low-dimensional linear subspace obtained with targeted dimensionality reduction (TDR).

For three fundamental reasons, this *qualitative resemblance* of low-dimensional trial-averaged trajectories is insufficient to draw conclusions about single-trial dynamics in neural data. First, task-optimized RNNs can implement different mechanistic solutions, which all produce the same low-dimensional trajectories (Pagan et al, *bioRxiv* 2022). Thus, a task-optimized RNN provides only a hypothesis (a guess) for a possible mechanism, but no evidence that this mechanism exists in experimental data. Second, the same trial-averaged responses can arise from qualitatively distinct dynamics on single trials (e.g., attractor, drift-diffusion or stepping dynamics, Latimer et al., *Science* 2016; Galgali et al., *Nat Neurosci* 2023). Thus, the geometry of trial-averaged population activity does not uniquely define a model of population dynamics on single trials (Galgali et al., *Nat Neurosci* 2023). Third, the geometry of projected low-dimensional trajectories depends on a dimensionality reduction method (Langdon et al, *Nat Rev Neurosci* 2023). Different projection methods yield a distinct low-dimensional geometry, which therefore does not uniquely identify population dynamics. For all these reasons, one cannot guess the dynamics in neural data by simply inspecting the geometry of low-dimensional projections of trial-averaged population activity.

The reviewer insightfully indicates the need for dissociating population dynamics and geometry, which are conflated in the trial-averaged population activity (Supplementary Fig. 2)—a challenge we directly addressed in this paper. Our modeling framework *explicitly dissociates* the dynamics and geometry of neural representations on single trials and enables estimating both simultaneously in experimental data (Fig. 1). The neural representation geometry (manifold) is defined by the combination of tuning functions of all neurons in the population (Kriegeskorte & Wei, *Nat Rev Neurosci* 2021). The dynamics on this manifold are controlled by Eq. 1 independently of the manifold geometry. In particular, using the same tuning functions, we can generate data with the same manifold geometry but different dynamics by choosing different potential shapes (Supplementary Fig. 1). Conversely, using the same potential, we can generate data with the same dynamics but different geometry by choosing different tuning functions. **We clarified this point in the new Supplementary Figs. 1, 2, and on lines 54–69 in Results and lines 367–372 in Discussion in the revised manuscript.**

Maybe separately fitting the error trials could show that these are due to some changes in the dynamics that previously we would have assumed was due to noise, without access to a model that explicitly fits latent dynamics?

Reply:

We thank the reviewer for the great suggestion to examine error trials in more depth. Our models fit spike data from all correct and error trials without receiving any information about animal choice (i.e., the models are not informed about which trials are correct or error). Thus, without assuming *a priori* whether the dynamics are the same or different on correct and error trials, we can answer this question in a data-driven way by inspecting the fitted models.

Our modeling framework dissociates population dynamics and geometry (Supplementary Figs. 1,2) and therefore enables us to test alternative hypotheses about how errors emerge in single-trial population activity. One plausible hypothesis is that single-trial neural trajectories take distinct paths through the population state space on correct versus error trials (Supplementary Fig. 16a), as may be suggested by differences in the corresponding trial-averaged firing rates (Supplementary Fig. 16b). In our framework, this scenario would be revealed in tuning functions being different between the left and right stimulus conditions, in which the left choice is correct versus error, respectively (and vice versa for the right choice). An alternative hypothesis is that all single-trial trajectories leading to the same choice follow the same path through the population state space irrespective of whether this choice is correct or error (Supplementary Fig. 16a), which would be revealed in tuning functions being the same for the left and right stimulus conditions. In this case, differences in trial-averaged responses (Supplementary Fig. 16b) result solely from differences in dynamics between correct and error trials. The geometry of trial-averaged trajectories (Supplementary Fig. 16b) cannot distinguish between these alternative hypotheses, because trial-averaged responses conflate the dynamics and geometry of neural representations on single trials (Supplementary Fig. 2).

Our results support the second hypothesis. We found that the tuning functions were largely invariant across stimulus conditions (Supplementary Fig. 7), indicating that neural trajectories evolve along the same manifold on correct and error trials. The dynamics on correct and error trials were distinct. On correct trials, the flow field drives the dynamics towards the correct-choice boundary (potential slope inclined towards the correct-choice boundary). On error trials, the dynamics first step over the potential barrier due to noise and then the flow field drives the dynamics towards the incorrect-choice boundary (potential slope inclined towards the incorrect-choice boundary). Thus, the flow field drives the dynamics towards opposite ends of the choice manifold on correct versus error trials for the same stimulus (Supplementary Fig. 16c). Moreover, the dynamics leading to the same choice evolve faster when this choice is correct than error, as indicated by a steeper slope of the potential on the side corresponding to the correct choice (Supplementary Fig. 16c). Thus, the geometry of the choice manifold is nearly invariant to whether the choice is correct or error, whereas the speed and direction of how the dynamics evolve along this manifold differ.

These differences in dynamics between correct and error trials are nontrivial. In particular, in a drift-diffusion model, the dynamics leading to correct and error choices evolve in a linear potential with the same constant slope everywhere in the state space and differ only due to noise (Supplementary Fig. 4a). Hence, the flow field drives the dynamics towards the correct-choice boundary on both correct and error trials (Supplementary Fig. 16d). Instead, differences in dynamics on correct and error trials revealed by our model are consistent with the attractor mechanism (Fig. 5). **We present these results in the new Supplementary Figs. 4, 7, 16, new Supplementary Note 3.1, and on lines 267–282 in the revised Results.**

2) The authors should help to solidify the mechanistic interpretation of the framework. The idea of tuning curves in the sensory system is a phenomenological model based on the idea that sensory neurons are extracting latent features (eg orientation of a bar) from sensory input (pixels). But in the context of cognition, it seems unclear to me why there would be a nonlinear mapping between the evidence accumulation variable, which is already encoded by PMd neurons, and the firing rates of individual neurons.

Reply:

The idea that firing rates linearly encode the decision variable is a common intuition, but the question is whether this intuition aligns with experimental data. If PMd neurons had linear tuning to the decision variable, their trial-averaged firing rates would ramp up or down monotonically. However, many PMd neurons show nonmonotonic

temporal response profiles inconsistent with the linear encoding of the decision variable (Fig. 2b). One might then speculate that only a fraction of PMd neurons with the canonical ramping responses encode the decision variable, while the remaining heterogeneous neurons have distinct functional roles and follow different dynamics (Zoltowski et al., *Neuron* 2019; Steinemann et al., *eLife* 2023). Contrasting this view, our analyses reveal a radically different picture: all neurons follow the same dynamics on single trials, and heterogeneous firing rates result from nonlinear tuning of PMd neurons to the latent decision variable.

While nonlinear encoding of a latent cognitive variable may seem counterintuitive at first glance, several pieces of evidence in the literature have already pointed towards this possibility. First, linear encoding models that seek a “choice axis” (a direction in the neural population state space along which activity correlates with animal’s choices) find that the choice axis changes through time in a trial. Specifically, neural trajectories initially move along a single choice axis and then curve onto the next choice axis etc. (Aoi et al., *Nat Neurosci* 2020; Soldado-Magraner et al., *bioRxiv* 2023). Our framework parsimoniously explains this multidimensional choice encoding shifting from axis to axis as a linear approximation to the nonlinear geometry of the choice manifold (similar to approximating a nonlinear function by a combination of linear pieces). Second, nonlinear encoding is not unique to sensory systems and well established, for example, in hippocampus, where neurons encode space and other abstract variables with nonlinear tuning functions (Nieh et al., *Nature* 2021). Finally, the computation of the decision variable arises from nonlinear interactions among neurons within highly complex recurrent circuitry in PMd, and we cannot always expect a uniform linear encoding to emerge in this heterogeneous population of neurons.

On a technical side, our modeling framework allows for the possibility that tuning functions are linear, and we confirmed that our model correctly discovers linear tuning functions on synthetic data with linear ground-truth tuning functions (Supplementary Figs. 3, 8, 9, 10). **We clarified this point on lines 401–412 in the revised Discussion.**

Going back to Mante et al., there the authors fit an RNN to find the attractor dynamics which has clear mechanistic interpretation (if lacking on some biological realism). Can the authors offer (perhaps in the discussion) a way to interpret their model mechanistically...

Reply:

Mante et al. did not fit an RNN to experimental data (see our response to point 1 above).

Other studies indeed fitted RNNs to trial-averaged neural responses and then analyzed dynamical mechanisms in these RNNs (Finkelstein et al., *Nat Neurosci* 2021). A dynamical mechanism describes neural computations as a dynamical system (Vyas et al., *Ann Rev Neurosci* 2020). Since RNNs are high-dimensional dynamical systems, a standard approach to describe dynamical mechanisms in these models is by finding fixed points and linearized dynamics around them (Mante et al., *Nature* 2013; Sussillo & Barak, *Neural Comput* 2013; Finkelstein et al., *Nat Neurosci* 2021; Pagan et al., *bioRxiv* 2023). These standard methods provide only an approximate locally-linear view of low-dimensional dynamics, which arise *implicitly* from high-dimensional dynamical system equations in RNNs.

Our framework yields an *explicit* low-dimensional nonlinear dynamical system model of spike data, and thus identifies dynamical mechanisms *directly* from spikes without an intermediate step of RNN fitting. Therefore, our framework offers mechanistic insights similar to methods based on RNN fitting, but has an advantage of providing a low-dimensional nonlinear dynamical system model, not merely its local linear approximations. Moreover, assumptions made during RNN fitting may distort the inferred mechanisms (Qian et al., *bioRxiv* 2024).

By referring to mechanistic interpretation of RNNs, the reviewer may also allude to the fact that RNNs are a type of neural circuit models, in which dynamics arise from recurrent connectivity. However, interpreting circuit mechanisms that generate low-dimensional dynamics in RNNs remains an open challenge. Most RNN studies analyze only dynamical mechanisms and provide no insight into how particular features of a dynamical system arise from the underlying connectivity. While new approaches begin to establish feasibility of inferring latent connectivity structure that generates low-dimensional dynamics in the data (Langdon & Engel, *bioRxiv* 2022; Valente et al., *NeurIPS* 2022), these emerging methods have not yet been developed to fit neural activity on single trials.

We use a spiking network model to establish a mechanistic interpretation of the one-dimensional decision variable and dynamics identified by our model in PMd (Fig. 5). In the spiking network, the decision-related activity evolves

along one-dimensional stereotypic trajectories, which can be parametrized by a single latent variable, corresponding to the decision variable x in our model (Fig. 5, Supplementary Note 3.2). The firing rate of neuron i at any point x along the one-dimensional trajectory is then given by the tuning function $f_i(x)$. Thus, the decision variable in our model parametrizes one-dimensional neural trajectories arising during decision-making and tuning functions capture the geometry of these trajectories. The potential governing the dynamics along these trajectories shows the same signatures of the attractor mechanism in the spiking network and PMd. However, the choice manifold in PMd spans more linear dimensions than in the spiking network due to heterogeneous tuning of PMd neurons to the decision variable. Thus, PMd and the classical attractor network have the same dynamics but different geometry of choice representation.

Emerging theoretical work has begun to elucidate how recurrent neural networks can generate low-dimensional dynamics with complex manifold geometry defined by heterogeneous tuning of single neurons to latent variables (Pezon et al., *bioRxiv* 2024). This new theoretical approach, just like our statistical framework, explicitly dissociates the low-dimensional latent dynamics and the manifold geometry and enables designing recurrent networks that generate the same latent dynamics with distinct representation geometry (Pezon et al., *bioRxiv* 2024). This emerging theory provides a possible bridge from our findings to the underlying circuit structure, and we find establishing this bridge an exciting direction for future work. **We clarified these points in the new Supplementary Note 3.2, and on lines 287–351 in Results and lines 370–372, lines 377–383, and lines 413–419 in Discussion in the revised manuscript.**

... and the relationship to other population dynamics models that have latent task representations at their core?
3) *I'm surprised that there isn't more discussion about other work in latent neural dynamics, including work from Maneesh Sahani, Srdjan Ostojic, Scott Linderman, John Cunningham ... and surely others, this is just an example. The authors should include more discussion about the relationship between their model and its assumptions to other formulations of latent dynamics which have become increasingly popular in recent years. As a start, how about discussing the relationship between the proposed model and latent linear dynamical systems? How important are the differences between this model and LDS? (nonlinear dynamics, nonlinear emission, Poisson noise from what I see)*

Reply:

Our approach introduces three fundamental advances over other methods: (i) simultaneous inference of nonlinear tuning functions and a nonlinear latent dynamical system, (ii) flexible nonparametric inference, and (iii) principled approach for selecting the complexity of inferred model features.

Most existing models of latent neural dynamics trade off flexibility for interpretability or vice versa. On one hand, flexible high-dimensional recurrent neural networks can approximate any dynamics but do not yield interpretable low-dimensional flow fields (Pandarinath et al., *Nat Methods* 2018; Cohen et al., *bioRxiv* 2020; Schimel et al., *ICLR* 2022; Hurwitz et al., *NeurIPS* 2021). On the other hand, interpretable models, such as linear dynamical systems (Soldado-Magraner et al., *bioRxiv* 2023) or Hidden Markov Models (HMM, Engel et al., *Science* 2016; Racanatesi et al., *Neuron* 2022), rely on rigid parametric assumptions about latent dynamics. These models merely approximate data with *a priori* assumed dynamics but do not allow for discovering the dynamical laws governing latent trajectories. For example, a linear dynamical system can only have a single fixed point, and an HMM approximates any trajectory as a sequence of discrete states even if the ground-truth trajectory is continuous. Such approximations may produce high data likelihood, but it does not prove that the *a priori* chosen model faithfully describes dynamics in the data. For example, if an n -state HMM accurately approximates a continuous trajectory, it does not follow that this trajectory arises from transitions among n metastable attractors.

Our framework belongs to a new class of *flexible* and *intrinsically interpretable* models, which discover the low-dimensional flow field of the latent dynamical system directly from data. Flexible models cover a continuous space of hypotheses about latent dynamics within a single model architecture and therefore enable discovering the flow-field by fitting the model to data. The low-dimensional flow field can be approximated with a set of basis functions (Zhao & Park, *Front Comput Neurosci* 2020), a Gaussian process (Duncker et al., *ICML* 2019), or a deep neural network (Kim et al., *bioRxiv* 2023). All these formulations have computationally intractable data likelihood and thus rely on approximate variational inference maximizing the evidence lower bound of the marginal log-likelihood. In contrast, we model the flow field explicitly as an unknown continuous function and analytically compute the functional derivative of the data

likelihood with respect to this continuous function. This approach enables us to perform the exact maximum likelihood inference without variational approximations and provides an additional advantage of imposing a uniform prior over the space of all continuous functions, avoiding inductive biases of the approximation schemes (Zhao & Park, *Front Comput Neurosci* 2020; Duncker et al., *ICML* 2019; Kim et al., *bioRxiv* 2023). In all these models, the inferred low-dimensional flow-field is immediately interpretable, in contrast to models that represent the flow-field implicitly within a high-dimensional connectivity matrix of an RNN and therefore lack interpretability (Pandarinath et al., *Nat Methods* 2018; Keshtkaran et al., *Nat Methods* 2022).

All flexible models are generally under-constrained by data and require regularization. However, choosing the optimal regularization level in flexible models to achieve correct interpretation of dynamics in the data is nontrivial (Genkin & Engel, *Nat Mach Intell* 2020). The classical bias-variance tradeoff, which is the foundation of validation-based model selection, does not hold for flexible models with high capacity (Belkin et al., *PNAS* 2019). As a result, flexible models optimized for predictive performance on validation data often contain spurious features and cannot be reliably interpreted (Genkin & Engel, *Nat Mach Intell* 2020; Wang et al., *NeurIPS* 2023). To overcome this problem, we developed an alternative model selection strategy that identifies models with correct interpretation by comparing features of models fitted on different data samples to separate true features from noise (Genkin & Engel, *Nat Mach Intell* 2020). Other methods for the flow field inference did not consider the problem of model selection and set the regularization level *ad hoc* to produce compelling results (Zhao & Park, *Front Comput Neurosci* 2020; Duncker et al., *ICML* 2019; Kim et al., *bioRxiv* 2023). The *ad hoc* regularization undermines the interpretation of the inferred flow field, because the model may be underfitted or overfitted if the regularization penalizes the model complexity too much or too little, which remains unknown in the absence of diagnostic tests.

The inference of nonlinear tuning functions simultaneously with a nonlinear latent dynamical system is unique to our study. Previous methods that inferred nonlinear tuning functions of neurons to latent variables either used a linear latent dynamical system (Gao et al., *arXiv* 2016) or approximated latent trajectories with a Gaussian process without modeling a dynamical system governing these trajectories (Wu et al., *NeurIPS* 2017). These methods approximated tuning functions either with a feedforward neural network (Gao et al., *arXiv* 2016) or with a Gaussian process (Wu et al., *NeurIPS* 2017). In both formulations, the data likelihood is computationally intractable, therefore these methods relied on approximate variational inference maximizing the evidence lower bound of the marginal log-likelihood. In contrast, we model tuning functions explicitly as unknown continuous functions and analytically compute functional derivatives of the data likelihood with respect to these continuous functions, which enables us to perform the exact maximum likelihood inference of all model components avoiding biases introduced by variational approximations.

Like many other methods, we model spikes of each neuron as a doubly stochastic Poisson process with an instantaneous firing rate varying as a function of the latent state. Doubly stochastic models attribute spiking irregularity on fast timescales to a discrete point process and therefore allow the single-trial firing rate to change smoothly in time. Separating shot noise arising from discrete spikes is necessary to obtain smoothly varying firing rates on single trials. In contrast to virtually all other methods, our approach does not bin spikes and processes data spike-by-spike in continuous time and hence does not depend on an arbitrary bin-size hyperparameter. **We clarified these points in the new Supplementary Note 3.3 and on lines 359–383 in the revised Discussion.**

4) *On this note, I was underwhelmed by the comparison in Fig. 4e to a logistic decoder, as the decoder had completely different assumptions from the authors' model (e.g., being smoothed, fed in trial averaged activity etc).*

Reply:

There was a typo in Methods on line 624 in the original manuscript. The supervised decoder was trained to predict animal choices from spike counts on single trials measured in 75ms bins sliding in 10ms steps, resulting in a 42-dimensional input vector to the decoder per neuron on each trial. We chose a supervised decoder as a baseline model, because supervised decoders are the gold-standard for assessing the amount of information about a behavioral variable in neural activity (Quiari Quiroga & Panzeri, *Nat Rev Neurosci* 2009). **We clarified the description of the decoder on lines 252–254 in Results and corrected the text on lines 886–892 in Methods in the revised manuscript.**

It would be more illuminating to strip down their model step by step to see what assumptions are actually important to predict performance. For example, replacing one or both of $d\Phi/dx$ and f by linear terms would help to determine what is actually important for decision dynamics (it looks like the potential wells can be well captured by quadratics in which case $d\Phi/dx$ would be linear). This is a specific example but it's the only place I saw an actual benchmarking plot to another method (other than PSTH which is too basic).

Reply:

We wish to clarify that the inferred shapes of tuning functions and potential *are not assumptions* but results of fitting our model to data. Our flexible inference could produce other shapes as well (e.g., linear potential or linear tuning functions) if the data would support this outcome (Supplementary Figs. 4, 8, 9, 10). We ask the question about what are the geometry and dynamics of neural activity on single trials and our nonparametric model has the full flexibility to discover this structure from data.

Following the reviewer's excellent suggestion, we evaluated the likelihood on validation data of the stripped down versions of the inferred model, in which we replaced the tuning functions, potential or both with linear approximations. All stripped down models had significantly lower likelihood than the original model (Supplementary Fig. 15). We also approximated the inferred potential with a quadratic function, and this version of the model produced slightly lower likelihood than the original model, consistent with the observation that the discovered potential shapes are approximately parabolic. The point here, however, is not to argue about which closed-form analytical function can best approximate the potential and tuning functions we discovered in PMd. *A priori*, there are myriad analytical functions that may plausibly describe neural dynamics, and the standard approaches perform model comparison among a finite subset of such *a priori* chosen analytical models. However, these approaches do not guarantee that any of the *a priori* guessed models faithfully describe dynamics in the data. In contrast, our nonparametric approach discovers the shapes of potential and tuning functions from data, and then *post hoc* we may wonder what analytical functions are similar to the discovered shapes.

We also used the stripped down models to predict animal choices as the reviewer suggested. The accuracy of choice prediction was similar between the original and all stripped down models (Supplementary Fig. 15). This result illustrates that predicting the binary choice is a relatively easy task, and many different models (including supervised decoder and all stripped down models) can do it nearly equally well. Thus, the accuracy of choice prediction alone is not sufficient to argue about what single-trial dynamics are consistent with neural responses in PMd. **We present these results in the new Supplementary Fig. 15 and on lines 231–233 and 260–262 in the revised Results.**

5) In some places the statistical analysis could be improved or clarified.

i. For example, permutation test to get an idea of what baseline is for the variance explained or for accuracy plots.

Reply:

We performed a permutation test to estimate the baseline for the explained variation R^2 in spike times on single trials. For all single-neuron models that provided a good fit, we inferred the latent trajectories on validation trials using the Viterbi algorithm and predicted single-trial firing rates of each neuron from these latent trajectories using their tuning functions. We then computed the explained spike-time variation R^2 on shuffled trials: we used the firing rate predicted on one trial to compute R^2 for spikes on another randomly chosen trial. The explained variation in spike times on the shuffled trials was -0.10 $[-0.24, -0.04]$ for monkey T and -0.02 $[-0.15, 0.01]$ for monkey O (median [Q1, Q3]), the negative values indicate that the shuffled prediction was worse than predicting a constant firing rate (which corresponds to $CV_{\text{total}}^2 = CV_{\text{residual}}^2$, $R^2=0$). The shuffled prediction was also significantly worse than the prediction based on PSTH. This result confirms that our models explain the spike times on single trials significantly better than the permutation baseline and PSTH. **We describe these analyses on lines 129–134 in the revised Results.**

When splitting the data into D1 and D2 for cross validation – how are these defined? Depending on the choice you might have temporal correlations between your datasets.

Reply:

We split all trials in halves by assigning even trials to D1 and odd trials to D2. In the experiment, stimulus conditions were sampled randomly on each trial. Therefore, the time difference between two adjacent trials of the same condition varies broadly, limiting possible temporal correlations between D1 and D2. **We clarified this point on lines 667–670 in the revised Methods.**

ii. L153-159 I feel uncomfortable interpreting this analysis since it's only based on a single animal and the exclusion criterion for Monkey O is unclear. Was 88 neurons really not enough for a leave-one-out analysis?

Reply:

The leave-one-neuron-out validation requires populations of neurons recorded simultaneously in the same session. While we had 88 neurons in total from monkey O, the yield of simultaneously recorded neurons was relatively low in this animal (number of simultaneously recorded neurons in a session: median 4, range from 3 to 7). Therefore, we chose to not perform the leave-one-neuron-out analysis for monkey O in the original manuscript. We have now performed the leave-one-neuron-out validation for all populations from monkey O. The log-likelihood was higher for the population model in the leave-one-neuron-out validation than for the baseline prediction based on the neuron's own trial-averaged firing rate, which was significant in monkey T and did not reach statistical significance in monkey O (Fig. 4a; $\log[L_{\text{LONO}}/L_{\text{PSTH}}]$, median [Q1, Q3]; monkey T: 48.9 [24.5, 88.6], $p < 10^{-10}$, $n=89$; monkey O: 6.5 [-3.4, 11.9], $p=0.07$, $n=59$, Wilcoxon signed-rank test). While this result in monkey O did not reach statistical significance it was consistent with the results in monkey T. **We present these analyses in the updated Fig. 4a and on lines 209–213 in the revised Results.**

iii. A lot of the results regarding the potential well shapes are qualitative and should have more quantitative descriptions of how this was assessed

Reply:

We quantified the potential shapes by counting the number of barriers in the potential (Results lines 107–112 and Methods lines 501–523 in the original manuscript). Further, we quantified the relationship between the potential slope and stimulus difficulty (Fig. 3i and Results lines 112–116 in the original manuscript). Finally, we quantified the shapes of tuning functions that define the geometry of the decision manifold using principal component analysis (Fig. 5f and Results lines 112–116 in the original manuscript).

We now added new analysis quantifying the similarity of inferred potential shapes across single neurons and populations. We computed correlation coefficients between all pairs of potentials for single neurons and populations in each stimulus condition. High values of the correlation coefficient indicated that the potential shapes were very similar across all single neuron and population models (average correlation coefficient, mean \pm std; monkey T: single neuron models 0.86 ± 0.16 , population models 0.93 ± 0.09 ; monkey O: single neuron models 0.88 ± 0.14 , population models 0.88 ± 0.14). **We present these analyses on lines 153–155 and 223–226 in the revised Results.**

iv. Overall I

6) The authors' mathematical framework is extensive and has been developed over multiple papers over the past few years. It is difficult to assess how much of that framework is new for this paper, or whether they simply applied an already-existing framework to a new dataset. Some clarification here would help, both in the main text, but also in the supplementary methods where they should point out which details (if any) have been developed especially for this study.

Reply:

Our modeling approach in this study introduces several innovations that go beyond existing methods, including our own previous work. In this paper, for the first time, we develop the inference of nonlinear tuning functions *simultaneously* with the latent dynamics. Our previously published work assumed that tuning functions are known and we inferred only the latent dynamical system (Genkin & Engel, *Nat Mach Intell* 2020; Genkin et al., *Nat Commun* 2021). In addition, our previous studies were theoretical and this paper is the first application of our approach to experimental data for scientific discovery.

In slightly more detail, our previous two publications presented theoretical advances in model selection and nonstationary dynamics inference, forming the stepping stones towards developing our comprehensive computational approach presented here.

Our first publication (Genkin & Engel, *Nat Mach Intell* 2020) showed that the standard approach of selecting the parameters of a flexible model by optimizing its predictions (likelihood) on validation data often produces models with spurious features that cannot be interpreted reliably. This behavior arises because validation-based model selection relies on the classical U-shaped bias-variance tradeoff, which, however, does not hold for flexible models with high capacity (Belkin et al., *PNAS* 2019). We developed an alternative strategy for identifying models with correct interpretation by comparing model features discovered from different data samples to separate true features from noise. Our work is unique in addressing overfitting to validation data in flexible models. All other flexible models of latent neural dynamics did not consider this problem, which undermines the interpretation of their inferred features.

Our second publication (Genkin et al., *Nat Commun* 2021) developed the inference of nonstationary latent dynamics, which accounts for the non-equilibrium initial and final states of the observed system and for the possibility that the system's dynamics define the duration of observations. Using synthetic data, we showed that our approach can correctly infer dynamics corresponding to the ramping and stepping models of decision-making proposed previously.

We clarified this point on line 26 in Introduction and lines 531–534 in Methods in the revised manuscript.

Minor comments

- Typo on line 57, should be x_0 or $x(0)$

Reply:

We fixed the typo.

- *The authors should mention somewhere that equation 1 only describes fixed point attractor dynamics (no limit cycles)*

Reply:

In one dimension, there are no rotational forces (curl is zero), hence any force results from the gradient of a potential. Thus, Eq. 1 is the most general equation for one-dimensional dynamics. We choose potential to visualize our results because potential landscapes with barriers and valleys are intuitively interpretable, making the inferred dynamics accessible for a wide audience. In practice, we directly fit the force $F(x)$ in the latent dynamical system. We analytically compute the variational derivative of the likelihood with respect to the force $F(x)$ and use it to optimize the force with the ADAM algorithm (Supplementary Note 2.2). We then obtain the potential from $F(x)$ via Eq. 6. Thus, our optimization is not limited by the representation of the flow field via potential. **We clarified this point on lines 482–485 in the revised Methods.**

- *The authors should mention somewhere that Phi is condition dependent*

Reply:

We mention that potential is condition dependent throughout the main paper. Moreover, we plot four potentials, one for each stimulus condition, in all main figures showing our results.

- *In equation 1, why is the noise amplitude also scaling the deterministic term? This means the gradient down the well is stronger when there is higher-variance noise. Could this lead to any issues comparing Phi across plots, if this scaling is not incorporated?*

Reply:

The parametrization of Eq. 1 in which the noise magnitude D scales the potential is a convenient choice which makes the equilibrium probability density of the Langevin dynamics invariant to the noise magnitude. This parametrization does not restrict in any way the space of dynamical systems spanned by our models. One can think about this parametrization as measuring the potential height in units of D . There are no issues with comparing potentials across different conditions and neurons, because we scale all potentials by D after fitting for plotting and comparisons.

We clarified this point on lines 486–490 in the revised Methods.

- L337 onwards, I personally find the time indexing odd as e.g. $Y(t)$ does not depend on t . But I leave it for the authors to decide how to clarify this.

Reply:

$Y(t)$ is a sample from a point process, which is a stochastic process modeling sequences of random events in time. We agree this notation is standard for stochastic processes and less common for samples. Nevertheless, we would like to keep this notation to indicate that the data are a sequence of times, as we did not find a clearly better alternative. **We clarified this point on lines 502 in the revised Methods.**

- Equation 4, some more explanation of A would be helpful even at a high level

Reply:

We derived the likelihood functional for nonstationary latent Langevin dynamics previously (Genkin et al., *Nat Commun* 2021). This previous work is entirely dedicated to the mathematical derivation, intuitive explanation, and numerical testing of the nonparametric inference of nonstationary latent Langevin dynamics. In brief, the nonstationary data $Y(t)$ arise in non-equilibrium systems that perform computations. Neural dynamics during decision-making is an example of such non-equilibrium computation. Neural activity transiently evolves from the initial state at the stimulus onset until a choice is made, and different choices correspond to different terminal states of neural activity. The initial and terminal states are fundamentally different from the equilibrium state of the system. In a reaction-time task, a subject reports the choice as soon as the neural trajectory reaches a decision boundary for the first time. Thus, trials have variable durations defined by the neural dynamics itself, and the latent trajectory always terminates at one of the decision boundaries at the trial end. Two components in our framework account for the statistics of latent trajectories in this case. First, the absorbing boundary conditions ensure that latent trajectories reaching a boundary before the trial end do not contribute to the likelihood. Second, the absorption operator A enforces that the likelihood includes only trajectories terminating on the boundaries at the trial end time. Without the absorption operator, the likelihood includes all trajectories that terminate anywhere in the latent space and do not reach the domain boundaries before the trial end. Omitting either of these components results in incorrect inference, in which erroneous features arise in the dynamics due to nonstationary data distribution (Genkin et al., *Nat Commun* 2021). **We clarified this point on lines 518–530 in the revised Methods.**

- How is the number of eigenvectors chosen? I didn't see this explained anywhere. Is it the same as the discretization N ? If not, how was this number chosen?

Reply:

For all model fits, we retain all eigenvectors of the Fokker-Planck operator. The number of retained eigenvectors is $N_r = N - 2 = 447$, where $N = 449$ is the size of the spectral elements method (SEM) grid. The dimensionality of the discretized Fokker-Planck operator is $N - 2$ because the boundary conditions constrain the solution space, reducing the effective dimensionality of the problem from N degrees of freedom to $N - 2$. **We clarified this point in the revised Supplementary Note 2.1.**

- The authors could further discuss Finkelstein et al., 2019, *Nature Neuro*, which is related thematically although a different region

Reply:

We believe the reviewer refers to Finkelstein et al., *Nat Neurosci* 2021. This work tests the hypothesis that attractor dynamics support working memory maintenance in the anterior lateral motor cortex (ALM) of mice by examining effects of distractors presented during the delay period on behavior and neural activity. This work also fits trial-averaged neural responses with RNNs and analyzes dynamical mechanisms in these RNNs. The design and findings of this study are conceptually similar to earlier work in primates that provided support for attractor hypothesis of working memory by examining effects of distractors on behavior and neural activity in frontoparietal cortex (Suzuki & Gottlieb, *Nat Neurosci*, 2012; Murray et al., *J Neurosci*, 2017). Advancing beyond these previous studies of working memory, our

work provides a *direct demonstration* of attractor dynamics in *single-trial* neural activity during decision-making and reconciles the attractor mechanism with heterogeneous responses of single neurons. **We clarified this point on lines 367–372 and 377–381 in the revised Discussion.**

Referee #3:

I have a love-hate relationship with this paper. I love the result, but I hate the fact that it was so hard for me to extract it. I partially blame the field, for encouraging words like "representation" and "geometry", both of which should be banned from all papers. OK, geometry can be used, but only if it's preceded by "Euclidean" or "non-Euclidean", but not when referring to neural activity. And representation never.

Reply:

We truly appreciate this feedback. We put forth our best effort revising the paper to present our findings in simpler terms and explain the meaning of more technical terms to make our work accessible to a broad audience. We still wish to use the terms “neural population geometry” or “representation geometry”, because the main contribution of our work is to reveal a unifying geometric principle for neural encoding of sensory and dynamic cognitive variables.

Our work connects two so far disparate approaches for understanding heterogeneous neural responses. On one hand, neural population geometry relates diverse tuning functions of single neurons to the structure of the population code, but it has only been applied to static externally observable variables. On the other hand, many methods have been developed for modeling neural population dynamics on single trials, but the concept of a tuning function has been completely missing in this literature. Virtually all previous models of population dynamics assume a rigid monotonic relationship between firing rates of single neurons and latent states, and thus infer complex dynamics with more latent dimensions that may not directly correspond to the encoded variables. Our results counter the widespread assumption that heterogeneous neural responses in association brain areas reflect complex dynamics involved in cognition. Instead, we find that the neural population dynamics encode a simple cognitive variable, while individual neurons have diverse tuning to the cognitive variable, similar to neural tuning curves for sensory stimuli. This insight was enabled by our computational approach which explicitly dissociates the dynamics and geometry of neural representations on single trials (Fig. 1).

Our use of the term geometry aligns with its well-established definition for describing neural population codes for static external variables. We wish to make this correspondence explicit because it enables a parsimonious explanation of heterogeneous neural responses: the population activity evolves along a one-dimensional manifold encoding the decision variable, and tuning curves of all neurons define the shape of this manifold (geometry) in the population state space. Emerging theoretical work has begun to elucidate how recurrent neural networks can generate low-dimensional dynamics with complex manifold geometry defined by heterogeneous tuning of single neurons to latent variables (Pezon et al., *bioRxiv* 2024). This emerging theory and our statistical framework align suggesting a path forward for understanding cognitive computations in heterogeneous recurrent circuits.

We have revised the entire paper from the ground up to clarify these ideas, as we detail in our responses below.

So let me start with what the authors did, which was pretty simple: they mapped sensory input to neural activity. The sensory input was a checkerboard stimulus composed of red and green squares; these were summarized by their difficulty, so there were effectively 4 stimuli: easy left, easy right, hard left and hard right. (Easy and hard refer to the difficulty of telling whether there were more green than red squares; left and right refer to the button the monkeys pressed.) The neural activity was in primate dorsal premotor cortex (PMd). So basically they built a map: $r_i(t) = g_i(s, t)$ where $r_i(t)$ is the firing rate of neuron i , s is stimulus, t is time, and $g_i(s, t)$ is some function.

Reply:

We thank the reviewer for the accurate summary of our approach. A small but important detail is that left and right correspond to the response side indicated by the stimulus, not to the monkey’s choice. Our unsupervised model receives no information about animal choices during fitting.

That sounds a bit mundane, but what was interesting was the function, $g_i(s, t)$. It consisted of two parts: one was a "decision" variable, x , which evolved according to a stochastic differential equation that depended on the stimulus; the other was the spike generation process, where neuron i emitted spikes according to a Poisson process with rate $f_i(x)$, and did not depend (directly) on the stimulus. So the model looked like

$$dx/dt = h(s,x)$$

$$r_i(t) = f_i(x)$$

where x is a 1-D variable (and $h(s,x)$ contains white noise, so x obeys a stochastic differential equation, but that's a detail).

And it gets better. First, the monkey made a decision when x reached a boundary (left when $x=+1$; right when $x=-1$). So x really can be interpreted as a decision variable. Second, and, I think, even more important, the "tuning curves", $f_i(x)$, were independent of the stimulus.

So this is very cool, and somewhat unexpected. We're used to low-dimensional trajectories. But here the activity follows the same one-dimensional trajectory on every trial. (OK, not exactly the same; there's noise, because x obeys a stochastic differential equation. But we never expected neurons to be noise-free.) The only thing that varies from trial to trial is the speed and direction along the 1-D trajectory.

Why do we care? I think it's because it has implications for what's going on in PMd. Let's write a very general equation for the membrane potential on neuron i in PMd,

$$dV_i/dt = f(\text{recurrent spikes, external spikes}).$$

Here "recurrent spikes" refers to spikes from within PMd and "external spikes" refers to spikes from other areas. There are two main ways this equation could be consistent with their results:

1) The external spikes are more or less constant in time, and they push the activity along a 1-D manifold.

2) The external spikes themselves evolve along a 1-D manifold (that corresponds to x), and the activity in PMd is driven to a fixed point that depends on the external spikes.

(1) seems a lot more likely to me, but (2) is also possible, and it could even be a combination. (And maybe something I haven't seen thought of?) But it seems to make strong, experimentally testable predictions about what PMd is doing, at least in this task.

So that's what I loved about the paper. Now what I didn't like.

Reply:

We thank the reviewer for this enthusiastic assessment and recognition of the broad significance and novelty of our findings, as well as for suggestions that helped us improve the paper.

First, I wasn't entirely convinced of their results. Meaning I wasn't entirely convinced of the above model ($dx/dt=h(s, x)$; $r_i(t)=f_i(x)$), for a number of reasons.

1) Supposedly Fig. 3a shows that their model explained the variance better than the PSTH. But that figure didn't make a lot of sense to me. The variance explained, which came (I believe) from Eq. 10, depends on the CV of the interspike

interval distribution, which measures how much the true firing rate changes over time. Which has nothing to do with model fit. Or did I misinterpret?

Reply:

In the original manuscript, we chose R^2 because of the advantage that its value is directly interpretable as the fraction of the total variation in the data explained by a model and therefore easy to compare across different models. For example, our model explains about twice as much variation in spike times than PSTH (Fig. 3a in the original manuscript). We designed our R^2 metric specifically for measuring variation in *spike times on single trials*, which is a new approach based on doubly stochastic point processes. We agree with the reviewer that since our R^2 metric is innovative, a standard likelihood would be more familiar to most readers. Therefore, in the revised manuscript, we switch to likelihood in all analyses comparing different models (e.g., Fig. 3a, Fig. 4a), and we use R^2 only to quantify the fraction of the total variation in spike times captured by our models (Fig. 3b).

We explain here in brief how our R^2 metric quantifies the goodness of model fit. We use the standard coefficient of determination R^2 defined as the proportion of the total variation in the data that is predicted by a statistical model. Since we model single-trial dynamics, our metric quantifies the variation in spike times on single trials. In Eq. 12, CV^2_{total} is the squared coefficient of variation of ISIs, which quantifies the total variation in the data. CV^2_{residual} is the residual variation in ISIs unexplained by the model. Our model predicts the firing rate on single trials but not individual spikes. Hence, the residual variation is the variation in spike times after accounting for the firing rate variation predicted by the model.

To compute the residual variation, we use the time rescaling theorem for doubly stochastic point processes (Brown et al., *Neural Comp* 2002). For a doubly stochastic point process, the total variation in spike times arises from two sources: the variability of the firing rate and the variability of the point process generating spikes from this firing rate. The time rescaling theorem states that we can eliminate the firing rate variation by mapping the spike times from the real time t to the operational time t' via squeezing or stretching the time locally in proportion to the cumulative firing rate. Accordingly, we use our model to predict the instantaneous firing rate of a neuron on each trial, map spikes to the operational time using this predicted firing rate, and compute the residual ISI variation CV^2_{residual} , which is the squared coefficient of variation of rescaled ISIs in the operational time. If we rescale time t using the ground-truth firing rate, then the variation in rescaled ISIs in the operational time t' reflects solely the point process variability. If the firing rate on single trials is not predicted accurately, then the firing rate variability contributes to the variation in rescaled ISIs resulting in larger CV^2_{residual} and lower R^2 . **We revised the description of the R^2 metric on lines 752–789 in the revised Methods. We also replaced R^2 with log-likelihood ratio in all analyses comparing different models (Fig. 3a, Fig. 4a, Supplementary Figs. 7, 15).**

In any case, the authors should do something simpler, and more standard. For instance, they could compare the log likelihood of the model versus a firing rate computed from the PSTH, or from a GLM.

Reply:

Following the reviewer's suggestion, we compared the likelihood of spike data between our model and the PSTH. We compute the PSTH likelihood using the standard likelihood formula for an inhomogeneous Poisson process (Dayan & Abbott, *Theoretical Neuroscience* 2001) with the instantaneous firing rate given by the trial-averaged firing rate computed for each chosen side (left vs. right) and stimulus difficulty (easy vs. hard). Single-neuron model produced significantly higher likelihood than the PSTH (Fig. 3a, log-likelihood ratio $\log[L_{\text{model}}/L_{\text{PSTH}}]$, median [Q1, Q3]; monkey T: 21.2 [7.27, 48.5], $p < 10^{-10}$, $n = 117$; monkey O: 9.39 [1.73, 25.6], $p = 1.3 \cdot 10^{-6}$, $n = 67$, Wilcoxon signed-rank test). This result indicates that single-trial firing rates deviate significantly from their trial-average, and our model successfully captures this trial-to-trial variation. Moreover, the population model had similar or higher likelihood than the single-neuron model (Fig. 4a; $\log[L_{\text{population}}/L_{\text{single neuron}}]$, median [Q1, Q3]; monkey T: 190.4 [91.8, 290.3], $p = 0.001$, $n = 11$; monkey O: 4.0 [-35.6, 9.4], $p = 0.74$, $n = 13$, Wilcoxon signed-rank test), which indicates that single neurons participate in the same shared dynamics unfolding on the population level. **We present these new results in updated Figs. 3a and 4a, on lines 119–125 and 199–203 in Results, and on lines 821–862 in Methods in the revised manuscript.**

We performed further comparisons of the likelihood between our model and several baseline models on suggestion of Reviewer 2. We evaluated the likelihood on validation data of the stripped down versions of the inferred models, in which we replaced the tuning functions, potential or both with linear approximations. All stripped down models had significantly lower likelihood than the original models. **We present these results in the new Supplementary Fig. 15 and on lines 231–233 in the revised Results.**

That said, I personally don't care that much if their model fits better than the PSTH or a GLM; that doesn't seem to be the point. What I really want to know is whether it's a reasonable fit. For that maybe they could just report log likelihood of their model relative to the expected log likelihood if the model were correct? Or drop this altogether? The next couple points are, I think, more important.

Reply:

We compared the performance of our model to the ceiling performance expected if the model was correct (Fig. 3b). For a doubly stochastic point process, the value of our R^2 metric can never reach 1, because our models predict firing rates on single trials and do not explain the point process variability. If the firing rate prediction was correct on each trial, then the residual variation CV^2_{residual} would match exactly the expected point process variability CV^2_{pp} of ISIs in the operational time. For an inhomogeneous Poisson process, the expected CV^2_{pp} of ISIs in the operational time equals one (Brown et al., *Neural Comp* 2002). However, the spiking of neurons across many brain regions deviates significantly from the Poisson irregularity. In particular, most neurons in the parietal and premotor cortical areas spike more regularly than expected for an inhomogeneous Poisson process (Maimon & Assad, *Neuron* 2009; Shinomoto et al., *PLoS Comput Biol* 2009). Therefore, we use an independent method based on doubly stochastic renewal point processes to estimate the point process variability CV^2_{pp} for each neuron (Aghamohammadi et al., *bioRxiv* 2024). This method does not require knowledge of the firing rate dynamics on single trials and assumes only that the firing rate evolves smoothly in time. Across neurons, we observed a tight correspondence between the residual variation in spike times unexplained by our model CV^2_{residual} and the independently estimated point process variability CV^2_{pp} , which is the expected residual variation if the firing rate prediction was correct on each trial (Fig. 3b). The tight correspondence between CV^2_{residual} and CV^2_{pp} indicates that our model accounts for nearly all explainable firing-rate variance in the data close to the ceiling performance. **We clarified this point on lines 135–144 in Results and lines 784–815 in Methods in the revised manuscript.**

We do not see how to perform an analogous analysis for the expected likelihood if the model was correct, because it would require knowledge of the ground-truth model for the data, which is not possible.

2) Lines 75-6: *"We noticed that the inferred tuning functions, initial state distribution $p_0(x)$ and noise magnitude D were similar across four stimulus conditions and only the potential shapes were different (Supplementary Fig. 2)."*

"Similar" is a squishy word; the authors need to be quantify how similar. Especially since I looked at Supplementary Fig. 2, and there are clear differences (especially where things are undersampled). I can suggest a simple test: run the model with all the tuning functions the same and run them with different tuning curves for each stimulus, and compare the log likelihoods under the two conditions. If they're not too far off, declare victory.

Reply:

We performed the analysis suggested by the reviewer for all converged populations (11 from monkey T, 14 from monkey O) and 52 single neurons (36 neurons from 3 sessions in monkey T and 16 neurons from 3 sessions in monkey O). We performed this analysis only for a subset of single neurons, because refitting all single neurons with a separate model in each stimulus condition is computationally costly, as it involves four times the total number of fits that we have already performed. The likelihood was not significantly different between population models fitted with shared and separate tuning functions in each stimulus condition (Supplementary Fig. 7c; $\log[L_{\text{shared}}/L_{\text{separate}}]$, median [Q1, Q3]; monkey T: 13.5 [−35.6, 72.7], $p=0.58$, $n=11$; monkey O: −4.6 [−18.8, 18.8], $p=0.95$, $n=14$, Wilcoxon signed-rank test). For the single-neuron model, the likelihood was only slightly lower for the model fitted with shared than separate tuning functions (Supplementary Fig. 7a; $\log[L_{\text{shared}}/L_{\text{separate}}]$, median [Q1, Q3]; monkey T: −6.10 [−17.45,

1.96], n=36; monkey O: $-18.60 [-31.27, -6.58]$, n=16) and within the range obtained on synthetic data from the ground-truth model with shared tuning functions and other statistics (number of trials, firing rate, and dynamics) matched to experimental data (Supplementary Fig. 7a; $\log[L_{\text{shared}}/L_{\text{separate}}]$, median [Q1, Q3]; shared ground-truth model: $1.33 [-8.75, 3.68]$, n=6). Furthermore, the inferred tuning functions were consistent between shared and separate models, as indicated by a high value of their correlation coefficient (Supplementary Fig. 7b; median [Q1, Q3]; monkey T: $0.91 [0.81, 0.93]$, n=36; monkey O: $0.94 [0.91, 0.95]$, n=16). These results support the conclusion that tuning functions were stable across stimulus conditions. **We present these results in the new Supplementary Fig. 7, and on lines 97–111 and 191–196 in the revised Results.**

The reviewer is correct that when fitting single-neuron models with separate tuning functions in each stimulus condition, the inference is less accurate in the undersampled regions (near the incorrect-choice boundary in easy stimulus conditions). In such cases, the inferred tuning functions appear different across stimulus conditions (Supplementary Fig. 5, gray highlight). We tested on synthetic data that this discrepancy likely results from undersampling rather than *bona fide* differences in tuning functions across conditions. We generated spikes with statistics similar to the PMd data from a ground-truth model that had the same tuning functions in four stimulus conditions. When fitting these synthetic data with models that had separate tuning functions in each stimulus condition, we observed differences in tuning functions in the undersampled regions similar to the experimental data (Supplementary Fig. 6). **We present these results in the new Supplementary Figs. 6.**

3) Two sets of simulations were run. In the first set, each neuron could have a different trajectory; that is, for neuron i there was a trajectory $x_i(t)$. In the second set (starting on line 122), all the neurons had a single trajectory (there was just $x(t)$). What seems most important for their model is to show that when using only one $x(t)$, the fits were still good. But again, they used the variance explained using Eq. 10, which I didn't understand at all. (And again, possibly because I misinterpreted it.) But here, why not do something simple: fit the model assuming individual trajectories $x_i(t)$, then fit it using a single trajectory $x(t)$, and compare the log likelihoods; in both case with the tuning curves, $f_i(x)$, independent of the stimulus.

I personally really hope that when they use a single latent trajectory and tuning curves that don't depend on stimulus, they do a good job predicting spike trains. Note that it doesn't have to be perfect, just good. But it's a strong claim, and it needs strong evidence.

Reply:

We performed the analysis suggested by the reviewer and found that the population model had similar or higher likelihood than the single-neuron model (Fig. 4a; $\log[L_{\text{population}}/L_{\text{single neuron}}]$, median [Q1, Q3]; monkey T: $190.4 [91.8, 290.3]$, $p=0.001$, $n=11$; monkey O: $4.0 [-35.6, 9.4]$, $p=0.74$, $n=13$, Wilcoxon signed-rank test). This result further strengthens our conclusion that the latent dynamical variable identified by our models is shared by all neurons in the population. **We present these results in the updated Fig. 4a and on lines 199–203 in the revised Results.**

The second reason I didn't like this paper is that It took me forever to extract the above picture. (Which hopefully is correct!) I think this was because the authors never spelled out what they were doing in a way I could understand. Consider lines 10-12,

"We hypothesized that neural encoding of dynamic cognitive variables follows the same principle as sensory variables: the encoded variable determines the topology of neural representation, and heterogeneous tuning curves of single neurons define the representation geometry."

I simply could not parse this (see first sentence of my review). I now know what it means, but only because I read the paper. About five times. So I had to absorb every fact in case it was important, and piece together a story on my own. If I hadn't been a reviewer, I would never have done this. Given that Nature papers are supposed to be for broad readership, it seems that one should avoid writing for a narrow audience.

Why not just say what was found, in plain language, which was: neurons are tuned to a single latent variable that evolves under simple and intuitive dynamics? And point out why this is an important, and unexpected, finding.

Reply:

We thank the reviewer for this honest and valuable feedback. **We rewrote Introduction from the ground up to make our results more accessible to the broad audience of Nature.**

And now I have lots of small, sometimes technical comments. As an author, I find that replies to reviewers are often longer than the paper, so feel free not to reply directly to these. Maybe reply only to the ones where you thought my suggestion was really bad?

1) Please put the figures in main text where they belong. I almost never review papers with the figures at the end; this one slipped by.

Reply:

We sincerely appreciate this feedback on figure placement. While we understand that preferences vary, we kindly ask for the reviewer's understanding as we prefer to keep the figures at the end of the manuscript.

2) When you say "see Methods", please say where. It's a bit hard to have to look at all of Methods to find something.

Reply:

We added references to the specific Methods subsections throughout the main text.

3) In Figs. 3d and g, Phi is almost identical. Is this typical? If so, you should comment on that, as it's pretty amazing. If not, it would be good to show example neurons where the Phi's are obviously different.

Reply:

Indeed, the observation that the potential shape was almost identical across neurons despite heterogeneous trial-averaged firing rates is a central finding in our paper. To strengthen this observation, we quantified the similarity of potential shapes using correlation coefficients between all pairs of potentials for single neurons and populations in each stimulus condition. High values of the correlation coefficient indicated that the potential shapes were indeed almost identical across all single neuron and population models (average correlation coefficient, mean \pm std; monkey T: single neuron models 0.86 ± 0.16 , population models 0.93 ± 0.09 ; monkey O: single neuron models 0.88 ± 0.14 , population models 0.88 ± 0.14). **We present these results on lines 153–155 and on lines 223–226 in the revised Results.**

4) l 201-4, "The dynamics we found in PMd are qualitatively distinct from the stepping and ramping models proposed previously [11,21,24,25 and, instead, are consistent with the attractor mechanism hypothesized in neural circuit models [30-35]."

First, I don't understand this. Under your model, firing rates of neurons change continuously in time, so it rules out the stepping model. But why does it rule out the ramping model? Is it because the firing rates aren't monotonic in time? You should expand on this. (Although I'm not sure why it's included; it seems beside the point. Maybe it's just to irritate Mike Shadlen and Jonathan Pillow? ;))

Reply:

Our flexible modeling framework spans a continuous space of hypotheses about neural dynamics, each defined by a different shape of the potential. In this hypothesis space, the previously proposed ramping and stepping models are contained as special cases (Supplementary Fig. 4). The ramping model assumes that on single trials neural dynamics evolve gradually towards a decision boundary as a linear drift-diffusion process, which mathematically corresponds to a linear potential with a constant slope in our framework. The stepping model assumes that on single trials neural activity abruptly jumps from the initial to a final state representing a choice, which corresponds to a potential with two barriers such that trajectories have to jump over a barrier to reach either decision boundary. Previously, we showed that

our approach can reliably identify the potential shapes corresponding to the ramping and stepping dynamics from a realistic amount of synthetic data with known ground truth (Genkin et al., *Nat Commun* 2021). However, there is no *a priori* reason why neural dynamics in the brain must adhere to either of these *ad hoc* hypotheses. Indeed, the single-barrier potential that we discovered from the PMd data reveals dynamics qualitatively distinct from both ramping (linear potential with no barriers) and stepping (potential with two barriers) hypotheses.

Our finding that PMd dynamics differ from both ramping and stepping hypotheses is important to discuss within the paper because the ramping versus stepping debate has prominently influenced the field of decision-making. Our results offer a surprising yet parsimonious resolution of this debate: neither of these two *ad hoc* hypotheses is consistent with the neural dynamics in PMd. This finding highlights the conceptual advance of our flexible approach exploring a continuous hypothesis space over the conventional model comparison between a discrete set of *a priori* chosen alternative hypotheses. A likely pitfall is that no *a priori* guessed hypotheses may be correct, as we found for the ramping and stepping dynamics. However, the conventional model comparison does not offer a procedure for rejecting all alternative hypotheses or indicating what hypothesis is missing in the compared set. Possible outcomes of the conventional model comparison are limited to the set of *a priori* guessed hypotheses without guarantees that any of them faithfully describes the data. Thus, our results show that the ramping versus stepping debate was ill-posed at least in PMd where dynamics are inconsistent with either of these two influential *ad hoc* hypotheses.

We clarify these points in the new Supplementary Fig. 4 and on lines 384–400 in the revised Discussion.

Second, it's not obvious why your model is an attractor network. Given that you have absorbing boundaries, it would seem hard to even tell if it's an attractor network.

Reply:

In dynamical systems, an attractor is a set of states towards which a system tends to evolve for a wide variety of initial conditions. In models inferred from the PMd data, all initial states to the left of the potential maximum evolve towards the left boundary, and all initial states to the right of the potential maximum evolve towards the right boundary. Thus, the potential maximum is an unstable fixed point separating the basins of two attractors located at the boundaries. It is indeed true that due to the reaction-time task design, each trial ends as soon as the trajectory reaches one of the boundaries, hence we cannot directly observe the persistence of the attractor states. However, we find potential shapes much the same as in the PMd data in a spiking network model which provably has two attractors, which reinforces the conclusion that dynamics identified by our models in PMd arise from the attractor mechanism.

5) Including the spiking network model doesn't seem to bolster your case, since for that model the latent variable is 2-D. And it doesn't really add anything, in my opinion. If I were you I would take it out. But that's totally optional.

Reply:

We use the classical spiking neural network model of decision-making to establish a mechanistic interpretation of the one-dimensional decision variable and dynamics identified by our model in PMd. To answer the reviewer's question, we must consider three distinct types of dimensionality: the number of variables in a neural network model, intrinsic dimensionality of a neural manifold, and linear dimensionality of a neural manifold.

Number of variables in a network model. The spiking neural network model is a high-dimensional dynamical system described by $3N_E+2N_I$ variables, which are membrane potentials and synaptic conductances of N_E excitatory and N_I inhibitory neurons. Using mean-field approximation, the spiking network can be reduced to a two-dimensional dynamical system in which the variables are the average NMDA conductances of two choice-selective excitatory populations. This two-dimensional model was developed for the ease of interpreting flow fields on a phase plane, not because two is the correct number of variables to describe mean-field dynamics of the spiking network. In fact, a more accurate mean-field approximation of the spiking network includes 11 variables: AMPA and NMDA conductances of three excitatory populations (two choice-selective and one non-selective), GABA conductance of the inhibitory population, and firing rates of all four neural populations (Eqs. 3–7 in Wong & Wang, *J Neurosci* 2006).

Intrinsic dimensionality of a neural manifold. In both the spiking network model (defined by $3N_E+2N_I$ variables) and its mean-field approximation (defined by 2 variables), the trajectories arising during decision-making are confined

to a nonlinear manifold with intrinsic dimensionality equal to one. Intrinsic dimensionality is the minimal number of continuous variables necessary to parameterize the manifold (Ostojic & Jazayeri, *Curr Opin Neurobiol* 2021; Langdon et al., *Nat Rev Neurosci* 2023). On each trial, network trajectories start at a symmetric low-activity state and follow nearly one-dimensional stereotypic paths to reach either of the two choice attractors. When varying stimulus difficulty, the shape of these one-dimensional paths remains nearly invariant and only the speed and direction of dynamics along these paths change, consistent with our findings in the PMd data. One-dimensional trajectories traced by the network during decision-making can be parametrized by a single latent variable, corresponding to the decision variable in our model. Thus, tuning functions to the decision variable in our model capture the geometry of neural trajectories during decision-making and the potential describes the dynamics along these trajectories.

Although the spiking network model has a high-dimensional state space and the mean-field model has two-dimensional state space, trajectories arising during decision-making do not explore the entire state space of these dynamical systems. In principle, appropriately designed inputs can drive the system to any corner in its state space. For example, a strong excitatory input delivered to both choice-selective populations can stabilize a symmetric high-activity state. However, such inputs do not occur during normal decision-making behavior and the corresponding regions of the two-dimensional state space are never visited. As a result, the intrinsic dimensionality of the neural activity manifold is lower than the dimensionality of the network state space.

Linear dimensionality of a neural manifold. The tuning functions in our model describe how firing rates of single neurons change along the trajectories taken by the network during decision-making. Thus, the tuning functions capture the geometry of decision-making trajectories in the population state space. Since these trajectories are nonlinear, the linear dimensionality of the manifold they form is greater than one. The linear dimensionality is the smallest number of orthogonal directions that span a linear subspace containing the manifold (Ostojic & Jazayeri, *Curr Opin Neurobiol* 2021; Langdon et al., *Nat Rev Neurosci* 2023). In the spiking network model, choice-selective excitatory neurons have only two types of tuning functions, hence the decision manifold they form spans two linear dimensions. In PMd, tuning functions are heterogeneous, and the manifold has higher linear dimensionality.

In summary, the spiking network model enables us to establish a mechanistic interpretation of the one-dimensional decision variable and dynamics identified in PMd by our model. In both spiking network and PMd data, the decision-related activity evolves along trajectories that form a manifold with intrinsic dimensionality close to one. This manifold is parametrized by the one-dimensional decision variable uncovered by our model. The potential governing the dynamics along this manifold shows the same signatures of the attractor mechanism in the spiking network model and PMd. However, the linear dimensionality of the manifold is higher in PMd than in the spiking network due to heterogeneous response profiles of PMd neurons. **We clarified these points in the new Supplementary Note 3.2 and on lines 287–348 in the revised Results.**

6) l 230: *what does representation geometry mean here?*

Reply:

We revised this sentence to read “Thus, the classical attractor network and PMd data have similar dynamics but different geometry of neural trajectories...” **on lines 346–347 in the revised Results.**

7) l 254-6: *"our results suggest that the dynamics of internal cognitive computations may differ across brain areas and thus cannot be described by the same global variable inferred from behavior."*

I can't parse this sentence. I think the problem is the word "same". Same as what?

Reply:

We removed this sentence in the revised manuscript.

9) *Under the change of variable $t = s/D$, Eq. 1 becomes*

$$dx/ds = -d\Phi/dx + \sqrt{2} \xi(s)$$

So D has no effect on the dynamics, other than to scale time and Φ . It doesn't matter for anything you did. But I'm curious why you put a D in front of $d\Phi/dx$. It seems non-standard.

Reply:

The parametrization of Eq. 1 in which the noise magnitude D scales the potential is a convenient choice which makes the equilibrium probability density of the Langevin dynamics invariant to the noise magnitude. This parametrization does not restrict in any way the space of dynamical systems spanned by our models. One can think about this parametrization as measuring the potential height in units of D . Our choice of this parametrization is historical and has no deep meaning. **We clarified this point on lines 486–490 in the revised Methods.**

10) paragraph starting on l 337: it's a little confusing to drop the subscript k : the first and second $Y(t)$ look the same but mean different things. Maybe call the first one \mathcal{Y} ?

And this is even more confusing in the next paragraph, which contains $Y(t)$. Which $Y(t)$ is it? Presumably a single trial. If so, is that the single trial likelihood?

Reply:

We fixed the notation as suggested by the reviewer on lines 501–506 in the revised Methods.

11) l 352 and Eq. 3 (and other places): should it be $P(X(t), Y(t)|\theta)$ rather than $P(X(t), Y(t))$?

Reply:

We omit conditioning on θ in expressions for the probability densities of the data and latent states to simplify notation. **We clarified this point on lines 516–517 in the revised Methods.**

12) Bins with no spikes seem to be missing from Eq. 4. I was expecting factors of $(1 - f_{i,j}(x_{t_j}) dt)$ whenever neuron i_j didn't fire. Or am I misinterpreting things?

Reply:

We wish to clarify that we do not bin spikes, but process data spike-by-spike in continuous time. Our likelihood calculation accounts for the absence of spikes during each interval between adjacent spike observations via the transition probability density $p(x_{t_j} | x_{t_{j-1}})$. This transition probability density satisfies a modified Fokker-Planck

equation Eq. 6, in which the term $-\sum_{i=1}^M f_i(x)$ accounts for the probability decay due to spikes emitted by any neuron in the population. In other words, $p(x_{t_j} | x_{t_{j-1}})$ includes only trajectories consistent with no spikes emitted between t_{j-1} and

t_j . We have derived this modified Fokker-Planck equation in Supplementary Note 5 in Genkin et al., *Nat Commun*, 2021. **We clarified this point on lines 539–552 and 557–563 in the revised Methods.**

13) Eq. 5: I'm assuming M is the number of neurons. If you didn't say this somewhere, you should.

Reply:

We clarified that M is the number of simultaneously fitted neurons on lines 509–510 in the revised Methods.

14) In Eq. 5, the probability decays. You say on line 371 that that's "due to spikes emitted by any neuron in the population". But why should spike cause decay? I thought the only decay was due to absorption on the boundary.

Reply:

The likelihood calculation needs to account for the spikes observed at times $Y(t)=\{t_j\}$ and also for the absence of spike observations during interspike intervals (t_{j-1}, t_j) . In Eq. 5, the terms $p(y_j | x_{t_j})$ represent the probability density of observed spikes, and the terms $p(x_{t_j} | x_{t_{j-1}})$ represent the transition probability density of latent states that accounts for the absence of spike observations during interspike intervals. This transition probability density decays with time at

a rate given by the Poisson firing rate of all neurons, because it becomes less likely to observe no spikes for longer time intervals. The decay of $p(x_{t_j} | x_{t_{j-1}})$ with time is analogous to the term $\exp(-\int_{t_{j-1}}^{t_j} \lambda(t) dt)$ in the standard likelihood formula for the inhomogeneous Poisson process with known firing rate $\lambda(t)$ (Dayan & Abbott, *Theoretical Neuroscience* 2001). The difference is that $p(x_{t_j} | x_{t_{j-1}})$ integrates over all latent trajectories consistent with the data because the true firing rate is unknown. We rigorously derived Eq. 5 in Supplementary Note 5 of our previous paper (Genkin et al., *Nat Commun* 2021). **We clarified this point on lines 539–552 in the revised Methods.**

15) *Speaking of absorption, when x hits a boundary, is the trial over? Or does the trial keep going, with firing rates $f_i(+1)$? That seems important for inference, because if the trial ends when x hits a boundary, then on each trial you have to choose a path that ends when that trial is over. Techniques exist to do that, but (after glancing at SI), I don't think you do it. Please clarify.*

Reply:

Since we model a reaction time task, the trial ends when the latent trajectory reaches the boundary for the first time. Therefore, as the reviewer correctly points out, the inference must integrate over all latent paths that terminate at one of the boundaries at the trial end time t_E and do not reach a boundary at earlier times. Two components in our framework account for the statistics of latent trajectories that satisfy these constraints. First, the absorbing boundary conditions ensure that latent trajectories reaching a boundary before the trial end do not contribute to the likelihood. Second, the absorption operator A enforces that the likelihood includes only trajectories terminating on the boundaries at the trial end time. Without the absorption operator, the likelihood includes all trajectories that terminate anywhere in the latent space without reaching the domain boundaries before the trial end. Omitting either of these components results in incorrect inference, in which erroneous features arise in the dynamics due to nonstationary data distribution. We dedicated an entire previous paper to the inference of nonstationary Langevin dynamics, in which we derive all equations and validate the inference on synthetic data with known ground truth (Genkin et al., *Nat Commun* 2021). **We clarified this point on lines 518–530 in the revised Methods.**

16) *What does $\hat{H}(t_j - t_{j-1})$ mean?*

Reply:

The notation $\exp(-\hat{H} \cdot (t_j - t_{j-1}))$ states for the exponential operator that propagates the latent probability density forward in time from t_{j-1} to t_j , which is a solution of the modified Fokker-Planck equation Eq. 6. **For clarity, we added a symbol “ \cdot ” in this expression on line 564 in the revised Methods.**

17) l 383: *why that particular choice for C ? And is the integral from -1 to 1?*

Reply:

Yes, the integral is from -1 to 1 . C is an arbitrary integration constant. The choice of C is not important as long as it is the same across all models. **We clarified this point on lines 578–579 in the revised Methods.**

18) *It would be very helpful if all important equations were displayed, not inline. In particular, (but not limited to!) the two expressions on line 382, which should definitely be displayed. Then you could refer back to them when you write down Eqs. 7 and 8 (which took me a long time to figure out). Most people (meaning me) do not remember definitions.*

Reply:

We edited the text to display the important equations as suggested by the reviewer **on lines 576–588 in the revised Methods.**

Referee #2:

I understand better the novelty of the manuscript, but I still have the same general hesitations. I hate to sound critical because I did like the paper and the modelling framework. I strongly agree that it is important to infer latent neural dynamics from data. They convincingly show that neural population activity and single neuron activity encodes some kind of decision variable that is consistent with attractor dynamics. What it's missing are (in order of decreasing importance): 1) causal perturbation tests to establish the attractor dynamics are correct, 2) a mechanistic interpretation of the model, and 3) a real link between dynamics and geometry as suggested in the title. I would appreciate if the authors would better demonstrate or defend these points.

Reply:

We thank the reviewer for these additional comments, which prompted us to make three key improvements to the manuscript. First, we further validated our modeling approach using optogenetic perturbation data. Second, we developed a low-rank recurrent network model to demonstrate a mechanism by which diverse tuning to the decision variable arises from distributed connectivity. Finally, we clarified the confusion about perceived link between dynamics and geometry in the title. Together, these revisions further solidify the rigor of our findings and enhance the clarity of their presentation in the paper.

1) I don't agree that the "results provide a direct demonstration of the attractor hypothesis in experimental data" without a perturbation test.

Reply:

We agree that the wording of this quoted statement from line 399 in Discussion of the previous revision is imprecise. This phrase appeared within the context of contrasting the traditional model comparison approach, limited to few *a priori* hypotheses (e.g., ramping versus stepping dynamics), with our nonparametric approach, which spans a continuous hypothesis space within a single model architecture, enabling flexible dynamics discovery directly from data. **We modified this text to read** "our approach discovers the attractor hypothesis directly from experimental data and reconciles it with heterogeneous single-neuron responses" **on lines 417–419 in revised Discussion.**

The model's ability to infer other kinds of dynamics is validated Supp Figure 4 is in synthetic data, but inferring dynamics from real data is a different story, and notoriously difficult to do without testing other dynamical regimes (e.g., O'Shea et al show that the wrong dynamics are inferred with LDS if perturbation data is not included).

Reply:

O'Shea et al. (*bioRxiv* 2022, 2022.12.16.520768) fit an LDS to trial-averaged neural activity, which is the primary reason their model fails to correctly identify single-trial dynamics when perturbation data are not included. Trial-averaged neural activity does not uniquely determine a model of population dynamics on single trials (Galgali et al., *Nat Neurosci* 2023). In particular, O'Shea et al. observe that trial-averaged activity evolves slowly along a specific dimension. This pattern may emerge in a normal linear dynamical system where this dimension aligns with an eigenvector characterized by a slow time constant. Alternatively, it may arise in a non-normal linear dynamical system, where this dimension has a fast time constant but receives transient feedforward inputs from activity along other eigenvectors. Trial-averaged data cannot distinguish between these two equally plausible explanations, because averaging obscures how quickly impulse responses along individual eigenvectors decay. O'Shea et al. resolve the ambiguity using perturbation data, which reveal that perturbations along this dimension decay rapidly, consistent with non-normal dynamics.

Comparable insights can be obtained without external causal perturbations by analyzing single-trial activity, which inherently contains transient deviations from the average that act as naturally occurring perturbations within the repertoire of activity patterns produced by the dynamical system (Galgali et al., *Nat Neurosci* 2023). Therefore, the

reviewer's concern does not apply to our work, as *we fit neural activity on single trials*, unlike trial-averaged activity used by O'Shea et al. **We clarified this point in new Supplementary Note 3.2 in the revised manuscript.**

By the way, I believe the authors partially misunderstood my point about Mante's work. I know that they didn't fit RNNs to neural data, but they fitted ("trained" if you prefer) RNNs to perform the task, then compared it to data.

Reply:

We did not misunderstand the reviewer's point; rather, we emphasized the *vast gap* between analyzing dynamics in task-optimized RNNs and single-trial experimental data. Identifying a dynamical mechanism in a task-optimized RNN does not prove that this mechanism operates in the brain, especially when based solely on a *qualitative resemblance* between low-dimensional projections of trial-averaged trajectories in the RNN and neural recordings. In our previous response, we stated three fundamental reasons for why such qualitative resemblance is insufficient to draw conclusions about dynamical mechanisms in neural data.

What I meant here is that I felt that I learned a new dynamical mechanism for contextual decision making for that task.

Reply:

In the specific case of Mante et al., TDR finds projections of population activity in which irrelevant stimuli are equally represented in both contexts, seemingly invalidating the well-known mechanism for context-dependent decisions based on suppression of irrelevant stimulus representations. Mante et al. find a selection-vector mechanism in RNNs that apparently also does not require suppression of irrelevant stimuli based on the TDR projections of RNN responses. However, we have recently shown that this selection-vector mechanism is merely a local linear description of the well-known mechanism based on suppression of irrelevant stimulus representations (Langdon & Engel, *Nat Neurosci*, 2025 in press, to appear at <https://doi.org/10.1038/s41593-025-01869-7>). The apparent contradiction with Mante et al. arises here precisely because their analysis of dynamical mechanisms in RNN is decoupled from the TDR used for comparing representations between the RNN and neural data. Since TDR does not model task computations, it provides a misleading picture of the mechanism (Langdon & Engel, *Nat Neurosci*, 2025 in press). This example illustrates that one cannot reliably identify dynamical mechanisms in data via a qualitative comparison with task-optimized RNNs using methods like TDR that do not model dynamics.

Of course it is important to validate classic attractor hypothesis from data, but even there, without a perturbation test I don't think the result is as definitive as the authors suggest.

Reply:

We wish to clarify that our study does not aim to validate the classic attractor hypothesis. Instead, we propose and test a *novel hypothesis* that complex responses of single neurons in association brain areas arise from the same coding principle as in sensory areas: the neural population dynamics encode simple cognitive variables, while individual neurons have diverse tuning to the cognitive variable, similar to neural tuning curves for sensory stimuli (**lines 18–21 in Introduction**). Our main conclusion—the confirmation of our hypothesis—remains valid regardless of the dynamics governing the decision variable. For example, had we discovered that the decision variable follows drift-diffusion or any other dynamics, it would not change our main conclusions at all.

Moreover, attractor dynamics are not our *a priori* hypothesis, but a surprising discovery, not anticipated from examining heterogeneous trial-averaged responses of single neurons. In contrast to traditional approaches that compare a small set of *a priori* hypotheses about neural dynamics, our modeling framework spans a continuous hypothesis space, enabling flexible discovery of neural dynamics directly from data. While the reviewer argues that attractor dynamics are not novel, in our view, the simplicity of the discovered dynamics is a beautiful instance of parsimonious explanation, made possible by our novel hypothesis that single neurons have diverse tuning to the decision variable, along with our unique computational approach.

This is my strongest criticism so if the authors have any perturbation data to include I would certainly reconsider.

Reply:

We thank the reviewer for encouraging us to further validate our computational approach using perturbation data. We agree that validating our approach in experimental data, where the ground truth has been established via causal perturbations, will strengthen confidence in the accuracy of our results.

Therefore, we analyzed an additional dataset ideally suited for such validation (Inagaki et al., *Nature* 2019). In this experiment, neural population activity was recorded from the anterior lateral motor cortex (ALM) in mice performing a delayed response discrimination task. The study used bilateral optogenetic inactivation in ALM to distinguish between three alternative hypotheses about the mechanism of memory maintenance during the delay period: line attractor, single attractor, or multistable discrete attractor dynamics. The observed recovery patterns of trial-averaged neural responses following optogenetic perturbations were consistent with the multistable discrete attractor hypothesis. Although this study did not analyze neural dynamics on single trials, the perturbation results establish multistable discrete attractor dynamics as the ground truth during the memory delay period of the task. We therefore tested whether our modeling approach could correctly identify multistable attractor dynamics from the observational data alone, without using perturbation data for model fitting.

We fitted our model to the delay-period population spiking activity on unperturbed trials. Our model robustly discovered a potential with multiple discrete attractors, in agreement with conclusions derived from optogenetic perturbations. These results confirm that our approach accurately identifies the ground-truth dynamics in experimental data by analyzing single-trial spiking activity, without using causal perturbation data for model fitting. Note that it makes no difference that the perturbation tests were conducted before model fitting. Had we first fitted our model and discovered multistable discrete attractors in ALM, and then Inagaki et al. performed their perturbation experiments to test this hypothesis derived from data, all conclusions would remain the same.

We present these results in new Supplementary Fig. 6, new Supplementary Note 3.2, and on lines 83–85 in revised Results.

The next two criticisms are more related to interpretation, writing, and presentation of their model.

2) The authors did not understand my second point about Mante that “RNNs are a type of neural circuit model”. I don’t agree that their model is more mechanistic than an RNN, mainly because I do not believe that tuning curves in a latent variable constitute a mechanistic model. This is a recurring theme in the paper that I frankly find difficult to accept.

Reply:

We are unsure how this confusion arose. As far as we know, we never stated that our model is “more mechanistic than an RNN” nor that tuning curves “constitute a mechanistic model”. If the reviewer perceives this to be a recurring theme in the paper, we would greatly appreciate references to specific lines in the text that led to this impression. This feedback would help us make targeted clarifications to prevent other readers from getting similarly misled.

A spiking neuron is simply not an inhomogenous Poisson process from some population wide latent variable. Even if that is a useful framework. This is why I find the mechanistic interpretation limiting.

Reply:

We thank the reviewer for this comment, which prompted us to better explain the connection between our latent variable model and neural circuit structure. Previously, we established a mechanistic interpretation of our modeling framework by linking it to a spiking network model of decision-making (Fig. 5a-f). In the revised manuscript, we further deepened this connection by demonstrating how diverse tuning to the population-wide decision variable arises from the interplay between recurrent connectivity and firing-rate nonlinearity in low-rank RNNs (**new Fig. 5g-k and new Supplementary Fig. 19**), as explained in detail in our response to question 3 below. These new results demonstrate that recurrent networks can generate identical dynamics with distinct population geometries, highlighting the power of our computational framework in discovering the dynamics and geometry directly from data, free of prior mechanistic assumptions.

3) I do not agree with the authors that the geometry of neural tuning curves generates neural geometry. I know and love the Kriegeskorte review. But what I mean is that, causally, mechanistically, the tuning curves are not really generating

neural geometry. It's the neural circuitry that generates the geometry of neural trajectories, and those generate the tuning curves.

Reply:

We absolutely agree that neural response properties arise from circuit dynamics governed by recurrent connectivity, as we emphasized in our recent perspective article (Langdon et al., *Nat Rev Neurosci* 2023). However, neither tuning curves nor population geometry is inherently closer to this mechanistic origin. Population geometry does not generate tuning curves, nor do tuning curves generate population geometry. Neither has a causal relationship to the other; both are merely equivalent descriptions of the same data. We presume that some confusion may have arisen here because traditionally neural tuning/selectivity during cognitive tasks has been characterized with respect to *static external task variables* (e.g., stimulus coherence or choice), whereas we introduce tuning functions to *internally generated dynamic variables*. We resolve this confusion in three steps in new Supplementary Fig. 1 and below in this response. First, we explain the duality between population geometry and tuning curves for sensory stimuli. Next, we discuss the conventional tuning/selectivity for external task variables, which lack temporal consistency and a reciprocal relationship with neural population trajectories. Finally, we introduce tuning functions for internally generated dynamic variables, which restore the duality with population geometry, following the same geometric principle as for sensory stimuli.

Tuning curves and population geometry provide dual views on the same neural responses: tuning curves at the level of single neurons, and population geometry at the level of population activity patterns. To understand this duality, consider time-averaged responses of N neurons to K static stimuli, arranged in an $N \times K$ neural response matrix (Supplementary Fig. 1a). The columns of this matrix define the neural population state space, where each axis represents activity of one neuron and each point represents a stimulus, collectively forming population geometry. The rows of the same matrix define the tuning curves of individual neurons to the stimuli. The neural response matrix is fully determined by specifying either all of its columns or all of its rows. Hence, the tuning curves of all neurons in the population uniquely define the geometry of stimulus representation in the neural population state space.

While the duality between neural tuning and population geometry is well established for encoding of sensory stimuli (Kriegeskorte & Wei, *Nat Rev Neurosci*, 2021), analogous frameworks have been lacking for the representation of dynamic cognitive variables that transiently unfold over time in single-trial neural activity. Conventionally, neural responses during cognitive tasks are characterized with respect to *static external task variables*, e.g., stimulus coherence or choice (Supplementary Fig. 1b). However, single neurons have complex temporal response profiles, such that their selectivity for static external variables often changes significantly over time. For example, a neuron may show higher firing rate for left choices early in the trial but switch its preference to right choices toward the trial end. The temporal inconsistency of neural tuning/selectivity for static external task variables led to a widespread view that cognitive computations arise from complex population dynamics, while tuning/selectivity of single neurons is not interpretable.

Contrasting this view, we propose and test a novel hypothesis which unifies the neural encoding of sensory stimuli and dynamic cognitive variables. Rather than selectivity for static external task variables, we consider neural tuning to *internally generated dynamic variables* that implement cognitive computations. These tuning functions uniquely specify the representation geometry of the internal dynamic variable in the population state space, precisely as they do for sensory stimuli. For example, the value of the decision variable at any given time is encoded by the position of population activity along a one-dimensional manifold in the population state space (Supplementary Fig. 1c). The population trajectories traverse this manifold as they evolve toward one or another choice on each trial. The geometry of these population trajectories is uniquely specified by tuning functions of single neurons to the decision variable, with the exact same dual relationship between neural tuning and population geometry as for sensory stimuli. In addition, the dynamics model captures the single-trial dynamics of the decision variable, which govern time evolution of population activity along the manifold on single trials.

We clarified this point in new Supplementary Fig. 1, new Supplementary Note 3.1, on lines 8, 16–17, 21 in revised Introduction and on line 63 in revised Results.

I can accept that allowing for heterogeneous tuning curves in the model are important for fitting neural dynamics but I really know how to interpret this without any mechanistic or even normative reason for why there should be such heterogeneous tuning curves.

Reply:

We thank the reviewer for this comment, which motivated us to directly demonstrate how diverse tuning to internally generated dynamic variables can emerge in recurrent network models with distributed connectivity.

Mechanistically, heterogeneous tuning curves—and, equivalently, complex nonlinear population trajectories—emerge from the interplay between recurrent connectivity and firing-rate nonlinearity of single neurons. This mechanism is intuitive to understand in low-rank RNNs. A rank- k connectivity matrix defines k mean-field variables $z=(z_1, \dots, z_k)$ governing population dynamics, which generate low-dimensional trajectories in the space of synaptic currents. The firing rate of each neuron is a nonlinear function of its input current, which is a one-dimensional projection of the variables z determined by its connectivity. In networks with distributed connectivity, these one-dimensional projections are diverse, resulting in heterogeneous inputs across neurons. Combined with the firing-rate nonlinearity, heterogeneous inputs generate diverse nonlinear tuning to internal dynamic variables.

To illustrate how this mechanism generates diverse tuning to the dynamic decision variable, we designed a rank-two recurrent network that replicates the classical attractor dynamics with distributed connectivity (Fig. 5g-k). The flow field governing the mean-field variables z_1 and z_2 matches that in the classical mean-field attractor network, generating similar trajectories parametrized by a one-dimensional decision variable x . In the firing rate space, single units in the low-rank RNN have diverse nonlinear tuning to the decision variable x , and the decision manifold has high-dimensional geometry, as in our PMd data. This example shows that recurrent networks can generate identical dynamics with distinct population geometry, highlighting the power of our computational approach in discovering dynamics and geometry directly from data, free of prior mechanistic assumptions.

We present these results in new Fig. 5g-k, new Supplementary Fig. 19, on lines 352–370 in revised Results and lines 949–980 in revised Methods.

While low-rank RNNs are valuable models for understanding how recurrent circuits can generate heterogeneous tuning functions, brain circuit connectivity may not necessarily be low-rank. Low-rank solutions are just one of many candidate mechanisms, and it remains unknown whether low-rank connectivity underlies neural manifolds in the brain. Our modeling framework identifies the dynamics and geometry of neural representations without imposing an inductive bias toward a specific circuit mechanism that generated them. Such statistical descriptions are extremely valuable, as they reveal principles of neural coding and computation and allow for precise quantitative comparisons between distinct mechanistic models and experimental data, enabling the testing of alternative hypotheses about circuit mechanisms generating task-relevant dynamics in the brain (Williamson et al., *Curr Opin Neurobiol* 2019).

We clarified this point on lines 432–441 in revised Discussion.

Finally, since the geometry is only generated through these tuning curves (as opposed to being shaped by high-dimensional neural dynamics) I don't see the link between geometry and dynamics that is suggested in the title. At the least I think the authors should have a more accurate title.

Reply:

To the best of our understanding, our title does not suggest a link between dynamics and geometry. If the reviewer interprets such a link from the title, we would appreciate further clarification on how this link might be implied. This feedback would help us make revisions to prevent other readers from being similarly misled.

Our modeling framework *dissociates* dynamics and geometry and enables estimating both simultaneously in single-trial experimental data. We agree that the population dynamics and geometry are mechanistically intertwined via circuit connectivity in each specific network. Yet, dynamics do not uniquely determine population geometry: recurrent networks can generate identical dynamics with distinct population geometries, as we show in revised Fig. 5 (see also Pezon et. al, *bioRxiv*, 2024). Accordingly, our framework enables flexible inference of dynamics and geometry without implying a specific rigid link between them.

We wish to emphasize the distinction between a statistical description of population dynamics and geometry and their mechanistic origin in circuit connectivity. Our model identifies dynamics and geometry of neural representations in single-trial experimental data, which per se does not require grounding in circuit connectivity. As an analogy, understanding stimulus representations in sensory areas does not require a model of the network mechanism that generated these representations (e.g., Chang & Tsao, *Cell* 2017). Statistical descriptions are just as valuable as

mechanistic models, because they reveal principles of neural coding and computation and enable precise quantitative comparisons between mechanistic models and experimental data (Williamson et al., *Curr Opin Neurobiol* 2019).

Referee #3:

Besides the fact that the authors continue to use geometry and didn't put the figure in the main text, this is a beautiful paper. I have only a few comments, almost all of them about places where I got confused. I'll leave it to the authors whether they want to make any changes. If they don't, that's fine; as far as I'm concerned, the paper could be published in its present form.

Reply:

We sincerely appreciate the insightful and thorough feedback of the reviewer, which helped significantly to strengthen our results and elevate the clarity of their presentation in the paper

1. l 64-5: "Our modeling framework can generate data with identical dynamics but different geometry".

OK, I can't resist. Here "geometry" just means "tuning curves", right? Which I got only from looking at SI. This sentence -- and, indeed the rest of the paragraph -- would be so much easier to make sense of if you replaced "geometry" with "tuning curves". As mentioned at the start of my review, you're free to ignore this, but it's an important paragraph and it would be helpful if it were crystal clear. And I promise to never mention geometry again. ;)

Reply:

We agree with the reviewer that the duality between tuning curves and population geometry is an advanced theoretical concept, which may be unfamiliar to many readers. Moreover, our paper is the first to establish this duality for dynamic cognitive variables, revealing a unifying geometric principle for neural encoding of sensory and dynamic cognitive variables. Since these concepts are central to our main findings but not widely known, **we added a brief tutorial explaining the relationship between neural tuning and population geometry in new Supplementary Fig. 1, new Supplementary Note 3.1, on lines 8, 16–17, 21 in revised Introduction and on line 63 in revised Results.** We hope that this background information will assist readers in better understanding our findings.

2. I know the title on line 87 is "Decision dynamics in single neurons", but that didn't stop me from getting confused; for some reason I thought you used the same $\Phi(x)$ for all neurons within a population. Rereading the paper I now see that that's not what you did, but it took me a while to figure out. I think there's an easy fix: in this section, instead of writing $\Phi(x)$, write $\Phi_i(x)$. If you did that everywhere Φ shows up, and also added it to the section starting on line 87, things would be a lot more clear.

Reply:

We felt that introducing the index i for the potential of each neuron may cause confusion, as it would then require introducing an additional index to label the potential for each stimulus condition for completeness, making the notation unnecessarily complex. Therefore, **we added the text** “we first fit the model for each stimulus condition separately, with a separate potential $\Phi(x)$ fitted to the activity of each neuron in each stimulus condition” **on lines 94–96 in revised Results.** We hope this revision clarifies the confusion.

3. l 217-9: "Since the population model predicted spike times just as well as single-neuron models, we expect that it recovered the same dynamics as identified in single neurons. Indeed, the dynamics discovered by the population model were consistent with single-neuron results (Fig. 4b-d)."

I was expecting a comparison between single neuron and population fits; either tuning curves or inferred trajectories, $x(t)$. But I didn't get that. I think a direct comparison would be helpful.

Reply:

We added a direct comparison between tuning functions in the population and single-neuron models on lines 223–226 in revised Results. The inferred tuning functions were similar between the population and single-neuron models, as indicated by a high value of their correlation coefficient (monkey T: 0.89 ± 0.12 , $n=85$; monkey O: 0.89 ± 0.15 , $n=46$, mean \pm std across neurons).

4. Fig. 4e. An insanely minor point: if you could move panel c to where panel e is, and put panel e at the end, that would help people like me who don't pay much attention (when you said Fig. 4b-d, I spent a while looking at panel e, which kind of confused me). I know, not really enough room, but you might be able to shrink panel a.

Reply:

We designed this figure layout to place panels c and d next to each other, because they show two parts—dynamics and geometry—of one fitted population model. The change suggested by the reviewer would break this cohesion. Therefore, we addressed this comment with text revisions on lines 222–230 in revised Results. We removed the reference to the collection of panels (Fig. 4b-d), and instead referred in the text to each panel individually. We hope this change will eliminate the confusion.

Referee #2:

Thanks to the authors for the new analyses. The optogenetic data from Inagaki et al is a nice validation of their method, and the low rank RNN adds a lot of value to the manuscript by providing mechanistic insight regarding heterogeneous tuning. Overall the authors have made some efforts to address the mechanistic and robustness issues I raised previously. I believe these efforts improve the manuscript and I think the method is interesting and the analyses are sound.

Reply:

We thank the reviewer again for the invaluable suggestions to validate our computational approach using optogenetic perturbation data and clarify the mechanism by which diverse tuning to a decision variable arises in recurrent networks. We are pleased that the reviewer finds our revisions have satisfactorily addressed these previously raised concerns and that these additional analyses have substantially strengthened our findings.

In contrast, I am still skeptical as to the novelty of the main pitch of the paper. On one hand, the attractor dynamics that they have found do not seem to constitute a new dynamical mechanism for decision making. The authors have responded that the real finding of their paper is “that complex responses of single neurons in association brain areas arise from the same coding principle as in sensory areas: the neural population dynamics encode simple cognitive variables, while individual neurons have diverse tuning to the cognitive variable, similar to neural tuning curves for sensory stimuli”. While I understood that this was the basis for their model, I was surprised to see that this is considered the novelty of their paper. I find the comparison to sensory systems rather weak, since cognitive tasks require computation, not simple encoding of external stimulus statistics. It’s not clear to me why, from any normative perspective, neurons should be tuned to a latent cognitive variable that is already performing the task. In the sensory system, the tuning curves are a part of the computation, for example by decomposing high-dimensional visual inputs into independent features. Here, the tuning is post computation. I don’t doubt that this framework can be useful as a phenomenological model that can infer latent dynamics - the authors have shown convincingly that this is the case. However I do not find this result in and of itself to be convincing as a computational mechanism, or even particularly exciting. Their concept of geometry is very particular to their model assumptions (see below) and not general nor mechanistic.

Reply:

A major confusion in this argument seems to stem from the reviewer’s assumption that neural responses can be divorced from cognitive computations generating “a latent cognitive variable that is already performing the task”. The reviewer’s assertion that tuning to a cognitive variable is “post computation” sidesteps a fundamental contradiction: if the computation is already complete, what substrate has carried it out?

Any system performing computation must encode, in some form, the variables it operates on. A normative reason for why neurons are tuned to the decision variable is that this variable is computed by these very neurons—the decision variable does not exist independently of the neural activity but is generated by it. Thus, tuning curves to a cognitive variable are part of the computation, much like in sensory systems. We demonstrate how decision computation can mechanistically arise in recurrent network models, with tuning functions representing the decision variable as it is being computed by the very same network (Fig. 5).

Specific comments responding to the authors’ rebuttal:

“the reviewer’s concern does not apply to our work, as we fit neural activity on single trials, unlike trial-averaged activity used by O’Shea et al”

I agree that fitting single trial activity is an improvement to fitting trial-averaged activity. However, whether it is sufficient to infer dynamics will depend on the noise statistics of the data and the robustness of the method. It is a strong

statement to say that the concern doesn't apply at all. The authors should demonstrate the robustness of their inferred dynamics in some way, possibly by using a different method.

Reply:

We have demonstrated the robustness of our method in multiple ways: (i) using synthetic data with known ground truth (Supplementary Figs. 1–5), (ii) using leave-one-neuron-out validation (Fig. 4a), and (iii) using optogenetic perturbation data (Extended Data Fig. 4, Supplementary Note 1.2). As explained in Introduction and detailed in Supplementary Note 1.4, our computational approach is unique and introduces three key innovations: (i) flexible nonparametric inference, (ii) simultaneous inference of nonlinear tuning functions and latent dynamics, and (iii) a principled approach for selecting the complexity of the inferred model. To our knowledge, no other computational method integrates all these advances to provide comparable insights or to serve as a validation of our approach.

“We are unsure how this confusion arose. As far as we know, we never stated that our model is “more mechanistic than an RNN” nor that tuning curves “constitute a mechanistic model”. “

This was a typing error on my part. I meant “as mechanistic as an RNN”, not “more mechanistic”. This is in response to the authors’ statement from the previous rebuttal: “our framework offers mechanistic insights similar to methods based on RNN fitting”. I hesitate to call the insights found in this paper “mechanistic”, and the only part of the model that could be interpreted to me as such would be the heterogeneity of tuning curves, which is one of the core hypotheses. I don’t find this to be mechanistic since it is not clear where the tuning comes from (as my point above).

Reply:

We have litigated this issue extensively in previous rounds of review. To briefly reiterate: our computational framework provides a statistical description of the data, and we use recurrent neural network (RNN) models to establish a mechanistic interpretation of the one-dimensional decision variable and dynamics identified by our model in PMd (Fig. 5, Extended Data Fig. 10).

The quoted text from our previous response appeared in the context where we clarified the distinction between dynamical mechanisms (Vyas et al., *Ann Rev Neurosci* 2020) and circuit mechanisms (Langdon et al., *Nat Rev Neurosci* 2023). Our modeling framework describes neural computation as a dynamical system—that is, a dynamical mechanism. Similarly, methods that linearize RNN dynamics around fixed points also reveal dynamical mechanisms (Mante et al., *Nature* 2013), offering insights comparable to those provided by our approach. In addition, RNNs can provide insights into circuit mechanisms through methods that uncover the connectivity structure generating task-relevant dynamics (Langdon & Engel, *Nat Neurosci* 2025). Correspondingly, we use RNNs to demonstrate how distinct connectivity structures can generate the same dynamics using either homogeneous or diverse tuning functions (Fig. 5).

“Population geometry does not generate tuning curves, nor do tuning curves generate population geometry.”

*I acknowledge the authors’ point about a duality between tuning curves and trial-averaged population geometry as per Kriegeskorte & Wei. However, this duality is lost when taking time and trial-by-trial variability into account. Neural circuits generate single-trial population trajectories, which over many trials forms some geometric structure, and these trajectories can be reduced to tuning curves by trial/time averaging and projecting onto single-neuron axes. This is the basis for my claim that the geometry of neural *trajectories* generate tuning curves.*

Reply:

We are surprised that the reviewer overlooked Extended Data Fig. 1 and Supplementary Note 1.1, which we specifically added to clarify this confusion in the previous round of review. In our framework, tuning functions are not derived from trial-averaged population trajectories, as the reviewer suggests. We introduce tuning to internally generated dynamic variables that unfold with a unique time course on single trials. Trial-to-trial variability in the dynamics of the latent variable generates trial-to-trial variability in firing rates at each time aligned to an external event (e.g., stimulus onset, Extended Data Figs. 2,3). Yet, when conditioned on a specific value of the dynamic variable, the firing rate is deterministic (Extended Data Figs. 2,3). Our tuning functions capture precisely this deterministic

dependence of the firing rate on the value of the latent dynamic variable. This definition restores the duality between population geometry and single-cell tuning, unifying neural encoding of sensory and dynamic cognitive variables.

The authors claim several times that “tuning curves of single neurons define the geometry of the population code”. I still think this is a very particular assumption that needs to be better addressed in this paper. I think this is really important because there is no standard definition for what population geometry means, and as a field it is important to be precise about these abstract concepts to avoid confusion. One simple work around would be for the authors to simply define early on what they mean by “population geometry”, and to acknowledge the other forms of geometry that are not accounted for in their model assumptions as caveats. In the ideal world they would give an example of where their model fails to recover the correct dynamics and geometry because their assumptions are not met.

Reply:

We fully agree that it is important to clearly define tuning functions, population geometry, and dynamics, as these concepts are not yet firmly established in the literature. Accordingly, we define these concepts early on in Introduction, Fig. 1, and at the beginning of Results. In addition, to help the readers, we discuss these definitions and their relationship to other definitions/uses of tuning and geometry in the literature in Extended Data Fig. 1 and Supplementary Note 1.1, which the reviewer might have inadvertently missed.

*“To the best of our understanding, our title does not suggest a link between dynamics and geometry.”
To me, “A and B in premotor cortex” as a title suggests a link between A and B. Otherwise, why would they be in the same paper?*

However this point about the title is really a detail. The point I was trying to make was about the specific type of geometry that is addressed in the paper (as in my previous point above). The paper is much stronger in capturing dynamics than capturing geometry. And I personally think a simple title like “The dynamics of choice in premotor cortex” sounds much better.

Reply:

We respectfully disagree with the assertion that our “paper is much stronger in capturing dynamics than capturing geometry”. The central hypothesis we propose and test is that heterogeneous responses of single neurons arise from their diverse tuning to the decision variable, offering a unifying geometric principle for how the brain encodes both sensory and dynamic cognitive variables. Our main conclusion—the confirmation of this hypothesis—remains valid regardless of the dynamics governing the decision variable. Therefore, removing “geometry” from the title would misrepresent our core findings.

Referee #3:

I was happy with the paper last time; I'm no less happy this time.

Reply:

We are deeply grateful to the reviewer once again for the invaluable feedback, which has significantly strengthened the quality and clarity of this manuscript.